# Dendritic autophagy degrades postsynaptic proteins and is required for long-term synaptic depression in mice

Emmanouela Kallergi[1,10], Akrivi-Dimitra Daskalaki[1,10], Angeliki Kolaxi[1], Come Camus [2], Evangelia Ioannou[3], Valentina Mercaldo[1], Per Haberkant[4], Frank Stein [4], Kyriaki Sidiropoulou [3], Yannis Dalezios[5,6], Mikhail M. Savitski [4,7], Claudia Bagni [1,8], Daniel Choquet [2,9], Eric Hosy [2] & Vassiliki Nikoletopoulou [1✉]

The pruning of dendritic spines during development requires autophagy. This process is facilitated by long-term depression (LTD)-like mechanisms, which has led to speculation that LTD, a fundamental form of synaptic plasticity, also requires autophagy. Here, we show that the induction of LTD via activation of NMDA receptors or metabotropic glutamate receptors initiates autophagy in the postsynaptic dendrites in mice. Dendritic autophagic vesicles (AVs) act in parallel with the endocytic machinery to remove AMPA receptor subunits from the membrane for degradation. During NMDAR-LTD, key postsynaptic proteins are sequestered for autophagic degradation, as revealed by quantitative proteomic profiling of purified AVs. Pharmacological inhibition of AV biogenesis, or conditional ablation of atg5 in pyramidal neurons abolishes LTD and triggers sustained potentiation in the hippocampus. These deficits in synaptic plasticity are recapitulated by knockdown of atg5 specifically in postsynaptic pyramidal neurons in the CA1 area. Conducive to the role of synaptic plasticity in behavioral flexibility, mice with autophagy deficiency in excitatory neurons exhibit altered response in reversal learning. Therefore, local assembly of the autophagic machinery in dendrites ensures the degradation of postsynaptic components and facilitates LTD expression.

[1] Department of Fundamental Neurosciences, University of Lausanne, Lausanne 1005, Switzerland. [2] University of Bordeaux, CNRS, Interdisciplinary Institute for Neuroscience, IINS, UMR 5297, F-33000 Bordeaux, France. [3] School of Biological Sciences, University of Crete, Heraklion 70013, Greece. [4] Proteomic Core Facility (PCF), European Molecular Biology Laboratory (EMBL), Heidelberg, Germany. [5] School of Medicine, University of Crete, Heraklion 71003, Greece. [6] Institute of Applied and Computational Mathematics (IACM), Foundation for Research and Technology—Hellas (FORTH), Heraklion, Greece. [7] Genome Biology Unit, European Molecular Biology Laboratory (EMBL), University of Rome Tor Vergata, Rome 00133, Italy. [8] Department of Biomedicine and Prevention, University of Rome Tor Vergata, Rome 00133, Italy. [9] University of Bordeaux, CNRS, INSERM, Bordeaux Imaging Center, BIC, UMS 3420, US 4, F-33000 Bordeaux, France. [10]These authors contributed equally: Emmanouela Kallergi, Akrivi-Dimitra Daskalaki. ✉email: vassiliki.nikoletopoulou@unil.ch

Macroautophagy (hereafter referred to as autophagy) is a highly conserved mechanism, whereby autophagic vesicles (AVs) deliver cellular constituents to the lysosome for degradation. Biogenesis of AVs is a multistep process that entails the nucleation of a U-shaped isolation membrane (also known as the phagophore), which sequesters cargo as it elongates and eventually closes to form the complete, double-membrane-bound AV, as previously reviewed[1,2].

Though initially considered as a process in bulk, it is increasingly appreciated that neuronal autophagy can also serve specific functions. In line with this notion, the contribution of autophagy to higher brain functions and to the underlying mechanisms of synaptic plasticity is beginning to be addressed[3–6]. For example, autophagy was recently shown to participate in memory formation[6–8] and erasure[9]. Moreover, our previous work indicated that neuronal autophagy is negatively regulated by a brain-derived neurotrophic factor (BDNF), a key regulator of synaptic plasticity, and that suppression of autophagy can rescue the synaptic defects associated with BDNF deficiency[10].

While in the presynaptic side autophagy has been shown to regulate neurotransmitter release possibly by degrading synaptic vesicles[11–14], as reviewed previously[4,15,16], or by controlling the axonal endoplasmic reticulum[17], its role in regulating postsynaptic processes is less understood. Recent findings suggest that autophagy can facilitate the degradation of AMPA[18] and GABA receptors[19]. In addition, it was shown to be cell-autonomously required in excitatory neurons for developmental spine pruning[20], a process suggested to be mediated by mechanisms similar to long-term depression (LTD) of synaptic strength[21].

LTD is a major form of long-lasting synaptic plasticity underlying key cognitive functions across a broad range of species[22–24]. This form of plasticity critically relies on the degradation of postsynaptic components and in some cases of postsynaptic structures, by mechanisms that remain poorly characterized. In this study, we address whether autophagy plays a role in this process by investigating the interplay and interdependence between LTD and this major cellular protein degradation pathway in pyramidal neurons. Our findings reveal that autophagy is indispensable for LTD.

## Results

**NMDAR- and mGluR-LTD trigger the rapid formation of autophagic structures in dendrites of cultured neurons.** In contrast to most cell types, where AV biogenesis occurs indiscriminately throughout the cytoplasm, in diverse neuronal populations, including hippocampal pyramidal neurons, biogenesis of AVs is spatially confined to the axon tip under baseline conditions[25,26]. Nascent phagophores mature to complete AVs as they are retrogradely transported via dynein motors from the axon tip towards the soma, where they fuse with lysosomes to deliver their cargo for degradation[27–31].

We examined whether AV distribution and biogenesis are altered by LTD-triggering neuronal activity. To this end, the two major types of LTD that co-exist in the brain, NMDAR- and mGluR-LTD, were chemically induced (cLTD) in cultured hippocampal neurons, as previously described[32–34], by a 5 min pulse of NMDA (50 μM) or DHPG (50 μM), respectively, and neurons were fixed 15 min after the pulse. LTD induction was tested by surface labeling for the AMPAR subunit GluA2 (GRIA2), confirming that both treatments significantly reduced the surface expression of this AMPAR subunit in dendrites 15 min after the pulse (Fig. 1a). Control and LTD neurons were immunolabeled with an antibody against LC3B (hereafter referred to as LC3), a protein that appears punctate when associated with autophagic membranes. In control neurons, the density of LC3-positive puncta in dendrites under baseline conditions decreases with increasing distance from the soma, consistent with previous findings indicating that autophagic structures are scarce in dendrites[26]. However, both forms of LTD triggered the rapid and significant increase of LC3-positive puncta in dendrites 15 min after the pulse (Fig. 1b). This increase was persistent, as it could also be observed 30 and 60 min after the NMDA and DHPG pulse (Supplementary Fig. 1a). By contrast, in the soma area, LC3 intensity remained similar between control and NMDAR-LTD conditions, and a small but significant decrease was observed in the mGluR-LTD conditions, compared to control, 15 min after the pulse (Supplementary Fig. 1b).

To ensure that the LC3-positive puncta represent autophagic structures, we used two inhibitors to target two distinct kinases that are required for the early steps of the AV biogenesis: First, Wortmannin, a broad inhibitor of PI3Ks which also inhibits Vps34 (a catalytic subunit of the PI3K complex), and has been widely used in the field to block AV biogenesis[35]; Second, SBI-0206965, a selective inhibitor of ULK1 kinase activity, which blocks the activation of the ULK1 complex[10,36,37]. Pretreatment with either Wortmannin or SBI-0206965 was effective in decreasing the levels of LC3-II (Supplementary Fig. 1c) compared to untreated neurons. Moreover, both pre-treatments significantly decreased the number of dendritic LC3-positive puncta 15 min after an NMDA or DHPG pulse, while there was no significant effect on the number of dendritic LC3-positive puncta in control neurons (Fig. 1c).

To further determine if the LC3-positive structures are Atg5-dependent, we infected neurons with a viral vector carrying 4 shRNA constructs against *atg5 (sh-atg5)*, or with a control vector carrying 4 scrambled sequences (*sh-scramble*) under the *CamK2a* promoter. The effective silencing of *atg5* was confirmed by western blot analysis with an antibody against Atg5, which recognizes its complex with Atg12. Neurons expressing *sh-atg5* exhibited significantly lower levels of Atg5-Atg12, as compared to those expressing *sh-scramble* or to uninfected controls (Supplementary Fig. 1d). The achieved *atg5* silencing was sufficient to reduce the levels of LC3-II approximately by half (Supplementary Fig. 1e). Notably, the number of dendritic LC3-positive puncta was significantly increased 15 min after an NMDA or DHPG pulse, in *sh-scramble* expressing neurons, but failed to increase in *sh-atg5* expressing neurons (Fig. 1d). No significant differences were observed in the number of dendritic LC3-positive puncta between *sh-scramble* and *sh-atg5* expressing neurons under control conditions. Taken together, these findings indicate that the rapid appearance of dendritic LC3-positive puncta is ULK1-, Vps34-, and Atg5-dependent.

Notably, pretreatment with ifenprodil, a selective inhibitor of NR2B receptors, known to have an important role in LTD[38] or with the mGluR inhibitors MTEP and JNJ16259685 (JNJ)[39,40], prevented the appearance of dendritic LC3-positive puncta after LTD-inducing stimuli (Fig. 1e). Therefore, the rapid increase in dendritic autophagic structures is specifically recruited by activation of NMDA and mGluR receptors.

To determine whether the surge of dendritic autophagic structures upon LTD is associated with an increase in autophagic flux, neurons were treated with Bafilomycin A1, an inhibitor of lysosomal acidification and hence of the autophagic flux, during and for 1 h after the LTD-inducing treatments. Western blot analysis for LC3 indicated increased autophagic flux, as determined by the difference in the levels of lipidated LC3 (LC3-II) (Supplementary Fig. 1f). This was further supported by a small yet significant decrease in the protein levels of the autophagic substrate p62 180 min, but not 15 min, after the LTD-inducing pulses (Supplementary Fig. 1g).

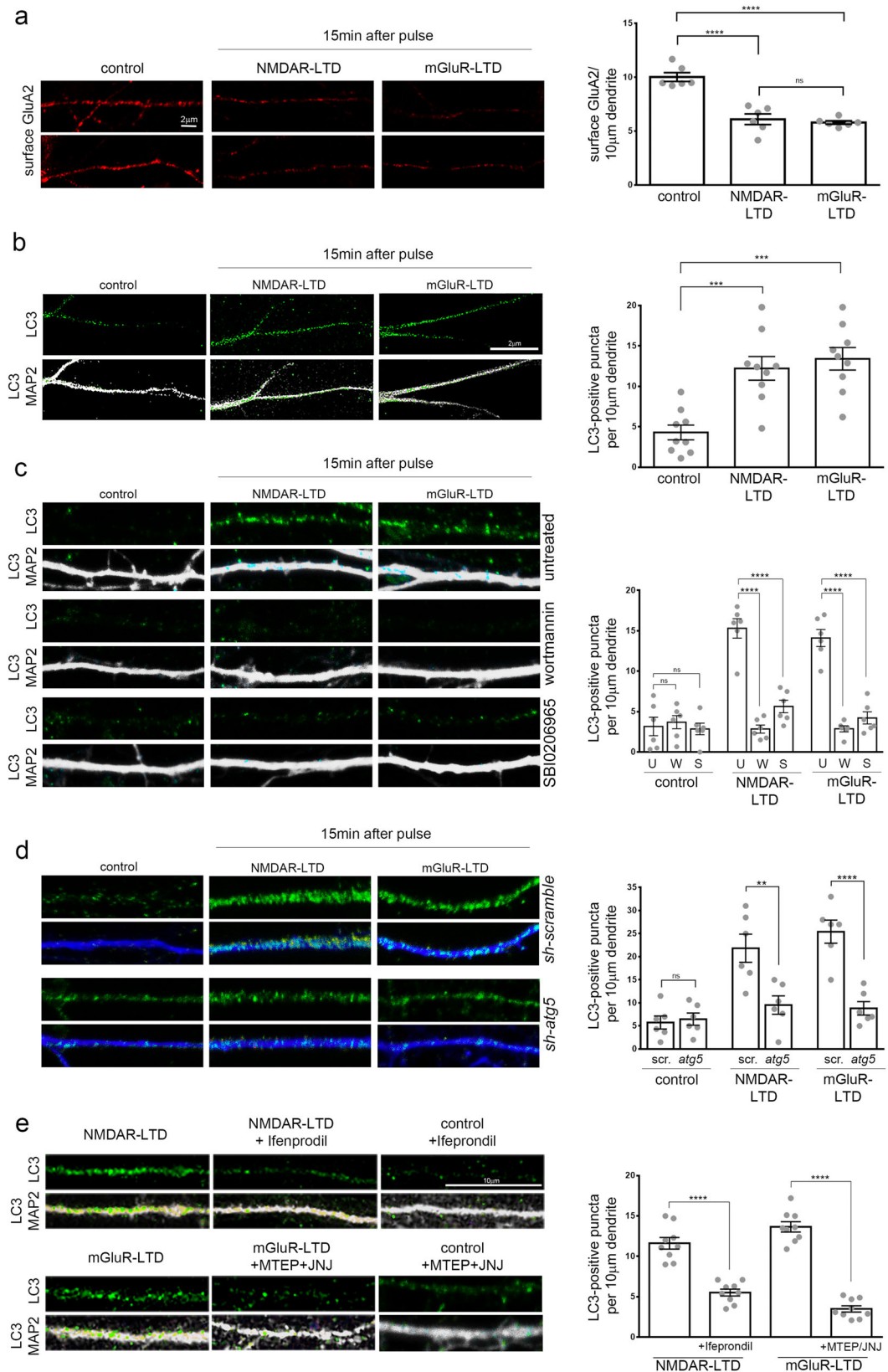

**Autophagic vesicles are locally generated in postsynaptic dendrites upon LTD.** Lipidated LC3 associates with autophagic membranes during phagophore biogenesis and is maintained in the mature autophagic vesicles, both in their inner and outer membranes[41]. The rapid emergence of dendritic LC3-positive puncta within 15 min after LTD, invites the speculation that

autophagic vesicles may be locally biosynthesized under this condition. To test this hypothesis, we first analyzed the ultrastructure of the dendritic LC3 puncta after chemical NMDAR- and mGluR-LTD in cultured neurons, using super-resolution microscopy (Fig. 2a). Notably, 15 min after chemical LTD there was a significant ~2.5-fold increase in the number of dendritic

**Fig. 1 NMDAR- and mGluR-LTD trigger the rapid appearance of autophagic structures in dendrites of cultured neurons. a** Representative confocal images of cultured neurons under control conditions or 15 min after chemical NMDAR- or mGluR-LTD, immunolabeled for surface GluA2 (under non-permeabilizing conditions). Graph showing the number of surface GluA2 labeling, normalized to the dendritic length, in the indicated conditions. Bars represent mean values ± SEM. $N = 6$ independent experiments per condition. Statistical analyses were performed by one-way ANOVA, F (2, 15) = 38.28) (Tukey's test $P_{control-NMDAR} < 0.0001$, $P_{control-mGluR} < 0.0001$, $P_{NMDAR-mGluR} = 0.8438$). **b** Representative confocal images of cultured neurons under control conditions or 15 min after chemical NMDAR- or mGluR-LTD, immunolabeled with an antibody against endogenous LC3 (autophagic structures) and MAP2 (dendrites). Graph showing the number of dendritic LC3-positive puncta in secondary dendrites, normalized to the dendritic length, in the indicated conditions. Bars represent mean values ± SEM. $N = 9$ independent experiments per condition. Statistical analyses were performed by one-way ANOVA (F2,24) = 15.11, $P < 0.0001$) (Tukey's test $P_{control-NMDA} = 0.0005$, $P_{control-mGluR} = 0.0001$). **c** Same as in **b**, but neurons were pretreated for 1 h before, during and after the pulse with wortmannin (500 nM) or SBI-0206965 (500 nM). Graph showing the number of dendritic LC3-positive puncta, normalized to the dendritic length, in the indicated conditions (U: untreated, W: wortmannin, S: SBI-0206965). Bars represent mean values ± SEM. $N = 6$ independent experiments per condition. Statistical analyses were performed by one-way ANOVA (F(8,45) = 33.83, $P < 0.0001$) (Tukey's test $P_{control/S-NMDA/S} = 0.3677$, $P_{control/W-NMDA/W} = 0.9986$, $P_{NMDA/U-NMDA/W} < 0.0001$, $P_{NMDA/U-NMDA/S} < 0.0001$, $P_{control/S-DHPG/S} = 0.9674$, $P_{control/W-DHPG/W} = 0.9989$, $P_{DHPG/U-DHPG/W} < 0.0001$, $P_{DHPG/U-DHPG/S} < 0.0001$). **d** Same as in **b** with neurons that were infected with AAV plasmids carrying 4 shRNA sequences against *atg5* (sh-atg5) or *scrambled* control (sh-scramble), under the CamK2a promoter. Graph showing the number of dendritic LC3-positive puncta, normalized to the dendritic length, in the indicated conditions. Bars represent mean values ± SEM. $N = 6$ independent inhibitor experiments per condition. Statistical analyses were performed by one-way ANOVA (F(5,30) = 16.94, $P < 0.0001$) (Tukey's test $P_{control/scr-control/atg5} = 0.9999$, $P_{NMDA/scr-NMDA/atg5} = 0.0025$, $P_{DHPG/scr-DHPG/atg5} < 0.0001$, $P_{control/scr-NMDA/scr} < 0.0001$, $P_{control/scr-DGPG/scr} < 0.0001$, $P_{control/atg5-NMDA/atg5} = 0.8959$, $P_{control/atg5-DHPG/atg5} = 0.9637$). **e** Same as in **b**, but neurons were immunolabeled 15 min after NMDAR- and mGluR-LTD and treated for 1 h before, during and after the pulse with Ifenprodil (10 μM) or MTEP (10 μM) and JNJ16259685 (10 μM) to pharmacologically inhibit NR2B and mGluR1/5 receptors, respectively. Graph showing the number of dendritic LC3-positive puncta, normalized to the dendritic length, in the indicated conditions. $N = 9$ independent experiments per condition. Statistical analyses were performed by one-way ANOVA (F (3,32) = 74.46, $P < 0.0001$) (Tukey's test, $P_{NMDA-NMDA+IFE} < 0.0001$, $P_{DHPG-DHPG+MTEP/JNJ} < 0.0001$). Scale bars: 10 μm for all panels.

LC3-positive U-shaped structures (Fig. 2b, left graph), indicative of phagophores[42]. This increase was also sustained at 60 min after chemical NMDAR- and mGluR-LTD (Fig. 2b, right graph) These structures ranged from 150 to 500 nm in size (Fig. 2c), with a mean size of approximately 300 nm both in NMDAR- and mGluR-LTD dendrites. The amount of LC3 which was not associated with a phagophore structure was unchanged (Supplementary Fig. 2a). LC3-positive structures did not colocalize with the early endosome marker EEA1 (Supplementary Fig. 2b), further indicating that it is not conjugated on single membrane endosomes. Furthermore, 15 min after LTD, dendritic LC3 puncta colocalized with WIPI2 (Fig. 2d), a protein responsible for Atg12–5-16L1 recruitment to PtdIns(3)P-positive donor membranes and LC3 lipidation[43], further supporting that the dendritic LC3 structures represent nascent autophagic membrane structures.

We analyzed additional markers associated with the initiation of AV biogenesis. One of the most upstream events in the biogenesis cascade is the formation and activation of the ULK1 complex, which in mammalian cells is comprised of ULK1, Atg101, Atg13, and FIP200[41]. We compared the distribution of the ULK1 complex components in dendrites of cultured neurons before and 15 min after chemical LTD, by immuno-labeling with antibodies against each of the four components and for MAP2. Under control conditions, all ULK1-complex components were present in dendrites at very low levels and with diffuse appearance. However, 15 min after NMDAR- or mGluR-LTD, the levels of all four components increased sharply in dendrites and exhibited a punctate pattern (Fig. 2e). Significantly increased levels of all ULK1 complex components as well as of the lipidated LC3-II were also measured by western blot analysis in neuronal lysates 15 min after LTD, as compared to control (Fig. 2f). Notably, pretreatment with the NR2B inhibitor ifenprodil and the mGluR inhibitors MTEP and JNJ for 1 h before and during the pulse and 15 min after, prevented the appearance of Atg13-positive puncta (Supplementary Fig. 2c, d) as well as of ULK1, Atg101, and FIP200 (Supplementary Fig. 2d), in dendrites after NMDAR- and mGluR-LTD, respectively. Therefore, similar to LC3 (Fig. 1e), the immediate increase in ULK1 complex component-positive structures is specifically induced by activation of NMDA and mGluR receptors.

The ability of LTD to trigger dendritic autophagy was next tested in hippocampal slices. NMDAR- or mGluR-LTD was induced in wild-type hippocampal slices by a 5 min pulse of NMDA (20 μM) or DHPG (50 μM), respectively, as previously described[44]. Slices were fixed 15 min after the pulse and were immunolabeled with antibodies against each ULK1 complex component and MAP2. As a positive control, slices were immunolabeled with an antibody against Arc, an early response protein, whose levels are rapidly increased in dendrites in response to LTD[45] (Fig. 3a). Focusing on the *stratum radiatum* of the CA1 area (Fig. 3a, schematic), we found that similar to Arc, the levels of all four ULK1 complex components sharply increased in these dendrites 15 min after NMDAR- and mGluR-LTD (Fig. 3b–e). Moreover, as in cultured neurons, all four exhibited punctate patterns. Quantification of the number of puncta of each ULK1-complex component in CA1 *stratum radiatum* dendrites revealed that they are all significantly increased 15 min after NMDAR- or mGluR-LTD, compared to control conditions (graphs in Fig. 3b–e).

In order to directly visualize the autophagic structures in postsynaptic dendrites in the hippocampus, LTD was induced by low-frequency stimulation (LFS) of Shaffer collateral fibers and neurobiotin was concomitantly delivered by the recording electrode, allowing the labeling of postsynaptic dendrites in the recording area of the CA1 *stratum radiatum*. Neurobiotin-labeling, which was visualized with avidin–biotin–peroxidase complex (ABC) - DAB method, was restricted to the recording area of the CA1 (Supplementary Fig. 3a). Moreover, neurobiotin labelling was specific, as in slices where LFS protocol was delivered in the absence of neurobiotin, no staining was observed after the DAB reaction (Supplementary Fig. 3a). Control (delivery of neurobiotin in the absence of LFS) or LTD-induced slices were fixed 15 min or 1 h later (following LFS in the case of LTD) and processed for electron microscopy, as previously described[46]. AVs were identified in labeled distal postsynaptic dendrites of pyramidal neurons in the *stratum radiatum*, as double-membrane vesicles and phagophores as U-shaped structures (Fig. 3f), which could be reconstituted in consecutive sections. No difference in the density of dendritic AVs was found between labeled and adjacent unlabeled dendrites in control slices (Supplementary Fig. 3b), indicating that neurobiotin does not

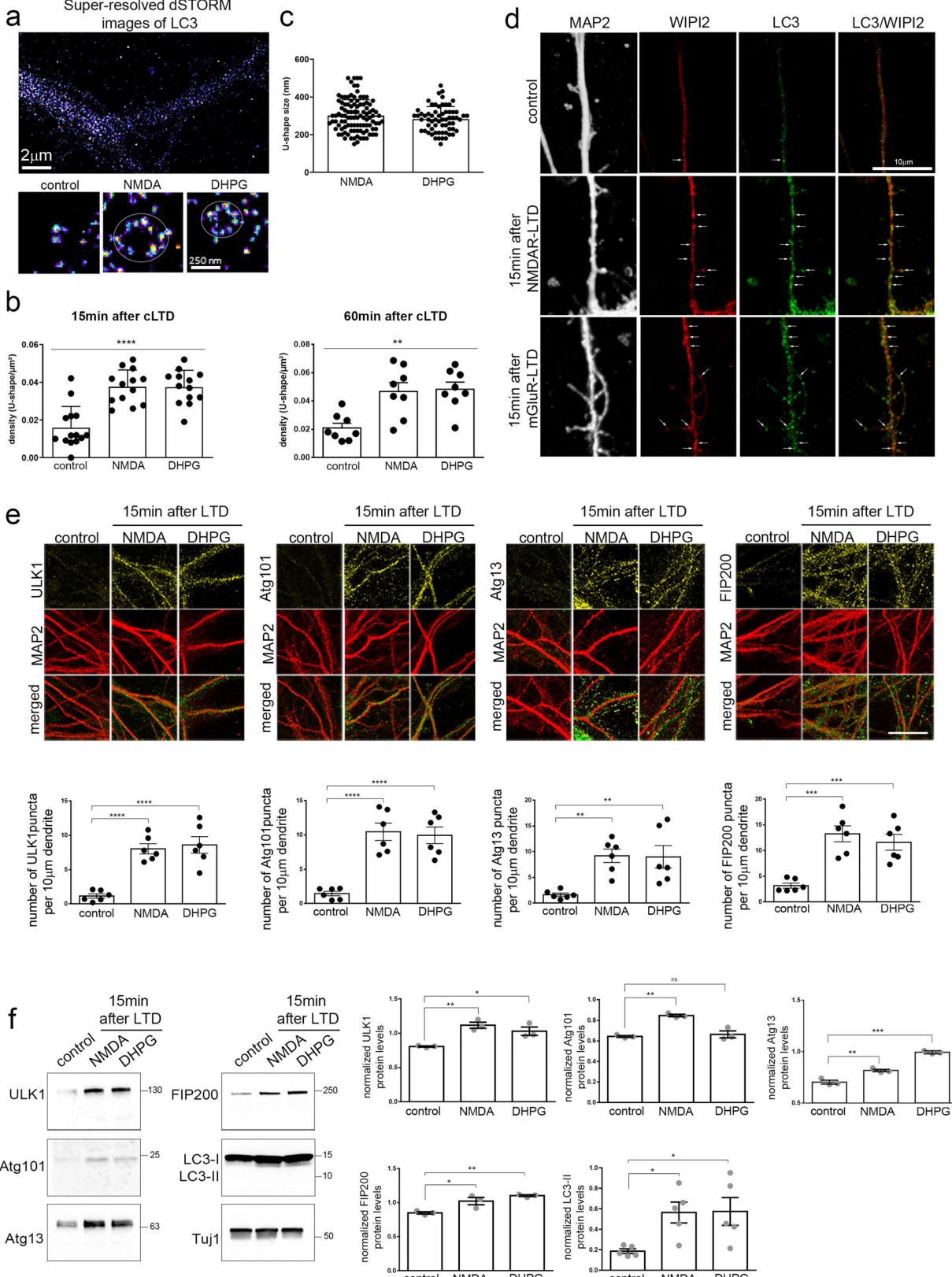

by itself alter the AV content of dendrites. Moreover, neurobiotin did not interfere with LTD upon LFS stimulation (Supplementary Fig. 3c).

Consistent with previous reports[25,26], labeled distal dendrites in control slices contained almost no phagophores and very few AVs, most often none or one vesicle per square μm surface

(Fig. 3f–h). However, 15 min after LTD, we observed a significant increase in the percentage of labeled dendrites that contained U-shaped structures (red arrow) or U-shaped structures more advanced towards closure (yellow asterisks) (Fig. 3f, g). In all cases, the structures were traced in consecutive sections to confirm their morphology. Bath application of Ciliobrevin-D, a

**Fig. 2 Autophagic vesicles are locally formed in dendrites of cultured neurons following LTD. a** Top, representative super-resolution microscopy dSTORM image of a secondary dendrite labeled with an antibody against LC3, 15 min after cLTD. Bottom, magnification of representative U-shaped LC3-positive structures in dendrites, 15 min after NMDA or DHPG pulses. Scale bars: 2 μm and 250 nm, as indicated. ($N = 3$ independent experiments). **b** Graph showing the number of LC3-positive U-shaped structures in secondary dendrites visualized in (a), before (control) and 15 min or 60 min after NMDAR- and mGluR-LTD. Bars represent mean values ± SEM. $N = 3$ independent experiments per condition ($n > 9$ dendrites per condition). Statistical analysis was performed by one-way ANOVA. For the time point of 15 min $F(2, 33) = 17.93$, $p < 0.0001$) (Tukey's test $P_{control-NMDA} < 0.0001$, $P_{control-DHPG} < 0.0001$). For the time point of 60 min $F(2, 21) = 9.459$, $p = 0.0012$) (Tukey's test $P_{control-NMDA} = 0.0041$, $P_{control-DHPG} = 0.0024$). **c** Graph showing the size distribution (nm) of the dendritic U-shaped LC3-positive structures visualized in **a** upon NMDAR- and mGluR-LTD. Bars represent mean values ± SEM for each analysed dendrite. $N = 3$ independent experiments ($n > 40$ dendrites per condition). **d**, Confocal images of dendrites immunolabeled with antibodies against WIPI2, LC3, and MAP2 before (control) or after 15 min of NMDAR- and mGluR-LTD. Scale bar: 10 μm. ($N = 6$ independent experiments). **e**, Representative confocal images of neurons immunolabeled with antibodies against ULK1, Atg101, Atg13, FIP200 and, along with MAP2 to label dendrites before (control) or 15 min after LTD-inducing pulses. Scale bar: 20 μm. Graphs showing the number of puncta positive for each ULK1-complex component in secondary dendrites, normalized for dendrite length, in every condition, as indicated. Graph bars represent mean values ± SEM. $N = 6$ independent experiments per condition. Statistical analyses were performed using one-way ANOVA. ULK1: $F(2,15) = 24.48$, $P < 0.0001$ (Tukey's multiple comparison test, $P_{control\_NMDAR} < 0.0001$, $P_{control\_mGluR} < 0.0001$, $P_{NMDAR\_mGluR} = 0.8825$). Atg101: $F(2,15) = 24.31$, $P < 0.0001$ (Tukey's multiple comparison test, $P_{control\_NMDAR} < 0.0001$, $P_{control\_mGluR} < 0.0001$, $P_{NMDAR\_mGluR} = 0.9329$). Atg13: $F(2,15) = 8.386$, $P = 0.0036$ (Tukey's multiple comparison test, $P_{control\_NMDAR} = 0.007$, $P_{control\_mGluR} = 0.0086$, $P_{NMDAR\_mGluR} = 0.9940$). FIP200: $F(2,15) = 17.66$, $P = 0.0001$ (Tukey's multiple comparison test, $P_{control\_NMDAR} = 0.0002$, $P_{control\_mGluR} = 0.0009$, $P_{NMDAR\_mGluR} = 0.6440$). **f** Western blot analyses for Atg13, FIP200, ULK1, Atg101, LC3, and β-III tubulin (Tuj1) in neuronal lysates, under control conditions and 15 min after NMDAR- and mGluR-LTD. Graphs showing the normalized protein levels of Atg13, FIP200, ULK1, Atg101 under the aforementioned conditions. Bars represent mean values ± SEM. $N = 3$ independent experiments for ULK1 complex proteins, $N = 5$ independent experiments for LC3-II. Statistical analyses were performed using one-way ANOVA. ULK1: $F(2,6) = 13.57$, $P = 0.0059$ (Tukey's multiple comparison test $P_{control\_NMDA} = 0.0056$, $P_{control\_DHPG} = 0.0253$). Atg101: $F(2,6) = 27.57$, $P = 0.0009$ (Tukey's multiple comparison test $P_{control\_NMDA} = 0.0013$, $P_{control\_DHPG} = 0.79$). Atg13: $F(2,6) = 113.1$, $P < 0.0001$ (Tukey's multiple comparison test $P_{control\_NMDA} = 0.0027$, $P_{control\_DHPG} < 0.0001$). FIP200: $F(2,6) = 15.14$, $P = 0.0045$ (Tukey's multiple comparison test $P_{control\_NMDA} = 0.0240$, $P_{control\_DHPG} = 0.041$). LC3-II: $F(2, 12) = 4,969$, $P = 0.0268$ (Tukey's multiple comparison test $P_{control\_NMDA} = 0.0478$, $P_{control\_DHPG} = 0.0421$).

specific inhibitor of dynein-mediated transport, 2 h prior to the LFS protocol and during the recording did not prevent the emergence of these phagophore-shaped structures in dendrites (Fig. 2g), supporting the notion that they are locally produced.

Similarly, we observed a significant increase in the density of complete, double-membrane AVs of approximately 2.5-fold compared to control. These vesicles were accumulated in the dendritic shaft 1 h after LTD induction, often within 1 μm from the base of dendritic spines (Fig. 3f, h). Of note, the number of AVs in buttons synapsing onto neurobiotin-labeled dendrites was unchanged between control and LTD conditions (Supplementary Fig. 3d). Bath application of Ciliobrevin-D 2 h prior, during and after the LFS stimulation did not prevent the increase in dendritic AV density 1 h after LFS (Fig. 3h). Instead, as with phagophores, Ciliobrevin-D further potentiated the increase in dendritic AV density 1 h after LFS, indicating that AVs may be trafficked away from dendrites and towards the soma via dynein.

Most double membrane, complete autophagic vesicles ranged from 200 to 700 nm in diameter, which was measured in the middle section of each structure (Fig. 3f, i). Ciliobrevin-D had no effect on the size of dendritic AVs (Fig. 3i). It is worth noting that often, a small vesicle that appeared incompletely sealed and ~100 nm in diameter was found in spine heads, as shown in Fig. 3f (black arrows), with a morphology clearly distinct from that of the spine apparatus.

Taken together, these findings indicate that LTD triggers the biogenesis of dendritic autophagic vesicles, both in cultured neurons and in acute hippocampal slices.

**Autophagy degrades AMPAR subunits and postsynaptic proteins during LTD.** It is well-established that clathrin-mediated endocytosis (CME) has a crucial role in the early steps of LTD, as it acts rapidly to remove AMPA receptors from the postsynaptic membrane[47–49]. Consistently, in cultured neurons, inhibition of dynamin, a GTPase driving membrane fission and regulating the rate-limiting steps of CME, with a selective inhibitory peptide for 25 min before, during and 15 min after chemical NMDAR- or mGluR-LTD, prevented the rapid decrease of surface GluA2

levels, but did not affect GluA2 surface levels under control conditions (Fig. 4a).

Next, we tested the effect of acute inhibition of autophagic vesicle biogenesis on AMPAR surface levels. Cultured neurons were pretreated with SBI-0206965 (Fig. 4a) or with Wortmannin (Supplementary Fig. 4a) for 25 min before, during and 15 min after chemical NMDAR- or mGluR-LTD. Notably, blockade of the autophagy initiation machinery completely prevented GluA2 internalization following both forms of LTD (Fig. 4a and S4a), as efficiently as inhibition of the endocytic machinery.

To further investigate whether these results reflect a role of autophagy in AMPAR internalization, we assessed the surface levels of GluA2 upon chemical LTD, in neurons infected with a viral vector carrying 4 shRNA constructs against *atg5 (sh-atg5)*, or with a control vector carrying 4 scrambled sequences (*sh-scramble*) under the *CamK2a* promoter (as in Fig. 1d). Surface levels of GluA2 rapidly decreased 15 min after NMDAR- and mGluR-LTD compared to untreated control conditions in *sh-scramble* expressing neurons, but they failed to decrease in *sh-atg5* expressing neurons (Fig. 4b). However, the surface levels of GluA2 were similar between *sh-scrambled* and *sh-atg5* expressing neurons under control conditions. Therefore, the internalization of GluA2 during LTD requires the activation of the ULK1 complex as well as Atg5.

Based on these findings, we hypothesized that the autophagic machinery may sequester scaffold proteins whose removal from the postsynaptic density and degradation are essential prerequisites for the endocytosis of AMPA receptor subunits. To test this hypothesis, we performed co-labeling experiments in cultured neurons for LC3 and PSD95, a scaffold protein that anchors AMPARs on the postsynaptic membrane and regulates synaptic strength[50–52]. We found that 15 min after NMDAR- or mGluR-LTD, PSD95 colocalized with LC3-positive puncta in consecutive z-planes, indicating that it may be indeed sequestered by phagophores (Fig. 4c). The percentage of PSD95 that colocalizes with LC3 is increased 15 min after NMDAR- or mGluR-LTD, as compared to control, and this increase is more pronounced in dendritic spines and less in dendritic shafts (Fig. 4c).

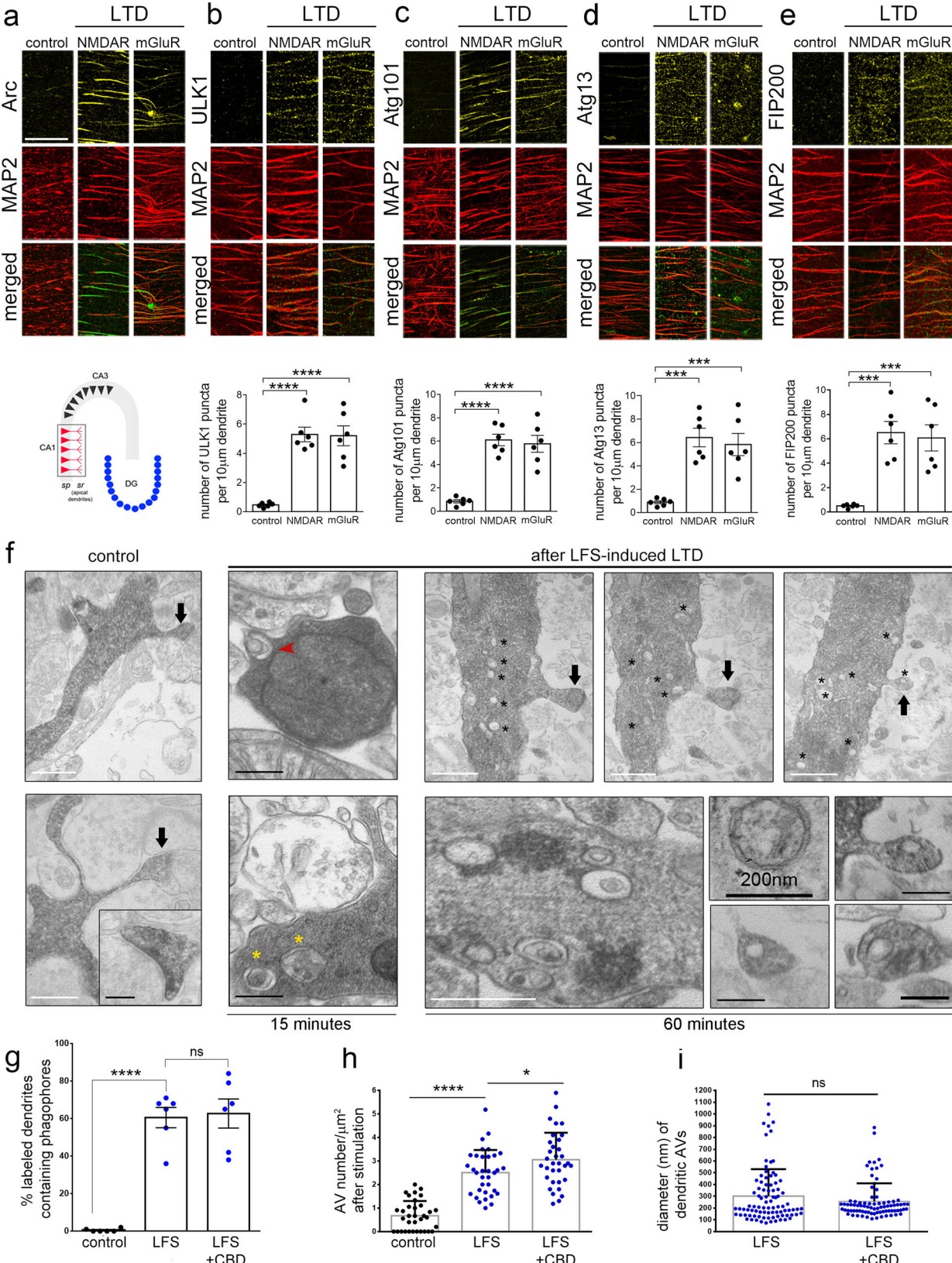

Furthermore, to test whether PSD95 and AMPARs undergo lysosomal degradation during the early phase of LTD, we compared their levels in neuronal lysates in control conditions or 15 min after NMDAR- or mGluR-LTD, in the presence or absence of Bafilomycin A1 (Fig. 4d). We found that both PSD95 (Fig. 4d) and GluA2 (Fig. 4d) levels decrease rapidly 15 min after

LTD, as compared to control conditions. However, when lysosomal acidification and autophagic flux are blocked, this decrease is fully prevented, indicating that they are degraded by the lysosome during LTD.

To further confirm that their lysosomal degradation is mediated by macroautophagy, we tested the levels of GluA2

**Fig. 3 LTD triggers dendritic autophagy in hippocampal slices. a–e** Representative confocal images of the CA1 *stratum radiatum* area of wild-type P22-P28 hippocampal slices (as shown in the schematic representation), immunolabeled with antibodies against **a** Arc, **b** ULK1, **c** Atg101, **d** Atg13, and **e** FIP200 along with MAP2. Immuno-labeling was performed in slices under control conditions or 15 min after chemical NMDAR- or mGluR-LTD. Scale bars: 50 µm. Graphs showing the number of puncta positive for each ULK1-complex component, normalized to MAP2. Bars represent mean values ± SEM. $N = 6$ independent experiments per condition. Statistical analyses were performed using one-way ANOVA. ULK1: $F_{(2,15)} = 32.10$, $P < 0.0001$ (Tukey's multiple comparison test $P_{control\_NMDAR} < 0.0001$, $P_{control\_mGluR} < 0.0001$, $P_{NMDAR\_mGluR} = 0.9906$). Atg101: $F_{(2,15)} = 32.92$, $P < 0.0001$ ($P_{control\_NMDAR} < 0.0001$, $P_{control\_mGluR} < 0.0001$, $P_{NMDAR\_mGluR} = 0.8920$). Atg13: $F_{(2,15)} = 17.88$, $P = 0.0001$ ($P_{control\_NMDAR} = 0.0002$, $P_{control\_mGluR} = 0.0006$, $P_{NMDAR\_mGluR} = 0.8267$). FIP200: $F_{(2,15)} = 16.91$, $P = 0.0001$ ($P_{control\_NMDAR} = 0.0003$, $P_{control\_mGluR} = 0.0006$, $P_{NMDAR\_mGluR} = 0.9247$). **f** Representative electron micrographs of neurobiotin-labeled dendrites of pyramidal neurons in the CA1 *stratum radiatum* area in hippocampal slices under control conditions and 15 min or 1 h after LTD induction by LFS. Black arrows indicate the labeled dendritic spines. Note the presence of multiple autophagic vesicles in the dendritic shaft near the neck of the spines (asterisks). Bottom panels show characteristic double-membrane-bound autophagic vesicles at higher magnification and the presence of vesicle-like structures inside dendritic spines. Scale bars: 500 nm (white bars), 200 nm (black bars). ($N = 6$ animals per condition). **g** Graph showing the percentage of labeled postsynaptic dendrites containing phagophores in control condition or 15 min after LFS and in the presence or absence of Ciliobrevin-D ($25\,\mu$M), bath applied for 2 h prior, during and after LFS. Bars represent mean values ± SEM. $N = 6$ animals per condition, $n = 48$ dendrites per condition. Statistical analyses were performed by one-way ANOVA, $F_{(2,15)} = 41.99$, $P < 0.0001$ (Tukey's multiple comparison test $P_{control\_LFS} < 0.0001$, $P_{LFS\_LFS/CBD} = 0.9574$, $P_{control\_LFS/CBD} < 0.0001$). **h** Graph showing the density of autophagic vesicles in postsynaptic labeled dendrites under control conditions and 1 h after LFS-induced LTD with or without bath application of Ciliobrevin-D. Graph bars represent mean values ± SEM. $n = 34$ dendrites per condition from $N = 6$ animals. Statistical analyses were performed using Mann–Whitney U test, comparing the LFS condition with or without bath application of Ciliobrevin-D to control ($P_{control\_LTD} < 0.0001$; $P_{control\_LTD/CBD} < 0.0001$; $P_{LTD\_LTD/CBD} = 0.0209$). **i** Graph showing the size distribution of autophagic vesicles in postsynaptic labeled dendrites 1 h after LFS-induced LTD with or without bath application of Ciliobrevin-D. Bars represent mean values ± SEM. $N = 89$ autophagic vesicles. Statistical analyses were performed using Mann–Whitney U test ($P = 0.2648$).

and PSD95 in control conditions and 15 min after chemical LTD, in neurons that were pretreated as before with SBI-0206965, or not, (Fig. 4e) as well as in neurons with a knockdown of *atg5* (Fig. 4f). Both the inhibition of autophagy initiation by SBI-0206965 treatment and the knockdown of *atg5* completely prevented the decrease in PSD95 and GluA2 protein levels 15 min after NMDAR- and mGluR-LTD. No significant differences were observed in the protein levels of PSD95 and GluA2 under control conditions after SBI-0206965 treatment or *atg5* knockdown (Fig. 4e, f), suggesting that autophagy does not regulate their baseline turnover, at least for the time frame examined in these experiments. Taken together, these findings indicate that the proper and rapid degradation of GluA2 and its scaffold PSD95 require the autophagy-lysosome system.

It is worth noting that inhibition of CME did not prevent the increase in dendritic LC3-positive structures in the case of NMDAR-LTD but significantly decreased their increase in the case of mGluR-LTD (Supplementary Fig. 4b).

**Proteomic profiling of the autophagic cargo upon NMDAR-LTD.** Although autophagy is a major cellular mechanism for protein degradation, its synaptic cargo remains elusive. Pertinent to LTD, in addition to the internalization and degradation of AMPARs, NMDAR-LTD, in particular, may also entail the shrinkage or elimination of entire spines. Therefore, we postulated that autophagy can facilitate the degradation of additional postsynaptic components and we sought to unravel the identity of the cargo sequestered in autophagic vesicles during LTD.

To this end, AVs were purified from hippocampal slices of P22-28 mice, before and 1 h after NMDAR-LTD using a method that we recently established[10]. Western blot analysis of the purified AVs, and of other fractions obtained along with the purification procedure, indicated that the final AV preparation is enriched in LC3-II and also contains p62, but is negative for the isolation membrane marker Atg9a, and the phagophore marker Atg16L1 (Fig. 5a), indicating that incomplete autophagic membranes are not purified in the final fraction. In line with this, electron microscopy analysis indicated that the purified AV pellet contains closed double-membrane AVs which were traced in consecutive sections and contained either amorphous material (Supplementary Fig. 5a, first two columns), membranous material

(Supplementary Fig. 5a, third column), or what appeared as inclusions of damaged mitochondria (Supplementary Fig. 5a, last two columns). The integrity of purified AVs was further tested by treating them with Proteinase-K with or without Triton X-100. As shown in Supplementary Fig. 5b, when purified AVs were permeabilized by triton X-100, both LC3 and p62 were digested by the protease. However, in the absence of permeabilization, p62, a known autophagic cargo, is fully protected in the presence of Proteinase-K, indicating that vesicles are normally intact and impermeable to the protease. Consistent with LC3-II being present on both the inner and outer autophagic membranes, Proteinase-K treatment reduced the levels of LC3-II approximately by half.

We further scrutinized the purity of the AV fraction, by testing the expression of TGN, LMAN1 (ERGIC53), and SAR1a, three proteins involved in ER to Golgi transport, and found that they were mainly found in the cytosolic and ER-containing fractions but absent from the final AV fraction (Fig. 5b). Similarly, Rab11 and EEA1, which mark recycling endosomes and early endosomes, respectively, were also not detected in the final AV fraction (Fig. 5c). Similarly, the AV fraction was negative for the plasma-membrane localized protein syntaxin-4 (STX4), the extracellular vesicle marker Alix and the nuclear marker TBP (TATA-binding protein) (Fig. 5d). Therefore, both the ultrastructural and the biochemical analyses we performed indicated that the purified AV fraction contains mature double-membrane and LC3-positive autophagic vesicles, allowing us to use them for proteomic analyses.

Control- and LTD-AVs were subjected to TMT labeling and quantitative proteomic analyses. We found a total of 2359 proteins, of which 69% were also found in a previous characterization of the autophagic cargo in cultured cell lines[53], whereas the remaining 31% were brain-specific (Supplementary Fig. 5c and Supplementary Data 1). As shown in Supplementary Fig. 5c, the cargo that was common between our analysis and the one in cell lines was mainly enriched for a protein involved in housekeeping functions such as mitochondrial proteins involved in the respiratory chain and ribosomal subunits, among others. By contrast, the unique, brain-specific cargo was enriched for proteins involved in synaptic functions, where both pre- and postsynaptic structures were represented.

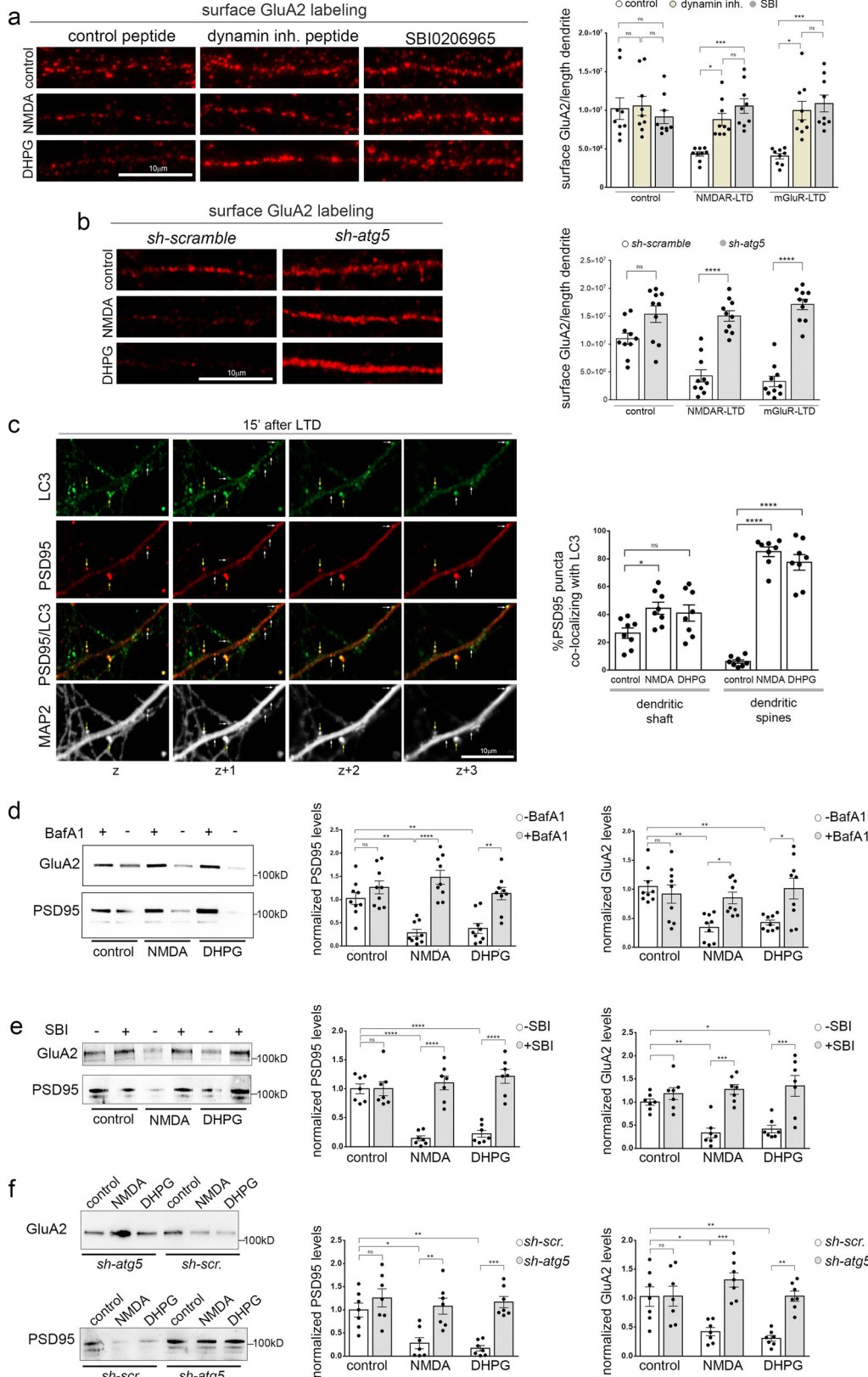

Next, we compared the cargo obtained under control and LTD conditions. Of the total 2359 proteins, the vast majority (1966 proteins, 83.3%) were equally abundant between control and LTD conditions and therefore constitute a stable or constitutive cargo of autophagy (Supplementary Fig. 5d). The remaining 393 proteins, summarized in Supplementary Data 2, showed a dynamic behavior between control and LTD conditions and were annotated as hits (38 proteins), candidates (94 proteins), and trends (261 proteins), according to their fdr value (<0.05, <0.2, and <0.6, respectively) (Supplementary Fig. 5d). Approximately half (48.6%) of the dynamic cargo was preferentially enriched in LTD-AVs, while the other half (51.4%) was preferentially excluded from LTD-AVs.

**Fig. 4 Autophagy degrades key postsynaptic proteins during LTD. a** Confocal images of dendrites immunolabeled with an antibody against the extracellular region of GluA2 under control conditions or 15 min after LTD induction and in the absence or presence of Dynamin-1 inhibitory peptide (50 μM) or SBI-0206965 (500 nM), a selective inhibitor of the ULK1 kinase activity. Inhibitors were applied 25 min before, during and 15 min after the pulses. Scale bar: 10 μm. Graph showing the surface labeling of GluA2, normalized to dendritic length under the aforementioned conditions. Bars represent mean values ± SEM. $N = 9$ independent experiments. Statistical analysis was performed using one-way ANOVA (F (8, 72) = 7.411, $P < 0.0001$) (Tukey's test $P_{control-control/D} > 0.99$, $P_{control-control/S} = 0.9971$, $P_{NMDA-NMDA/D} = 0.0451$, $P_{NMDA-NMDA/S} = 0.0008$, $P_{DHPG-DHPG/D} = 0.0017$, $P_{DHPG-DHPG/S} = 0.0002$). **b** Confocal images of dendrites of neurons expressing 4 scrambled sequences (*sh-scramble*), or 4 sh-RNAs against atg5 (*sh-atg5*), immunolabeled with an antibody against the extracellular region of GluA2 under control conditions or 15 min after LTD induction. Graph showing the surface labeling of GluA2, normalized to dendritic length under the aforementioned conditions. Bars represent mean values ± SEM. $N = 10$ independent experiments. Statistical analysis was performed using one-way ANOVA (F (5, 54) = 30.02, $P < 0.0001$) (Tukey's test, $P_{control/scr-control/atg5} = 0.0626$, $P_{NMDA/scr-NMDA/atg5} < 0.0001$, $P_{DHPG/scr-DHPG/atg5} < 0.0001$, $P_{control/atg5-NMDA/atg5} > 0.99$, $P_{control/atg5-DHPG/atg5} = 0.8602$, $P_{control/scr-NMDA/scr} = 0.0008$, $P_{control/scr-DHPG/scr} < 0.0001$). **c** Representative images of consecutive confocal z-planes of cultured neurons immunostained with antibodies against PSD95, LC3, and MAP2 to label the dendrites, 15 min after cLTD. Note the colocalization of PSD95 and LC3 in dendritic spines (yellow arrows) and in the dendritic shaft (white arrows), in consecutive z-planes. Scale bar: 10 μm. Graph showing the percentage of PSD95 puncta co-localizing with LC3 in consecutive confocal z-planes in dendritic spines and shafts in control neurons or 15 min after chemically induced NMDAR- or mGluR-LTD. Bars represent mean values ± SEM. $N = 8$ independent experiments. Statistical analysis was performed by one-way ANOVA (F(5,42) = 48.43, $P < 0.0001$) (Tukey's test for dendritic shaft, $P_{control-NMDA} = 0.0569$, $P_{control-DHPG} = 0.1948$, for dendritic spines, $P_{control-NMDA} < 0.0001$, $P_{control-DHPG} < 0.0001$). **d** Western blot analysis for GluA2 and PSD95 in lysates of cultured neurons in control conditions or 15 min after NMDAR- and mGluR-LTD and in the presence or absence of Bafilomycin A1 (50 μM) for 15 min before, during, and 15 min after the NMDA and DHPG pulses. **e** Western blot analysis for GluA2 and PSD95 in lysates of cultured neurons in control conditions or 15 min after NMDAR- and mGluR-LTD and in the presence or absence of SBI-0206965 (500 nM) for 30 min before, during, and 15 min after the NMDA and DHPG pulses. **f** Western blot analysis for GluA2 and PSD95 in lysates of cultured *shscrambled* or *sh-atg5* expressing neurons in control conditions or 15 min after NMDAR- and mGluR-LTD. **d–f** Graphs showing the levels of PSD95 and GluA2 levels in the indicated conditions, normalized to total protein levels. Bars represent mean values ± SEM. Statistical analysis was performed by one-way ANOVA. **d** (N = 9 independent experiments) PSD95: F(5,48) = 15.08, $P < 0.0001$ (Tukey's test $P_{control-control/Baf} = 0.7566$, $P_{control-NMDA} = 0.0016$, $P_{control-DHPG} = 0.0081$, $P_{NMDA-NMDA/Baf} < 0.0001$, $P_{DHPG-DHPG/Baf} = 0.0013$. GluA2: F(5,48)=6.627, $P < 0.0001$ (Tukey's test $P_{control-control/Baf} = 0.9692$, $P_{control-NMDA} = 0.0014$, $P_{control-DHPG} = 0.0067$, $P_{NMDA-NMDA/Baf} = 0.0421$, $P_{DHPG-DHPG/Baf} = 0.0127$. **e** (N = 7 independent experiments) PSD95: F(5,36) = 23.80, $P < 0.0001$. (Tukey's test $P_{control-control/SBI} > 0.99$, $P_{NMDA-NMDA/SBI} < 0.0001$, $P_{DHPG-DHPG/SBI} < 0.0001$, $P_{control-NMDA} < 0.0001$, $P_{control-DHPG} < 0.0001$, $P_{control/SBI-NMDA/SBI} = 0.9764$, $P_{control/SBI-DHPG/SBI} = 0.6286$). Panel e, GluA2: F(5,36)=11.73, $P < 0.0001$. (Tukey's test $P_{control-control/SBI} = 0.9179$, $P_{NMDA-NMDA/SBI} = 0.0001$, $P_{DHPG-DHPG/SBI} = 0.0002$, $P_{control-NMDA} = 0.0099$, $P_{control-DHPG} = 0.0323$, $P_{control/SBI-NMDA/SBI} = 0.9959$, $P_{control/SBI-DHPG/SBI} = 0.9407$). **f** (N = 7 independent experiments) PSD95: F(5,36) = 10.93, $P < 0.0001$. (Tukey's test $P_{control/scr-control/atg5} = 0.7927$, $P_{NMDA/scr-NMDA/atg5} = 0.0045$, $P_{DHPG/scr-DHPG/atg5} = 0.0003$, $P_{control/scr-NMDA/scr} = 0.0134$, $P_{control/scr-DHPG/scr} = 0.0030$, $P_{control/atg5-NMDA/atg5} = 0.9488$, $P_{control/atg5-DHPG/atg5} = 0.9976$). GluA2: F(5,36) = 10.79, $P < 0.0001$. (Tukey's test $P_{control/scr-control/atg5} > 0.99$, $P_{NMDA/scr-NMDA/atg5} = 0,0001$, $P_{DHPG/scr-DHPG/atg5} = 0.0019$, $P_{control/scr-NMDA/scr} = 0.0134$, $P_{control/scr-DHPG/scr} = 0.0021$, $P_{control/atg5-NMDA/atg5} = 0.5844$, $P_{control/atg5-DHPG/atg5} > 0.99$).

Cell component enrichment analysis indicated that the dynamic "LTD-enriched" cargo consists mainly of proteins localized to the postsynaptic cytoskeleton, including myosin-II filament proteins and to postsynaptic membrane anchoring complexes (Fig. 5e). This group also contained AMPA receptor complex proteins, including in particular AMPA receptor subunits (GRIA1, 2/3) (Fig. 5e,f). As summarized in Fig. 5f, the "LTD-enriched" cargo further included prominent scaffold proteins involved in receptor localization to the postsynaptic density, kinases involved in synaptic plasticity, cell adhesion molecules, and cytoskeletal proteins, as well as a number of presynaptic proteins involved in neurotransmitter release.

The presence of synaptic proteins in the AV cargo urged us to exclude the possibility that it may result from contamination of the AV fraction with broken synapses, which may have formed synaptosome-like structures during the homogenization process. To further ensure that the AV fraction is largely devoid of synapses, we included an additional step, whereby we used magnetic beads to immuno-purify AVs from the AV fraction with an antibody against LC3. As shown in Supplementary Fig. 5e, the vast majority of the AV fraction content was immuno-precipitated with the LC3 antibody (lane 2), whereas no material was precipitated with an IgG control immunoprecipitation (lane 3). Only a very small portion of the input was unbound in the LC3-immunoprecipitation condition (lane 4), and it contained some LC3, suggesting that the immunoprecipitation was largely, but not 100% effective as some LC3-positive membranes were left in the unbound material. However, the unbound material of the control IgG precipitation contained much more protein (lane 5). Importantly, the levels of PSD95 were depleted in the LC3 unbound material (lane 4) compared to the IgG unbound (lane 5), indicating that synaptic proteins are not contaminants, but are specifically precipitated with AVs.

In order to biochemically validate the proteomic data, we performed western blot analyses in purified control and LTD-AVs, which were treated with Proteinase-K to shave externally associated proteins off their outer surface. A sample of purified postsynaptic density (PSD) was also loaded for reference. This analysis confirmed that the levels of AMPAR subunits GluA1 (GRIA1) and GluA2 (GRIA2) are significantly enriched in LTD-AVs, compared to control (Fig. 5g), in line with the results presented in Fig. 4d–f. Moreover, scaffold proteins such as PICK1 and SAP97 are also enriched (Fig. 5g,f). Similarly, enrichment was confirmed for all other postsynaptic proteins that were selected from the proteomic list, including kinases such as αCamKII and FYN, cytoskeletal proteins and their modifiers, such as alpha-internexin (INA), myosin 10 (MYH10), and ITPKA, among others (Fig. 5g, f). The levels of the validated proteins in control and LTD-AVs were quantified from 3 independent experiments (Fig. 5g) and normalized to p62, a cargo protein whose levels are not affected, at least during early LTD (Supplementary Fig. 1g).

Enrichment analyses for cellular component (Fig. 5e) also indicated that the "LTD-excluded" cargo is enriched for proteins that localize in the postsynaptic endocytic zone, including dynamin, which is required for AMPAR endocytosis, and proteins that localize to the spine head, including cofilin-1, an actin filament-severing protein, with established function in dendritic spine shrinkage and elimination during LTD[54,55]. In line with the proteomic data, western blot analysis confirmed that

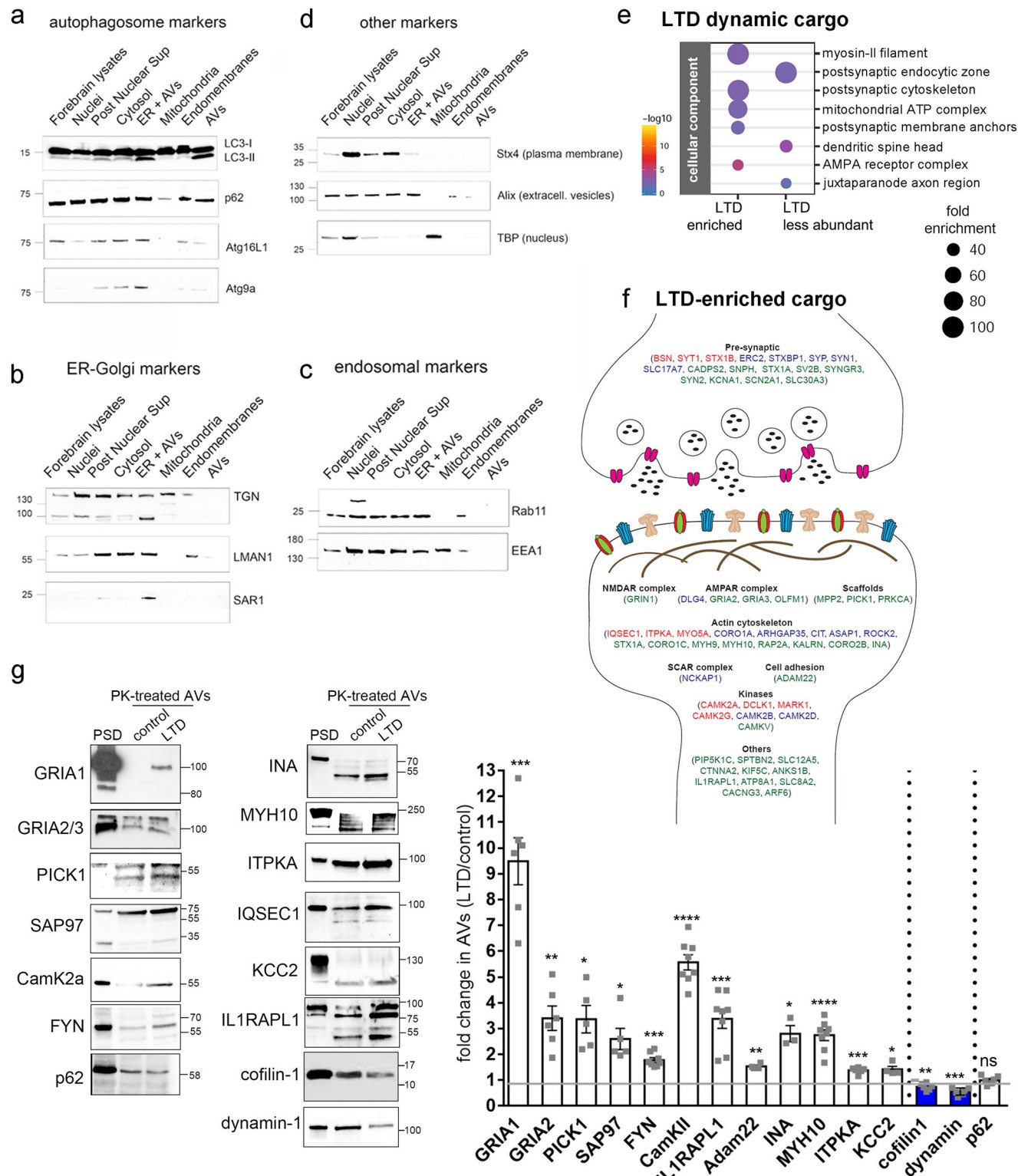

the levels of dynamin-1 and cofilin-1 are significantly reduced in Proteinase-K-treated AVs in LTD compared to control (Fig. 5g).

**Blockade of the autophagy initiation machinery abolishes LTD.** Next, we examined the role of AV biogenesis in LTD in hippocampal slices. To this end, we bath applied SBI-0206965 to acute hippocampal slices for 15 or 60 min, while they were maintained in oxygenated aCSF and then collected lysates. We performed western blot analysis with antibodies against the phosphorylated

form of Atg13 at Ser318, which is the direct target of the ULK1 kinase, as well as with an antibody against total Atg13 and β-III tubulin. As shown in Supplementary Fig. 6a, after 15 min of SBI-0206965 treatment, p-Atg13 levels significantly decreased to about 40% of untreated controls and further decreased at the 60 min time point, indicating that the drug penetrates well into the slice and acts rapidly.

Therefore, we bath applied SBI-0206965 to wild-type P22-P28 hippocampal slices for 15 min before, during and 15 min after

**Fig. 5 Proteomic profiling of the autophagic cargo during NMDAR-LTD. a–d** Western blot analyses of different fractions along the autophagic vesicle purification procedure, using antibodies against **a** autophagosomal markers (LC3B, p62, Atg16L1, and Atg9A), **b** ER-Golgi markers (TGN, LMAN1, SAR1a), c endosomal markers (Rab11b, EEA1), and **d** markers of the plasma-membrane (Stx4), extracellular vesicles (Alix) and nuclear extracts (TBP). $N = 3$ independent experiments. **e** Graph showing the cell component analysis, as false discovery rate (FDR)-corrected $p$-values, of the dynamic cargo (total of 393 proteins) that is enriched (up) or less abundant (down) in AVs after LTD, compared to control. **f** Graphical representation of proteins enriched in AVs upon LTD, with relation to the synapse. **g** Western blot analysis of PK-treated control and LTD-AVs, validating the presence of the proteins identified by the proteomic analyses in the autophagic vesicles. Postsynaptic density (PSD) fraction was used as reference. Graph showing the fold change of the normalized levels of the proteins validated by western blot, as a ratio of LTD to control. Cargo proteins were normalized to the levels of p62, which remains unaffected at the early phase of LTD. $N = 3$ independent AV preparations. Bars represent mean values ± SEM. Statistical analysis was performed using paired, two-tailed Student's $t$-test (GluA1, $N = 6$, $P = 0.0002$; GluA2, $N = 6$, $P = 0.0039$; Pick1, $N = 5$, $P = 0.011$; SAP97, $N = 5$, $P = 0.0179$; FYN, $N = 8$, $P < 0.0001$; CamKIIa, $N = 8$, $P < 0.0001$; IL1RAPL1, $N = 8$, $P = 0.0004$; Adam22, $N = 4$, $P = 0.0018$; INA, $N = 3$, $P = 0.0287$; MYH10, $N = 8$, $P < 0.0001$; ITPKA, $N = 6$, $P = 0.0006$; KCC2, $N = 4$, $P = 0.0352$; cofilin-1, $N = 6$, $P = 0.005$; dynamin, $N = 6$, $P = 0.0005$; p62, $N = 6$, $P = 0.9809$). All indicated molecular weights in **a–d** and **g** are in kDaltons (kD).

LTD induction, using either an NMDAR-dependent LFS protocol (1200 pulses at 1.4 Hz) which is blocked in the presence of AP5 (Supplementary Fig. 6b) (Fig. 6a), an NMDA (20 μM, 3 min) (Fig. 6b) or a DHPG (100 μM, 10 min) (Fig. 6c) bath application to induce LTD, as previously described[44,56,57]. While untreated slices (aCSF control) exhibited the expected depression following all three different protocols, SBI-0206965 treated slices failed to exhibit depression and instead responded with sustained potentiation, as shown in the bar graph and time-plots in Fig. 6a–c. In line with our previous findings[36], SBI-0206965 did not affect the baseline synaptic transmission (Supplementary Fig. 6c), the paired-pulse facilitation ratio (Supplementary Fig. 6d), the fiber volley amplitude to fEPSP slope curve (Supplementary Fig. 6e), or the fiber volley amplitude to intensity curve (Supplementary Fig. 6f) as compared to control, untreated slices, indicating that this acute inhibition of AV biogenesis does not alter presynaptic transmission.

Taken together, these findings indicate that acute inhibition of AV biogenesis during the induction of NMDAR- or mGluR-LTD is sufficient to abolish long-term depression.

**Autophagy is required cell-autonomously in postsynaptic pyramidal neurons for LTD.** To test whether autophagy is cell-autonomously required in excitatory neurons for LTD, the *atg5* gene was conditionally ablated in excitatory neurons by crossing *atg5^f/f* with *thy1-Cre^ERT2* animals. Tamoxifen was administered to *thy1-Cre^ERT2;atg5^f/f* progeny for five consecutive days starting at postnatal day 15 (P15). Atg5 protein levels were examined by Western blot in hippocampal lysates of tamoxifen-treated *atg5^f/f* controls, and tamoxifen-treated *thy1-cKO*s at P22. As expected, Atg5 protein levels were dramatically reduced in *thy1-cKO*s compared to controls (Supplementary Fig. 7a). At the same time, we observed increased levels of the autophagic cargo/receptor p62 and decreased levels of LC3B-II, further confirming the efficient reduction of autophagy.

As a cross-talk between autophagy and the ubiquitin-proteasome system (UPS) has been described in several cell types[58], including cultured neurons[59], we tested whether *thy1-cKO* animals exhibit altered proteasome activity compared to control littermates. We found no significant differences between young adult (P40) *thy1-cKO* and control animals when comparing proteasomal activity in hippocampal (Supplementary Fig. 7b) and cortical (Supplementary Fig. 7c) lysates.

Therefore, we performed electrophysiology recordings in acute hippocampal slices prepared from P22 *thy1-cKO* and *atg5^f/f* control animals, both of which were treated with tamoxifen, as described above (Fig. 7a). This age was purposefully chosen, as *atg5* ablation starts at P15, thus reducing the accumulation of any defects from prolonged autophagy ablation. LFS protocol (1200 pulses at 1.4 Hz) in Shaffer collateral fibers successfully induced a sustained depression of the field excitatory postsynaptic potential

(fEPSP) in control slices but failed to trigger LTD in *thy1-cKO* slices (Fig. 7a). Similarly, NMDAR- and mGluR-LTD were successfully induced in control slices by bath application of NMDA (20 μM, 3 min) (Fig. 7b) or DHPG (100 μM, 10 min) (Fig. 7c), but completely failed in *thy1-cKO* slices. Taken together, these results indicate that both major types of LTD depend on the autophagic machinery in pyramidal neurons.

Notably, *thy1-cKO* hippocampal slices did not exhibit any differences in paired-pulse facilitation (Supplementary Fig. 7d) or in fiber volley to fEPSP slope curves (Supplementary Fig. 7e), suggesting that the ablation of *atg5* for a few days (from P15 to P22) in pyramidal neurons does not alter basal synaptic transmission. However, as *thy1-cKO* animals have an ablation of *atg5* in all pyramidal neurons, it is not clear if autophagy is required specifically in the postsynaptic partner to facilitate LTD. To answer this question, we selectively knocked down *atg5* in excitatory neurons of the CA1 area of the hippocampus in young adult (P60) animals. To this end, a single construct carrying GFP and four shRNAs against *atg5* under a pyramidal-neuron specific promoter (*CamK2a*) was stereotaxically injected in the CA1 area of the hippocampus in the one brain hemisphere, while a construct carrying mCherry and four control scrambled sequences were injected in the contralateral side of the same animal (Fig. 7d, schematic).

As shown in Figs. 7e, 2 months after the injections the constructs were expressed in the ipsi- and contralateral sides of the CA1, leaving the CA3 area unaffected. Western blot analysis indicated that LC3-II levels were significantly reduced in lysates from the shRNA-*atg5* compared to the shRNA-*scrambled* side, indicating that autophagy is efficiently impaired by the knockdown (Fig. 7f). LTD was measured in the CA1 *stratum radiatum* after LFS stimulation of the Schaffer collateral fibers using a protocol that successfully induces LTD in adult animals (3 trains of 1200 pulses at 2 Hz with 10 min intervals)[60] and compared between the control and knockdown sides. While LTD was successfully induced in the control side, it was abolished in the *atg5* knockdown side (Fig. 7g). Instead, LFS triggered a significant level of sustained potentiation in the *atg5* knockdown side of the hippocampal slices, which is similar to the potentiation observed after LFS in *thy1-cKO* (Fig. 7a) and in SBI-0206965-treated (Fig. 6a) hippocampal slices. Taken together, these results indicate that ablation of autophagy in postsynaptic neurons is sufficient to prevent LTD.

**Autophagy deficiency in pyramidal neurons affects cognitive flexibility.** LTD has been particularly recognized for its role in cognitive flexibility[61]. Therefore, we hypothesized that by compromising the ability of synapses to undergo LTD and instead triggering sustained potentiation, autophagy deficiency in excitatory neurons may alter cognitive flexibility in animals. To test this hypothesis, we compared *thy1-cKO* mice with control *atg5^f/f* mice, which were also treated with tamoxifen, in a home cage

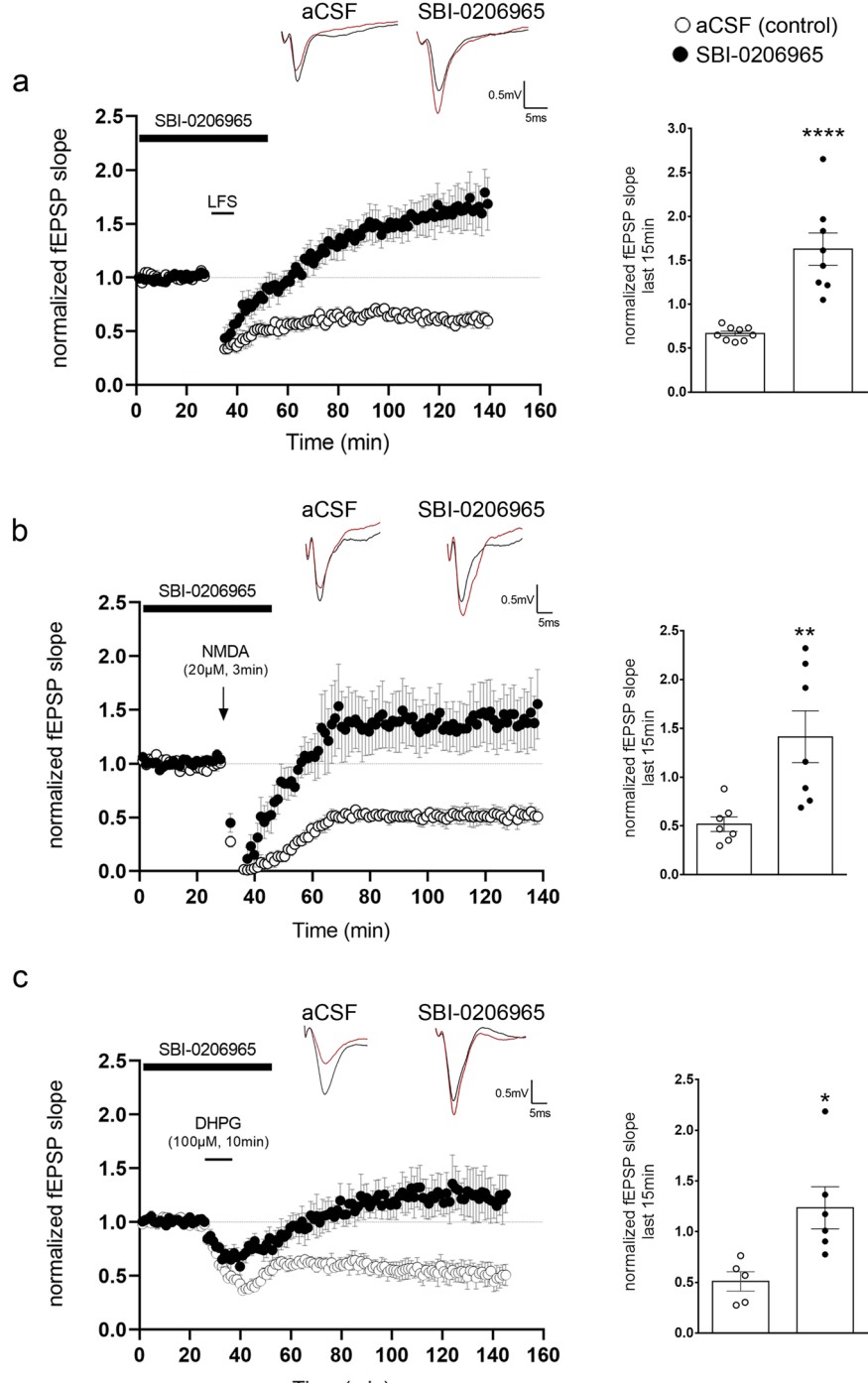

environment, using the CognitionWall test. In this test, a CognitionWall with three entry holes is placed in front of a reward dispender in the PhenoTyper and regular chow is removed. Mice were tested in a discrimination learning (DL) and reversal-learning (RL) task, consisting of 2 days of initial DL and 2 days of RL. Before the learning stages commenced, preference for one of the three holes was assessed. During the learning stages, mice had to learn to earn all their food by going through one of the three holes in the wall. Passing through the other holes was without any programmed consequences. One reward was delivered for every fifth entry through the correct hole (FR5 schedule of reinforcement)[62].

The initial preference for one of the three holes was not different between genotypes ($p = 0.094$, Supplementary Fig. 8a).

During initial DL *thy1-cKO* mice were not different from the *atg5^{f/f}* controls, in the total number of entries needed to reach an 80% performance criterion ($p = 0.7877$, Fig. 8a). Unexpectedly, during the subsequent RL phase, *thy1-cKO* mice were faster in reaching the 80% performance criterion ($p = 0.0454$, Fig. 8b). Differences in RL could not be explained by differences in overall locomotor activity, as both genotypes had similar values in the total distance traveled (Fig. 8d) during the task duration. However, *thy1-cKO* mice had a small but significant increase in the total number of wall hole entries (Fig. 8c).

The better performance of the *thy1-cKO* mice was surprising, given that reversal learning is LTD-dependent and our electrophysiology recordings clearly showed that both pharmacological and genetic ablation of autophagy in excitatory neurons prevents

**Fig. 6 Inhibition of autophagic vesicle biogenesis prevents NMDAR- and mGluR-LTD. a** Time-plot of normalized fEPSP slope recorded from WT P22-P28 hippocampal slices under control conditions (white circles) or bath application of SBI-0206965 (1 µM) (black circles) 15 min before, during and 15 min after LFS protocol (1200 pulses at 1.4 Hz). Representative traces of fEPSPs before (black) and after (red) LTD induction under control conditions (aCSF) and bath application of SBI-0206965. Graph showing the average normalized fEPSP slope during the last 15 min of recording following LFS under control conditions (white bar) or bath application of SBI-0206965 (black bar). Bars represent mean values ± SEM. $N = 9$ animals for control and $N = 8$ animals for SBI-0206965. Statistical analysis was performed using unpaired, two-tailed Student's $t$-test ($P < 0.0001$). **b,** Time-plot of normalized fEPSP slope recorded from WT P22-P28 hippocampal slices after bath application of NMDA (20 µM) for 3 min, under control conditions (white circles) or bath application of SBI-0206965 (black circles) 15 min before, during and 15 min after cLTD. Representative traces of fEPSPs before (black) and after (red) NMDA-induced LTD (cLTD) under control conditions and bath application of SBI-0206965. Graph showing the average normalized fEPSP slope during the last 15 min of recording following cLTD under control conditions (white bar) or bath application of SBI-0206965 (black bar). Bars represent mean values ± SEM. $N = 7$ animals per condition. Statistical analysis was performed using unpaired, two-tailed Student's $t$-test ($P = 0.0068$). **c** Time-plot of normalized fEPSP slope recorded from WT P22-P28 hippocampal slices after bath application of DHPG (100 µM) for 10 min, under control conditions (white circles) or bath application of SBI-0206965 (black circles) 15 min before, during and 15 min after cLTD. Representative traces of fEPSPs before (black) and after (red) DHPG-induced LTD (cLTD) under control conditions and bath application of SBI-0206965. Graph showing the average normalized fEPSP slope during the last 15 min of recording following cLTD under control conditions (white bar) or bath application of SBI-0206965 (black bar). Bars represent mean values ± SEM. $N = 5$ animals for DHPG and $N = 6$ animals for DHPG with SBI-0206965. Statistical analysis was performed using unpaired, two-tailed Student's $t$-test ($P = 0.0159$).

LTD and instead triggers potentiation. To test whether the enhanced reversal learning of the *thy1-cKO* mice is LTD-dependent, we repeated the behavioral experiments in *thy1-cKO* and *atg5^f/f* control mice that were administered with TAT-GluR23Y peptide, a cell-permeable peptide that contains the tyrosine-based signal ($_{869}$YKEGYNVYG$_{877}$), required for regulated AMPAR endocytosis. Previous work has demonstrated that TAT-GluR23Y blocks LTD-induced AMPAR endocytosis and specifically prevents LTD[63] without altering basal synaptic transmission or long-term potentiation (LTP). As a control, both genotypes received a control peptide (TAT-GluR23A). Peptides were administered on day 3 and day 4 of the task (reversal days) 1 h before commencing the test.

The initial preference for one of the three holes was similar between *thy1-cKO* or *atg5^f/f* mice, that were administered either with TAT-GluR23Y or the control peptide (Supplementary Fig. 8b). As shown in Fig. 8e, TAT-GluR23Y did not have any effect on the discrimination learning of *thy1-cKO* or *atg5^f/f* control mice, as compared to the control peptide ($p = 0.5549$ and $0.0740$ for *thy1-cKO* and *atg5^f/f*, respectively).

In reversal learning, *thy1-cKO* animals administered with the control peptide maintained a faster response to reaching the 80% performance criterion, as compared to *atg5^f/f* mice that were administered with the control peptide (Fig. 8f). This result was in line with the better performance of the *thy1-cKO* animals in Fig. 8b. As expected, and in line with previous studies[23,64], TAT-GluR23Y significantly reduced the reversal learning of *atg5^f/f* mice ($p = 0.0002$, Fig. 8f), compared to *atg5^f/f* mice that were administered the control peptide. However, TAT-GluR23Y failed to reduce the reversal learning of the *thy1-cKO* animals ($p = 0.1163$, Fig. 8f), compared to *thy1-cKO* mice that were administered the control peptide, suggesting that their performance is not due to LTD. Differences in RL could not be explained in differences in locomotor activity, as all groups exhibited similar values for the total number of wall hole entries ($p = 0.0979$, Fig. 8g) and in the total distance traveled ($p = 0.5916$, Fig. 8h) during the task duration. As it was previously shown that TAT-GluR23Y has no effect on long-term potentiation[65], it is plausible that the enhancement of reversal learning observed in the *thy1-cKO* animals may be at least partly accounted for by the potentiation of synapses following LTD-triggering stimuli.

## Discussion

The role of autophagy in long-term synaptic plasticity, conducive to protein turnover, is beginning to be elucidated. Our previous work demonstrated that BDNF, a key regulator of long-term potentiation (LTP) and memory, suppresses baseline autophagy in the adult brain. As a result, conditional *bdnf* mutants in the neural lineage exhibit increased autophagic flux and an over-abundance of AVs in the hippocampus. Interestingly, inhibition of autophagy fully rescues the LTP impairment caused by sequestration of endogenous BDNF[10]. Although the implication of autophagy in memory formation, erasure, and learning[6–8] is well-documented, so far autophagy has only been indirectly studied in the context of synaptic plasticity underlying these behaviors, leading to inconclusive and contradictory results[10,18,66]. Therefore, it remains unclear how long-term synaptic plasticity regulates autophagy at the cellular level, and whether autophagy is required in excitatory neurons for long-term plasticity.

In the present study, we address these questions in the context of LTD. We demonstrate that LTD-inducing activity triggers the initiation of autophagy in the dendrites of postsynaptic excitatory neurons, thus revealing the cellular site of the cross-talk between neuronal activity and autophagy during LTD. Moreover, we show that in turn, autophagy is cell-autonomously required in postsynaptic pyramidal neurons for the induction of both major forms of LTD that co-exist in the hippocampus, mediated by activation of either NMDA or group I metabotropic glutamate receptors. In our models, we did not observe altered synaptic transmission after acute pharmacological inhibition of the ULK1 kinase activity, or when *atg5* was genetically ablated in mature excitatory neurons for a short period. By contrast, a recent study showed that developmental ablation of *atg5* in pallial progenitors, using the *emx1-cre* deleter, results in enhanced excitatory transmission in the adult animals[17]. Thus, we believe that these differences are due to the duration and developmental time of the ablation.

Our findings contradict a recent study showing that autophagy inhibition facilitates LTD[66]. We believe that the controversy arises from the use of different experimental paradigms and genetic tools. For example, in our study, we monitor autophagy by imaging endogenous LC3-positive puncta in cultured neurons following chemical LTD, as well as by assessing the presence of autophagic vesicles upon LFS-induced LTD in CA1 postsynaptic dendrites in acute hippocampal slices (prepared from P22-28 animals) by electron microscopy. By contrast, the Shen *et al.*, study monitors autophagy by imaging overexpressed, fluorescently tagged LC3 in hippocampal slices (prepared from P6-8 animals) which are cultured for several days in vitro. Moreover, Shen et al., study examines LTD in adult animals, where *atg5* is ablated in the CA1 by the T29-1 deleter, before the window of developmental spine pruning. Given the well-described requirement of autophagy

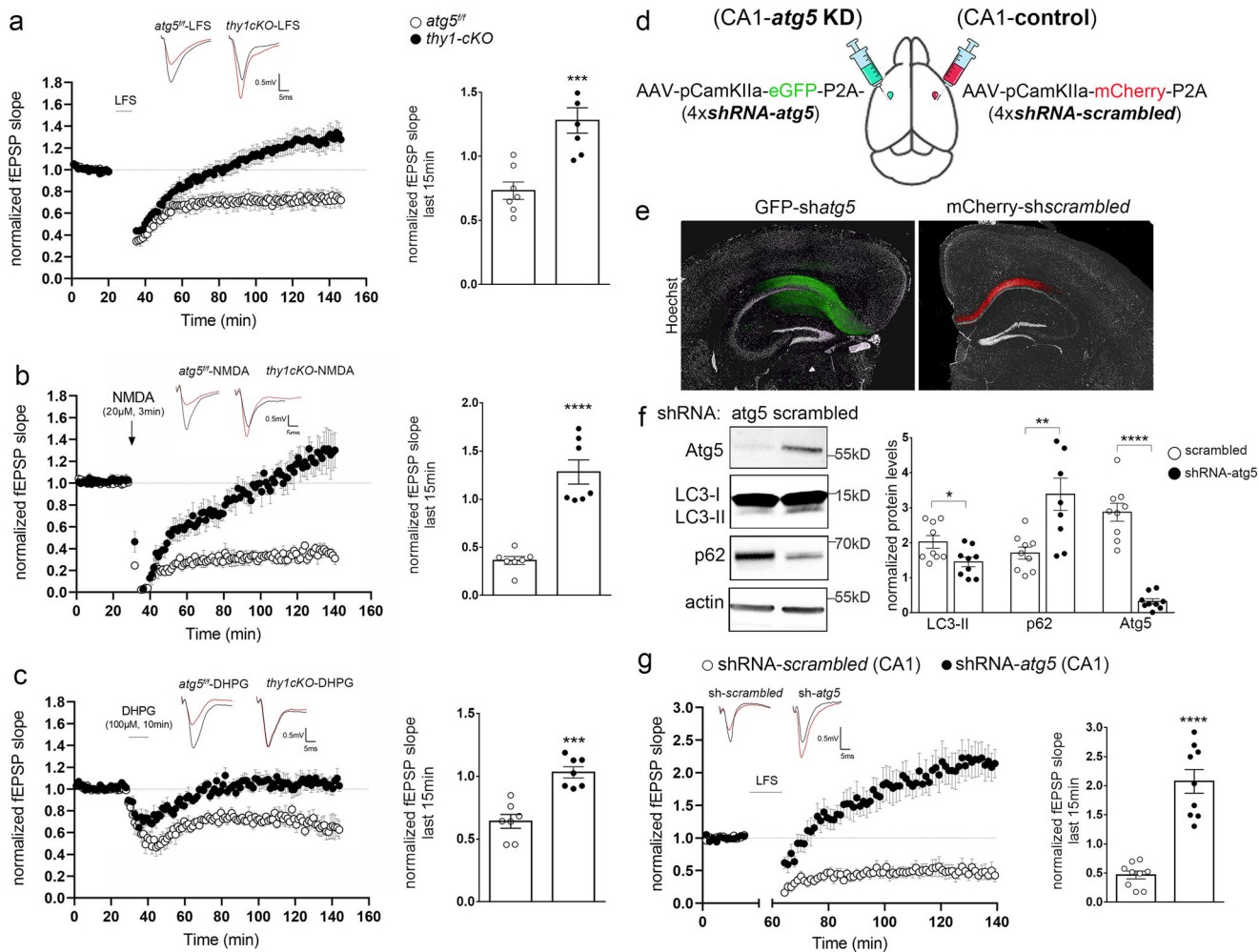

**Fig. 7 Autophagy is required cell-autonomously in postsynaptic pyramidal neurons for LTD. a–c** Time-plots of the normalized fEPSP slope and bar graphs of the average normalized fEPSP slope during last 15 min of recording after LFS (**a**) or NMDA (**b**) or DHPG (**c**) bath application in P22 hippocampal slices from control (atg5^f/f, white circles and bars) and thy1-cKO (black circles and bars) animals. N = 7 animals per genotype. Representative traces of fEPSPs before (black) and after (red) LTD induction for the two genotypes in every LTD-triggering protocol. **d** Schematic representation of protocol used to knockdown atg5 selectively in excitatory neurons of the CA1 area of the hippocampus in adult (P60) animals. **e** Images showing representative slices expressing AAV-CamK2a-shatg5-eGFP or AAV-CamK2-shscrambled-mCherry in CA1 area, stained with Hoechst dye (nuclei). **f** Western blot analyses of Atg5, LC3, p62, and actin in lysates from shRNA-atg5 and control hippocampal slices to validate the efficacy of the knockdown protocol. Graph showing the normalized protein levels of LC3, p62, and Atg5 in the indicated condition. N = 9 animals. **g** Time-plot of the normalized fEPSP slope and bar graphs of the average normalized fEPSP slope during the last 15 min of recording following LFS in slices expressing AAV-CamK2a-shscrambled-mCherry (white circles and bars) or AAV-CamK2a-shatg5-eGFP (black circles and bars) in the CA1 area. N = 9 animals per condition. Representative traces of fEPSPs before (black) and after (red) LTD induction for the two conditions. **a–c**, f, **g** Bars represent mean values ± SEM. Statistical analyses were performed using unpaired, two-tailed Student's t-test. **a**, P = 0.0007, **b**, P < 0.0001, **c**, P = 0.0001. In **f**, P = 0.0231 for LC3, P = 0.0034 for p62, and P < 0.0001 for ATG5. For **g**, P < 0.0001.

in this process in the cortex[20], these animals are likely to carry developmental deficits in spine pruning that can alter their LTD response and may explain the different results. However, as pruning periods may vary across different brain areas, this possibility would need to be experimentally examined.

There are also key differences in the pharmacological compounds that the two studies use to modulate autophagy in electrophysiology experiments measuring LTD. In our study, we have used SBI-0206965, a drug that inhibits the kinase activity of ULK1 and prevents the earliest steps of autophagic vesicle biogenesis. By contrast, Shen et al., study has used three different inhibitors that act on lysosomes to prevent their acidification or protease activity, hence targeting not autophagic vesicle formation and cargo recognition, but the latest step of autophagic cargo degradation. Moreover, the authors used

trehalose to pharmacologically induce autophagy. In addition to being a pleiotropic drug affecting multiple cellular processes, it has been recently debated whether trehalose induces autophagy and if its effects in the nervous system are autophagy-dependent[67,68]. Similarly, they use rapamycin as an autophagy inducer, which however also suppresses protein synthesis. However, it is also possible that the discrepancy actually reveals important differences in the interplay between autophagy and synaptic plasticity during different developmental windows, as in this paper most experiments with pharmacological modulation of autophagy were performed earlier time-points, during synaptogenesis. It is plausible that during earlier windows LTD is not accompanied by degradation of AMPA receptors and their scaffold proteins and may therefore not require autophagy.

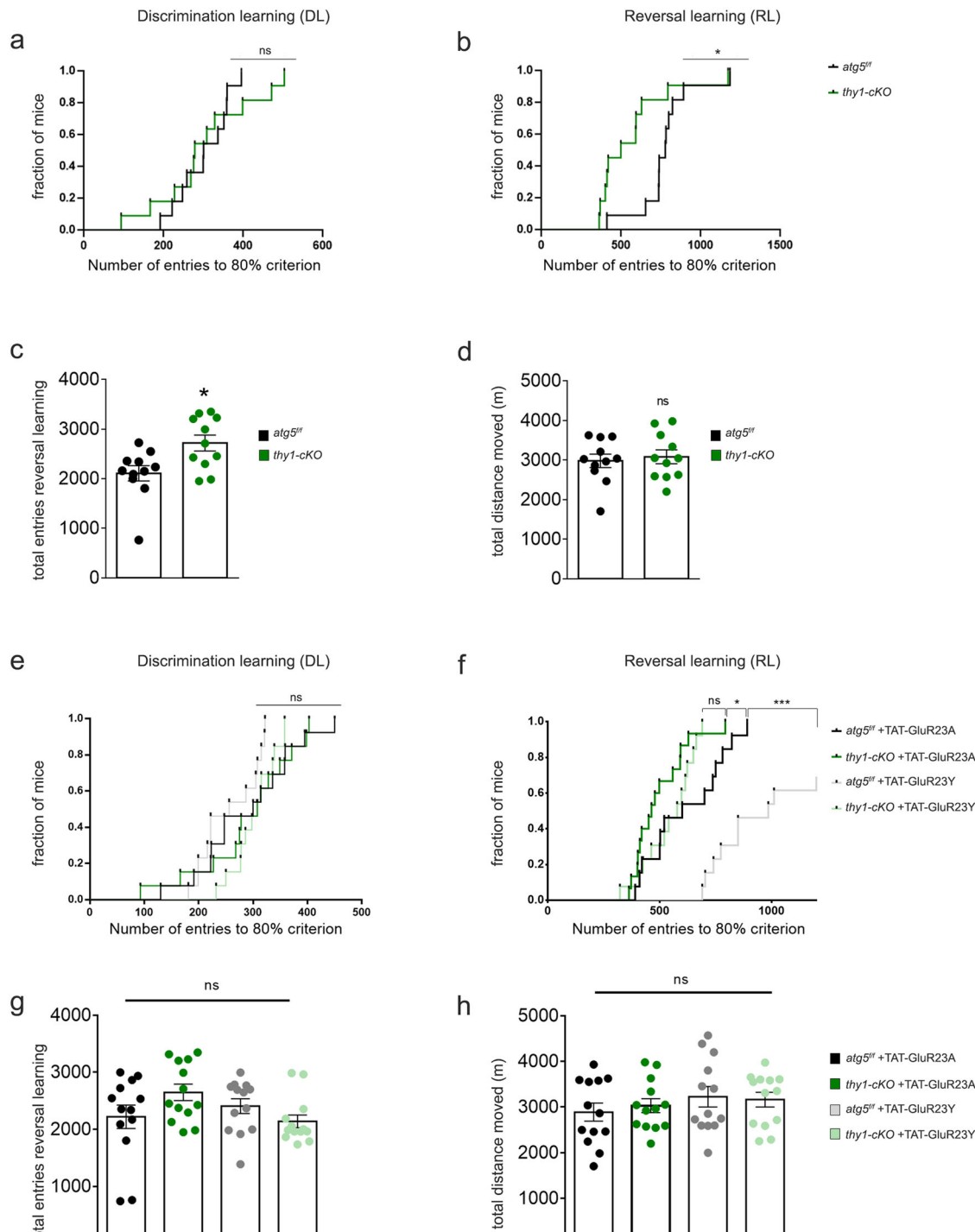

It is well-documented that under baseline conditions autophagic vesicle biogenesis is spatially restricted to the axon tip[25], making dendritic material inaccessible for selection by phagophores and for autophagic degradation. As LTD is expressed postsynaptically, it is difficult to conceptualize how autophagy can rapidly contribute to degradation in remote dendritic compartments. Therefore, the induction of autophagic biogenesis in dendrites by LTD-inducing activity allows postsynaptic material to be rapidly accessible on-site to phagophores and is a fine example of how autophagy can be specifically regulated in different neuronal compartments on demand. These findings also invite the speculation that

other forms of activity may also regulate autophagy in a compartment-specific manner.

A key question relates to the nature of the synaptic and plasticity-related proteins that are degraded by autophagy during LTD. Previous studies used candidate-based approaches to suggest that autophagy can degrade AMPA[18] and GABA_A receptor subunits[19]. In order to gain insight into autophagic cargo dynamics upon LTD, we performed unbiased quantitative proteomic analyses comparing intact AVs under baseline conditions or after NMDAR-LTD. An important first conclusion is that the autophagic cargo is largely stable, as the vast majority of cargo proteins show comparable abundance between these conditions,

**Fig. 8 Autophagy deficiency in excitatory neurons alters cognitive flexibility. a** Survival plot of discrimination learning (DL) indicating the total number of entries in the cognition wall till the 80% learning criterion for *atg5$^{f/f}$* and *thy1-cKO* animals. Each vertical segment of the curve represents a mouse that reached the criterion at the represented number of entries on the *x*-axis ($N = 11$ animals per genotype). Statistical analysis was performed using log-rank (Mantel-Cox) test ($P = 0.5354$). **b** Survival plot of reversal learning (RL) indicating the total number of entries in the cognition wall till the 80% reversal-learning criterion for *atg5$^{f/f}$* and *thy1-cKO* animals. Each vertical segment of the curve represents a mouse that reached the criterion at the represented number of entries on the *x*-axis ($N = 11$ animals per genotype). Statistical analysis was performed using log-rank (Mantel-Cox) test ($P = 0.0472$). **c** Graph showing the total entries from all entrances of the cognition wall during the RL phase between *atg5$^{f/f}$* and *thy1-cKO* animals ($N = 11$ animals per genotype). Statistical analysis was performed by unpaired, two-tailed Student's *t*-test ($P = 0.012$). **d** Graph showing the total distance moved in meters during both DL and RL of *atg5$^{f/f}$* and *thy1-cKO* animals ($N = 11$ animals per genotype). Statistical analysis was performed by unpaired, two-tailed Student's *t*-test ($P = 0.6912$). **e** Same as in **a**, survival plot of DL for *atg5$^{f/f}$* and *thy1-cKO* animals with administration of TAT-GluR23Y or TAT-GluR23A in the following two days of the experiment ($N = 13$ animals per genotype and treatment). Statistical analysis was performed using log-rank (Mantel-Cox) (for *thy1-cKO* animals $P_{GluR23Y\_GluR23A} = 0.1163$; for *atg5$^{f/f}$* animals $P_{GluR23Y\_GluR23A} = 0.00002$). **f** Same as in **b** survival plot of RL among *atg5$^{f/f}$* and *thy1-cKO* animals with administration of TAT-GluR23Y or TAT-GluR23A, 1h before RL on day 3 and 4 of the experiment ($N = 13$ animals per genotype and treatment). Statistical analysis was performed using log-rank (Mantel-Cox) (for *thy1-cKO* animals $P_{GluR23Y\_GluR23A} = 0.5549$, for *atg5$^{f/f}$* animals $P_{GluR23Y\_GluR23A} = 0.0740$). **g** Same as in **c**, for *atg5$^{f/f}$* and *thy1-cKO* animals with administration of TAT-GluR23Y or TAT-GluR23A ($N = 13$ per genotype and treatment). Bars represent mean values ± SEM. Statistical analyses were performed by one-way ANOVA (($F3,48$) = 2.220, $P = 0.0979$) (Tukey's test $P_{atg5f/f/TAT-GluR23A-thy1cKOTAT-GluR23A} = 0.2042$, $P_{atg5f/f/TAT-GluR23a-atg5f/f/TAT-GluR23Y} = 0.8198$, $P_{atg5f/f/TAT-GluR23A-thy1cKOTAT-GluR23Y} = 0.9816$, $P_{thy1cKOTAT-GluR23A-atg5f/f/TAT-GluR23Y} = 0.6768$, $P_{thy1cKOTAT-GluR23A-thy1cKO/TAT-GluR23Y} = 0.0965$, $P_{atg5f/f/TAT-GluR23A-thy1cKOTAT-GluR23Y} = 0.6004$). **h** same as in **d**, for *atg5$^{f/f}$* and *thy1-cKO* animals with administration of TAT-GluR23Y or TAT-GluR23A ($N = 13$ animals per genotype and treatment). Bars represent mean values ± SEM. Statistical analyses were performed by one-way ANOVA (($F3,48$) = 0.6423, $P = 0.5916$) (Tukey's test $P_{atg5f/f/TAT-GluR23A-thy1cKOTAT-GluR23A} = 0.9492$, $P_{atg5f/f/TAT-GluR23A-atg5f/f/TAT-GluR23Y} = 0.5819$, $P_{atg5f/f/TAT-GluR23A-thy1cKOTAT-GluR23Y} = 0.7271$, $P_{thy1cKO/TAT-GluR23A-atg5f/f/TAT-GluR23Y} = 0.8810$, $P_{thy1cKOTAT-GluR23A-thy1cKOTAT-GluR23Y} = 0.9583$, $P_{atg5f/f/TAT-GluR23Y-thy1cKOTAT-GluR23Y} = 0.9952$).

and only a small fraction of the cargo is dynamically regulated during LTD. Interestingly, proteins enriched in AVs after LTD comprise key players of the postsynapse. These include cell adhesion molecules, numerous modulators of the actin cytoskeleton, but also two AMPA receptor subunits (GRIA1 and 2/3), whose removal from the synapse is a well-documented prerequisite for NMDAR-LTD[48,49,69–72], and scaffold proteins, such as PSD95. This is consistent with our recent findings demonstrating that NMDAR-dependent LTD triggers a profound reorganization of PSD95, which requires the autophagy machinery to remove the T19-phosphorylated form of PSD95 from synapses and leads to an increase in AMPAR surface mobility and internalization[36].

On the other hand, part of the dynamic cargo consists of proteins that are less abundant in autophagic vesicles during LTD, and therefore selectively escape degradation. Two notable examples are cofilin-1, an actin filament-severing protein with an established role in spine plasticity[73], and dynamin, the GTPase required for the internalization of AMPAR subunits during LTD. These findings suggest that cargo selectivity may be regulated by synaptic plasticity, emphasizing the need to delineate the key players, such as specific adaptors, and potential post-translational modifications involved.

At the behavioral level, cognitive flexibility and reversal learning are known to depend on LTD. Unexpectedly, we found that animals with conditional autophagy deficit in excitatory neurons have enhanced performance in cognitive flexibility, compared to controls. This was despite the fact that these animals exhibit impaired LTD and instead LTD-triggering stimuli elicit potentiation. Consistent with this, the TAT-GluR23Y peptide failed to reduce the reversal-learning performance of the autophagy knockouts, whereas it was effective in suppressing reversal learning in control animals. Therefore, we postulate that the enhanced performance of the knockout animals may be explained by the enhanced potentiation triggered by LTD-stimuli. We speculate that synaptic potentiation may be underlined by impaired proteostasis of key molecules involved in synaptic plasticity in autophagy-deficient neurons. In fact, our proteomic analysis indicates that CamKII proteins, as well as calmodulin and several phosphatases, are contained in the purified AV fraction (Supplementary data 1). Both pharmacological and genetic experiments have shown that activation of CamKII is necessary for sustained potentiation and LTP induction[74–76].

In fact, active CaMKII can mimic LTP[77–81], suggesting that CamKII can orchestrate the biochemistry that underlies sustained potentiation. Therefore, one possibility, is that autophagy ablation leads to changes in the levels, phosphorylation pattern, or activation of CamKII proteins that may change the response of postsynaptic neurons to calcium levels during plasticity. In line with this speculation, previous work also reported that in mice overexpressing type1 adenylyl cyclase (AC1) under the *CamK2a* promoter, LFS protocol failed to induce LTD, whereas LTP was enhanced, without affecting basal transmission. Interestingly, these animals also exhibited enhanced flexibility in reversal learning in the Morris water maze test[82]. Conversely, mice lacking the phosphodiesterase 4B gene (PDE4B), which mediates the hydrolysis of cAMP, display enhanced LTD, yet they exhibit a deficit in reversal learning in the Morris water maze test[83]. Similarly, ArcKR knock-in mice, where the predominant Arc ubiquitination sites are mutated and Arc degradation is disrupted, display enhanced mGluR-LTD in the hippocampus, but exhibit cognitive inflexibility[84]. Another study demonstrated that vGLUT1 heterozygote mice displayed reduced LTP and exhibited a specific deficit in spatial reversal learning in the Morris water maze test[85]. Taken together, there is amble evidence that reversal learning may depend on the fine-tuning of synaptic plasticity and that interfering with this tuning may have both positive and negative effects on cognitive flexibility.

In concluding, our findings highlight the role of degradative autophagy in the context of synaptic plasticity and associated behaviors. They also contribute towards unraveling the diverse roles of autophagy in pre and postsynaptic function, as they are emerging from recent studies.

## Methods

**Animals.** The animal protocols of this study were approved by the Animal Ethics Committee of the Foundation for Research and Technology Hellas (FORTH), the veterinary service of the University of Bordeaux, and by the Swiss veterinary services of the Canton Vaud (Direction generale de l'agriculture, de la viticulture et des affaires veterinaires). The mice were maintained in a pathogen-free environment and housed in groups of five animals per cage with constant temperature and humidity and a 12 h/12 h light/dark cycle. The animals used were of C57BL/6 genetic background. Atg5$^{f/f}$ (Hara et al., 2006) and *SLICK-H-Thy1-cre/$^{ERT2}$-EYFP* (Jax labs strain 012708) mice were used for electrophysiology experiments and behavioral analyses. In *thy1-Cre$^{ERT2}$;atg5$^{f/f}$* progeny, tamoxifen was administered by intraperitoneal injections at a dose of 75 mg/kg body weight for five consecutive days starting at P15. For behavioral experiments, tamoxifen administration was as before but starting at P45.

**Chemicals, antibodies, and viruses.**

### Chemicals

| Chemical | Source | Cat # | Function | Final concentration |
|---|---|---|---|---|
| NMDA | Hellobio | HB0454 | Agonist of NMDAR | 20 or 50 μM |
| DHPG | Hellobio | HB0026 | Agonist of mGluR1/5 | 50 or 100 μM |
| Ifeprodil (Co 101244 hydrochloride) | Tocris | 2456 | Antagonist of NR2B | 10 μM |
| SBI-0206965 | Sigma | SML1540 | ULK1 kinase inhibitor | 500 nM or 1 μM |
| Wortmannin | Sigma | W1628 | Phosphatidylinositol 3 kinase (PI3K) inhibitor | 500 nM |
| Lactacystin | Provided by the Proteasomal activity kit, Millipore | APT280 | Proteasome inhibitor | 5 nM |
| MTEP hydrochloride | Hellobio | HB0348 | Antagonist of mGluR5 | 10 μM |
| JNJ16259685 | Hellobio | HB0348 | Antagonist of mGluR5 | 10 μM |
| Bafilomycin A1 | Sigma | B1793 | Inhibitor of V-ATPase-dependent acidification of lysosomes | 20 or 50 nM |
| Ciliobrevin-D | Calbiochem | 250401 | Specific blocker of AAA + ATPase motor cytoplasmic dynein | 25 μM |
| Dynamin inhibitor peptide myristoylated | Tocris | 1775 | Dynamin-1 inhibitor | 50 μM |
| Neurobiotin tracer (N-(2-Aminoethyl) biotinamide hydrochloride | Vector laboratories | SP-1120 | Tracer | 46.4 mM |
| DAB | Sigma | D5637 | Detection of neurobiotin-labeled dendrites | 0.5 mg/ml |
| Avidin–biotin-horseradish peroxidase complex | Sigma | PK-6100 | For detection of neurobiotin-labeled dendrites | 1:200 |

*Peptides*

| | | | | |
|---|---|---|---|---|
| TAT-GluR23A Fusion Peptide | ANASPEC | AS-64984 | Control peptide of TAT-GluR23Y | 3 μmol/kg |
| TAT-GluR23Y | ANASPEC | AS-6429 | Blockage of GluR1 internalization | 3 μmol/kg |

*Virus strains*

| | | | | |
|---|---|---|---|---|
| pAAV[miR30] CamKIIa_short>mCherry: Scramble[mir30-shRNA#1-4]:WPRE | VectorBuilder | VB191206-1077NHW | Infection of neuronal cultures and in vivo stereotaxic injections | |
| pAAV[miR30] CamKIIa_short>EGFP: mAtg5[shRNA#1]: mAtg5[shRNA#2] mAtg5[shRNA#3] mAtg5[shRNA#4] [mir30-shRNA#1] | VectorBuilder | VB191126-2687aqw | Infection of neuronal cultures and in vivo stereotaxic injections | |

*Antibodies*

| | | | |
|---|---|---|---|
| Atg16L1 E-10 | Santa Cruz | Sc-393274 | 1/1000 |
| LC3B | Santa Cruz | Sc-376404 | 1/1000 |
| LC3B | Sigma | L7543 | 1/1000 |
| Atg13 | Sigma | SAB4200100 | 1/1000 |
| MAP2 | Synaptic systems | 188004 | 1/100 |
| FIP200 | Cell signaling | 12436 | 1/1000 |
| Atg101 | Abcam | ab229235 | 1/1000 |
| ULK1 | Cell signaling | 8054 | 1/1000 |
| p62 | Calbiochem | DR1057 | 1/1000 |
| WIPI2 | Abcam | Ab105459 | 1/1000 |
| GluA2 (N-terminus) | Alomone | AGP-073 | 1/1000 |
| PSD95 | Invitrogen | MA1-046 | 1/1000 |
| Arc | Synaptic Systems | 156003 | 1/1000 |
| SAR1A | Invitrogen | PA19124 | 1/1000 |
| Atg13 pS318 | Rockland | 600-401-C49 | 1/1000 |
| β III tubulin (Tuj1) | Santa Cruz | Sc-8005 | 1/5000 |
| β III tubulin (Tuj1) | Abcam | Ab18207 | 1/1000 |
| actin (2Q1055) | Santa Cruz | Sc-58673 | 1/5000 |
| Atg5 | Abcam | Ab109490 | 1/1000 |
| Atg5 | Novus | NB110-53818 | 1/1000 |
| GRP78Bip | Abcam | Ab21685 | 1/1000 |
| cofilin-1 | ProteinTech | 66057-1-Ig | 1/1000 |
| GRIA2/3 | Milipore | AB1506 | 1/1000 |
| CamKII alpha | ThermoFischer | 13-7300 | 1/1000 |
| Dynamin-1 | Abcam | Ab52611 | 1/1000 |
| Fyn (FYN-01) | Invitrogen | MA1-19331 | 1/1000 |
| IL1RAPL1 | Invitrogen | PA5-96244 | 1/1000 |
| IQSEC1 | Invitrogen | PA5-95835 | 1/1000 |
| ITPKA | Invitrogen | PA5-85786 | 1/1000 |
| KCC2 | Invitrogen | PA5-78544 | 1/1000 |
| MYH10 | Invitrogen | PA5-88304 | 1/1000 |
| PICK1 | Invitrogen | PA1-073 | 1/1000 |
| SAP97 | Invitrogen | PA1-741 | 1/1000 |
| alpha-Internexin | ThermoFischer | 32-3600 | 1/1000 |
| Rab11B | Novus biological | NBP2-15085 | 1/1000 |
| GluA2 | Synaptic systems | 182211 | 1/1000 |
| PSD95 | Invitrogen | MA1-046 | 1/1000 |
| Atg9α | Novus Biologicals | NBP2-67616 | 1/1000 |
| EEA1 | Abcam | Ab2900 | 1/1000 |
| Stx4 | Synaptic systems | SYSY110041 | 1/1000 |
| Alix | Cell Signalling | 2171 S | 1/1000 |
| TBP | Abcam | Ab61411 | 1/1000 |
| LMAN1 | Invitrogen | PA1-074 | 1/1000 |
| TGN46 | Abcam | Ab16059 | 1/1000 |
| IgG | Milipore | 12-370 | 1/1000 |

**Primary neuronal cultures.** Cortices and hippocampi were isolated at embryonic day 15.5 (E15.5), rinsed in ice-cold PBS 1X, and centrifuged for 5 min at $1000 \times g$ at room temperature. Neurons were treated with 0.25% trypsin in PBS 1X for 30 min at 37 °C, followed by mechanical dissociation. After DMEM (Thermo Fisher Scientific, #41966) and FBS (Biosera) were applied in a one-to-one ratio for trypsin inactivation, the cells were centrifuged 5 min at $1000 \times g$. Neurons were plated at an initial density of 125,000 cells/cm$^2$ in 6 cm plates and 12-well plates containing 18 mm glass coverslips. All plates were coated overnight with poly-D-lysine (Sigma–Aldrich, A-003-E). Neurons were cultured in Gibco$^{TM}$ Neurobasal$^{TM}$ Medium (Fisher Scientific, A13712-01) supplemented with 2% B-27, 200 mM L-glutamine, 5 mg/ml penicillin, and 12.5 mg/ml streptomycin for 16–17 days in vitro. For experiments involving viral infections of cultured neurons, DIV6 neurons were infected with the aforementioned viruses (see table above) in a final dilution 1:1000. Eighteen to twenty-four hours after infection, the medium was replaced and cells were fixed or harvested for further analysis 10 days post-infection.

**Biochemical purification of synaptosomes.** Synaptosomes were isolated as previously described[86]. Briefly, brains were isolated from adult C57BL/6 male mice. The brains were rinsed and homogenized in solution A (0.32 M Sucrose, 1 mM NaHCO₃, 1 mM MgCl₂, 0.5 mM CaCl₂•H₂O, 10 mM Na pyrophosphate, nanopure water, protease inhibitors) using a glass homogenizer. Following dilution to 10% weight/volume in solution A, and centrifugation for 10 min at $710 \times g$ at 4 °C, the pellet was resuspended in solution A supernatant using the homogenizer. After a 10 min centrifugation at $1400 \times g$ at 4 °C for the removal of nuclei, the supernatant was collected and centrifuged for 10 min at $13,800 \times g$ at 4 °C. The pellet was resuspended in solution B (0.32 M sucrose, 1 mM NaHCO₃) using the homogenizer and 8 ml of the resulting sample were layered on a discontinuous sucrose gradient (10ml-layers of 1.2 M, 1 M and 0.85 M sucrose). Following centrifugation of 2 h at $82,500 \times g$ at 4 °C, synaptosomes were isolated between the 1.2 M and 1 M sucrose layers and stored at −80 °C.

**Biochemical purification of autophagic vesicles from hippocampal slices.** Isolated brains from P22-P28 C57BL/6 mice were sectioned in 200 μm thick coronal slices using a vibratome (Leica, VT1200S), in the presence of chilled (2–4 °C) artificial cerebrospinal fluid (aCSF), containing 124 mM NaCl, 3 mM KCl, 26 mM NaHCO₄, 1 mM MgSO₄, 2 mM CaCl₂, 1.25 mM NaH₂PO₄ and 10 mM glucose, equilibrated with 95% O₂ and 5% CO₂ gas mixture at pH 7.4. The slices were stored in oxygenated aCSF, at a constant temperature of 31 ± 0.5 °C, for at least 1 h and were treated with a NMDA pulse (50 μM for 10 min). The sections were collected in 10 ml of Buffer B (0.25 M sucrose, 1 mM EDTA pH 8.8, and 20 mM HEPES pH 7.4) for homogenization using a glass homogenizer (15 dounces, on ice). Following centrifugation for 2 min at $2000 \times g$ at 4 °C, the post-nuclear supernatant (PNS) was collected and autophagic vesicles was purified as previously described[10]. For the validation of the NMDAR-LTD cargo, the purified vesicles were subjected to proteinase-K (20 μg/ml) treatment for 20 min on ice to digest externally associated proteins to the outer surface of the vesicle. Proteinase-K was then inactivated with 4 mM PMSF for 10 min on ice and the material was centrifuged at $16,000 \times g$ for 20 min at 4 °C to pellet the autophagic vesicles.

**Immunoprecipitation of LC3b-positive autophagic vesicles.** Biochemically-purified autophagic vesicles (AV fraction) were further immunopurified using the Dynabeads$^{TM}$ Protein G immunoprecipitation kit (Life Technologies Europe BV, #10007D). Two hundred fifty micrograms of the AV fraction was immunopurified using the LC3b antibody (Sigma, L7543) or the IgG antibody (Milipore, #12-370) as control. Dynabeads were pre-incubated with the aforementioned antibodies for 30 min at room temperature with rotation. The purified AV fraction was added to the Ab-beads and was further incubated for 1 h incubation at room temperature with rotation. Elution of the immuno-precipitated structures was performed based on the manufacturer's instructions. Immunopurified material was then used for western blot.

**Quantitative proteomic analyses of purified autophagic vesicles.** For the mass spectrometric comparison of purified AVs from control and LTD conditions, 30 μg of lysates were subjected to an in-solution tryptic digest using a modified version of the Single-Pot Solid-Phase-enhanced Sample Preparation (SP3) protocol[87,88]. Here, samples were added to Sera-Mag Beads (Thermo Scientific, #4515-2105-050250, 6515-2105-050250) in 10 μl 15% formic acid and 30 μl of ethanol. The binding of proteins was achieved by shaking for 15 min at room temperature. SDS was removed by 4 subsequent washes with 200 μl of 70% ethanol. Proteins were digested with 0.4 μg of sequencing grade modified trypsin (Promega, #V5111) in 40 μl HEPES/NaOH, pH 8.4 in the presence of 1.25 mM TCEP and 5 mM chloroacetamide (Sigma–Aldrich, #C0267) overnight at room temperature. Beads were separated, washed with 10 μl of an aqueous solution of 2% DMSO, and the combined eluates were dried down. Peptides were

reconstituted in 10 μl of H$_2$O and reacted with 80 μg of TMT10plex (Thermo Scientific, #90111)[89] label reagent dissolved in 4 μl of acetonitrile for 1 h at room temperature. Excess TMT reagent was quenched by the addition of 4 μl of an aqueous solution of 5% hydroxylamine (Sigma–Aldrich, 438227). Peptides were mixed to achieve a 1:1 ratio across all TMT-channels. Mixed peptides were subjected to a reverse-phase clean-up step (OASIS HLB 96-well μElution Plate, Waters #186001828BA) and analyzed by LC-MS/MS on a Q Exactive Plus (Thermo Scentific), as previously described[90].

Briefly, peptides were separated using an UltiMate 3000 RSLC (Thermo Scientific) equipped with a trapping cartridge (Precolumn; C18 PepMap 100, 5 lm, 300 lm i.d. × 5 mm, 100 A°) and an analytical column (Waters nanoEase HSS C18 T3, 75 lm × 25 cm, 1.8 lm, 100 A°). Solvent A: aqueous 0.1% formic acid; Solvent B: 0.1% formic acid in acetonitrile (all solvents were of LC-MS grade). Peptides were loaded on the trapping cartridge using solvent A for 3 min with a flow of 30 μl/min. Peptides were separated on the analytical column with a constant flow of 0.3 μl/ minutes applying a 2 h gradient of 2–28% of solvent B in A, followed by an increase to 40% B. Peptides were directly analyzed in positive ion mode applying with a spray voltage of 2.3 kV and a capillary temperature of 320 °C using a Nanospray-Flex ion source and a Pico-Tip Emitter 360 lm OD × 20 lm ID; 10 lm tip (New Objective). MS spectra with a mass range of 375–1.200 $m/z$ were acquired in profile mode using a resolution of 70,000 [maximum fill time of 250 ms or a maximum of 3e6 ions (automatic gain control, AGC)]. Fragmentation was triggered for the top 10 peaks with charge 2–4 on the MS scan (data-dependent acquisition) with a 30-second dynamic exclusion window (normalized collision energy was 32). Precursors were isolated with a 0.7 $m/z$ window and MS/MS spectra were acquired in profile mode with a resolution of 35,000 (maximum fill time of 120 ms or an AGC target of 2e5 ions).

Acquired data were analyzed using IsobarQuant[91] and Mascot V2.4 (Matrix Science) using a reverse UniProt FASTA Mus musculus database (UP000000589) including common contaminants. The following modifications were taken into account: Carbamidomethyl (C, fixed), TMT10plex (K, fixed), Acetyl (N-term, variable), Oxidation (M, variable) and TMT10plex (N-term, variable). The mass error tolerance for full scan MS spectra was set to 10ppm and for MS/MS spectra to 0.02 Da. A maximum of 2 missed cleavages were allowed. A minimum of 2 unique peptides with a peptide length of at least seven amino acids and a false discovery rate below 0.01 were required on the peptide and protein level[92].

**Electron microscopy of purified AVs**. Purified forebrain AVs (13 mg) were pelleted by centrifugation at 16,000 × $g$ for 30 min at 4 °C. The pellet of AVs was resuspended and incubated for 20 min in 2.5% glutaraldehyde in 0.1 M cacodylate buffer. The AVs were washed for 20 min twice in 0.1 M cacodylate buffer and post-fixed in 1% OsO$_4$ in 0.1 M cacodylate buffer. After two washes of 20 min each in 0.1 M cacodylate buffer the pellet was dehydrated in a series of increasing concentrations of ethanol (30%, 50%, 70%, 90%, and 100%) for 10 min each time. Two more washes of 15 min each in absolute ethanol followed at room temperature and two washes in propylene oxide for 20 min each. Then, the pellet of AVs was treated in 1:3, 1:1, 3:1 of epoxy resin:propylenoxide, for 1 h per case at room temperature, and in the end the AVs were resuspended in pure epoxy resin for 1 h at room temperature. Finally, the AVs were transferred in pure epoxy resin for an overnight incubation by mild shaking. The next day, the AVs were transferred again in pure epoxy resin for 1-hour incubation at room temperature and then in molds with fresh resin for 48 h at 60 °C overnight for their analysis by electron microscopy. Ultrathin 70 nm-thick sections were placed on pioloform-coated copper slot grids, stained with lead nitrate, and were observed using a JEM-2100 transmission electron microscope (JEOL Ltd, Akishima, Tokyo, JAPAN) at 80 kV. Images were taken with an Orius camera (Gatan, Pleasanton, CA, USA). Electron micrographs were analyzed using the open-source ImageJ software.

**Immuno-labeling**. Cultured cortical and hippocampal neurons were rinsed in PBS 1X and then fixed for 15 min in 4% paraformaldehyde (PFA) in PBS 1X. After fixation, cells were rinsed with PBS 1X and incubated in blocking solution (10% Fetal Bovine Serum (FBS) and 0.2% Triton X-100 in PBS 1X) for 1 h at room temperature. Neurons were then incubated with primary antibodies, diluted in blocking solution at 4 °C overnight. The primary antibodies used were: LC3 (1:1000, Santa Cruz, sc-376404 and Sigma, L7543), Atg13 (1:1000, Sigma–Aldrich, SAB4200100), MAP2 (1:1000, Synaptic Systems, #188004), FIP200 (1:1000, Cell Signaling, #12436), Atg101 (1:1000, Cell Signaling, #13492), ULK1 (1:1000, Cell Signaling, #8054), p62 (1:2000, Calbiochem, DR1057), WIPI2 (1:1000, Abcam, ab105459), GluA2 (N-terminus, 1:1000, Alomone, AGP-073), PSD95 (1:1000, Invitrogen, MA1-046) and Arc (1:1000, Synaptic Systems, #156003). Following three rinses with PBS 1X, cells were incubated with secondary antibodies diluted in PBS 1X for 1 h at room temperature. The secondary antibodies used were: anti-rabbit Alexa 488 (1:1000, Abcam, ab150073), anti-mouse Alexa 594 (1:1000, Abcam, ab150116), and anti-guinea pig Alexa 647 (1:1000, Abcam, ab150187). Hoechst 33342 nuclear dye (1:5000, HelloBio, HB0787) was used for nuclei staining. After three PBS 1X rinses, neurons were mounted onto slides and confocal images of the fluorescently labeled proteins were captured using the Leica TCS SP8 and the Zeiss LSM900 inverted confocal microscopes.

**Western blotting**. Western blots, with slight modifications, were performed as previously described (Nikoletopoulou et al., 2017). Cells were collected in cold PBS 1X, incubated in RIPA buffer (50 mM Tris-HCl pH 7.4, 150 mM NaCl, 1 mM EDTA, 1% Triton X-100, 0.1% Na-deoxycholate, 0.1% SDS) supplemented with protease inhibitors for 1 h on ice, and centrifuged (20 min at 16.200 g, 4 °C). Protein samples were separated on a 7.5%, 10%, or 15% polyacrylamide gel and transferred to a nitrocellulose membrane (Millipore). Membranes were incubated in 5% milk (or 5% BSA in the case of phospho-antibodies) for 1 h at room temperature, and then in 5% milk with primary antibodies overnight at 4 °C. The primary antibodies used were: LC3 (1:1000, Santa Cruz, sc-376404 and Sigma L7543), p62 (1:5000, Calbiochem, DR1057), Atg13 (1:1000, Sigma–Aldrich, SAB4200100), Atg13 pS318 (1:1000, Rockland, #600-401-C49), FIP200 (1:1000, Cell Signaling, #12436), Atg101 (1:1000, Cell Signaling, #13492), ULK1 (1:1000, Cell Signaling, #8054), β III tubulin (Tuj1) (1:5000, Santa Cruz Biotechnology, sc-80005 and Abcam, ab18207), actin (2Q1055) (1:5000, Santa Cruz Biotechnology, sc-58673), Atg5 (1:1000, Abcam, ab109490, Novus NB110-53818), Atg16L1 E-10 (1:1000, Santa Cruz Biotechnology, sc-393274), GRP78Bip (1:1000, Abcam, ab21685), cofilin-1 (1:1000, Proteintech, #66057-1-Ig), GRIA1 (1:1000, Milipore, #AB1504), GRIA2/3 (1:1000, Milipore, #AB1506), CamKII alpha (1:1000, ThermoFischer, #13-7300), Dynamin-1 (1:1000, Abcam, ab52611), Fyn (FYN-01) (1:1000, Invitrogen, MA1-19331), IL1RAPL1 (1:1000, Invitrogen, PA5-96244), IQSEC1 (1:1000, Invitrogen, PA5-95835), ITPKA (1:1000, Invitrogen, PA5-85786), KCC2 (1:1000, Invitrogen, PA5-78544), MYH10 (1:1000, Invitrogen, PA5-88304), PICK1 (1:1000, Invitrogen, PA1-073), SAP97 (1:1000, Invitrogen, PA1-741), alpha-Internexin (1:1000, ThermoFischer, #32-3600), Rab11B (1:1000, Novus biological, NBP2-15085), GluA2 (1:1000, Synaptic Systems, #182211), PSD95 (1:1000, Invitrogen, MA1-046), Atg9α (1:1000, Novus Biologicals, NBP2-67616), EEA1 (1:1000, Abcam, ab2900), Stx4 (1:1000, Synaptic Systems, SYSY110041), Alix (1:1000, Cell Signalling, #2171 S), TBP (1:1000, Abcam, ab61411), TGN46 (1:1000, Abcam, ab16059), LMAN1 (1:1000, Invitrogen, PA1-074), SAR1A (1:1000, Invitrogen, PA19124). Following three 10-minute rinses with PBS 1X with 0.1% Tween-20, or TBS 1X with 0.1% Tween-20 in case of phospho-antibodies, membranes were incubated in 2% milk with the corresponding secondary horseradish peroxidase-conjugated antibodies (1:10.000, Abcam) for 1 h at room temperature. Finally, membranes were developed by chemiluminescence (SuperSignal Chemilumines-cent Substrate, West Pico and Femto, ThermoFisher Scientific) according to the manufacturer's instructions.

**Measurement of proteasome activity**. Proteasome activity was measured in mouse cortex or hippocampus lysates from P40 atg5$^{f/f}$;thy1-Cre and atg5$^{f/f}$ mice, using the 20 S Proteasome Activity Assay Kit (Millipore, APT280) according to manufacturer's instructions. In both cases, lactacystin a proteasome inhibitor was used as a negative control to block the proteasome activity.

**Ex vivo hippocampal preparations for staining**. Free-floating hippocampal slices of 100 μm were prepared from P25 male C57BL/6 mice. The slices were kept in aCSF with simultaneous oxygen supply (95% O$_2$, 5% CO$_2$) for at least 1 h before treatment as well as during treatment. Sections were incubated with or without 50 μM NMDA or DHPG for 10 min. After treatment the slices were allowed for 15 min in oxygenated aCSF in order for the protein synthesis to take place. After fixation with 4% paraformaldehyde (PFA) and permeabilization overnight with 0.5% Triton X-100 in PBS 1X, nonspecific binding sites were blocked for 5 h using 20% BSA in PBS 1X, incubated overnight with primary antibodies (as described under immuno-labeling section), and then with secondary antibodies for 5 h in PBS 1X. Finally, sections were mounted in 80% glycerol, and images were acquired using Leica TCS SP8 inverted confocal microscope.

**Electrophysiological recordings**. Electrophysiological experiments were performed in brains of P22-P28 male mice that were isolated and sectioned in 400 μm thick coronal slices using a vibratome (Leica, VT1000S, Leica Biosystems GmbH, Wetzlar, Germany) in the presence of chilled (2–4 °C) oxygenated (95% O$_2$, 5% CO$_2$) artificial cerebrospinal fluid (aCSF) containing 124 mM NaCl, 3 mM KCl, 26 mM NaHCO$_3$, 1 mM MgSO$_4$, 2 mM CaCl$_2$, 1.25 mM NaH$_2$PO$_4$ and 10 mM glucose at pH 7.4. The slices stored in oxygenated aCSF, at a constant temperature of 31.5 ± 0.5 °C, for at least 1 h before recording, and then transferred to a submerged recording chamber, continuously perfused with oxygenated aCSF (3 ml/min) by a peristaltic pump. Glass microelectrodes were placed in the *stratum radiatum* (SR) layer of the CA1 area. Platinum/iridium metal microelectrodes (Harvard apparatus) or glass microelectrodes filled with 2 M NaCl connected to a constant current stimulator (Digitimer Ltd, UK) were also placed in the SR layer, about 300–400 μm away from the recording electrode to stimulate Schaffer collateral fibers in the CA1 area. The voltage responses were amplified using a Multiclamp 700B (Molecular Devices LLC., USA) digitized using the Axon Digidata 1550B (Molecular Devices LLC., USA). The electrical stimulus consisted of a 100 μs constant current pulse given at an intensity that generated 50% of the maximum fEPSP amplitude. Data were acquired and analyzed using pCLAMP 10 Software Suite (Molecular Devices LLC).

For NMDAR- and mGluR- LTD in P22 thy1-cKO mice, NMDA (20 μM) or DHPG (100 μM) was applied for 3 min or 10 min respectively in the aCSF during

the recording and following at least 20 min of monitoring baseline responses. The fEPSP was recorded for at least 100 min following wash-out of NMDA or DHPG.

For NMDAR- and mGluR- LTD in P22-P28 wild-type mice, NMDA (20 µM) or DHPG (100 µM) was applied for 3 min or 10 min respectively in the aCSF during the recording and following at least 20 min of monitoring baseline responses, under control conditions or with simultaneous bath application of SBI-0206965 (1 µM) for 15 min before, during and 15 min after NMDA or DHPG treatment.

For LFS-induced LTD, baseline responses were monitored for at least 20 min before the LFS protocol (1200 pulses at 1.4 Hz). The fEPSP was monitored for at least 100 min following the LFS protocol. In order to test the contribution of autophagy to LFS-induced LTD, brain slices were either incubated in aCSF or with the selective autophagy inhibitor SBI-0206965 (1 µM) for 15 min before, during and 15 min after the LFS.

Brain slices were incubated in aCSF with the dynein inhibitor Ciliobrevin-D (25 µM) for at least 20 min before, during and for at least 50 min following the end of the LFS protocol.

For LFS-induced LTD in adult mice injected with AAV-Camk2-sh-scramble-mCherry or AAV-Camk2-shatg5-GFP, baseline responses were monitored for at least 15 min before the LFS protocol (3 stimulations of 1200 pulses at 2 Hz, applied with a time interval of 10 min).

The fEPSP of each response was normalized to the 20 min pre-LFS, pre- NMDA or pre-DHPG average fEPSP. Analyses were performed using Student's t-tests.

The basal excitatory synaptic transmission was measured by an input/output stimulus-response curve and the fEPSP slopes were plotted against fiber volley amplitude as a function of increasing stimulation intensity. The response at each stimulation intensity was the average response of 3 consecutive stimulations delivered every 30 s. Short-term synaptic facilitation was measured by delivering two pulses with interstimulus intervals 50–500 ms. The stimulus was delivered at an intensity inducing one-third of the maximum response. The paired-pulse facilitation was calculated as the ratio of the second response amplitude to the first. The measurement at each interstimulus interval was the average of 3 consecutive two-pulse stimulations delivered every 20 s.

**Neurobiotin-labeling and electron microscopy of hippocampal neurons**. In a series of experiments, the recording electrode was filled with 1.5% neurobiotin (N-(2-Aminoethyl) biotinamide hydrochloride, FW 322.85 g/mol; Vector laboratories Inc., Burlingame, CA, USA) in 2 M NaCl, and the recording in slices was carried out as previously described. After the recording, the slices were fixed overnight in a fixative containing 4% paraformaldehyde in 0.1 M phosphate buffer (PB). After several washes in PB, the slices were embedded in 4% low melting agarose and cut in 70 µm thick sections in a vibrating microtome (Leica, VT1000S). Sections around the tip of the recording electrode were collected and cryoprotected in 10% and 20% sucrose solutions in 0.1 M PB, freeze-thawed in liquid nitrogen and processed for the visualization of neurobiotin-labeled structures. In brief, sections were incubated for approximately 65 h in avidin–biotin–horseradish peroxidase complex (VECTASTAIN® Elite® ABC-HRP Kit, Vector Laboratories, Burlingame, CA, U.S.A.; ABC, diluted 1:200 in TBS), and neurobiotin was visualized with 3,3′-Diaminobenzidine tetrahydrochloride hydrate (DAB) (0.5 mg/mL; Sigma–Aldrich, Gillingham, UK) and 0.02% $H_2O_2$ as substrate. Sections were treated with 1.33% $OsO_4$ and contrasted in 1% uranyl acetate, dehydrated in a series of ethanol and propylene oxide, and flat embedded in epoxy resin (Durcupan ACM, Fluca, Sigma-Aldrich, Gillingham, UK) on microscope slides. After polymerization of resin (60 °C, overnight), regions of interest were re-embedded in epoxy resin blocks. Serial electron microscopic sections (70–75 nm) were collected on pioloform-coated copper slot grids, contrasted with lead, and observed using a JEM-2100 transmission electron microscope (JEOL Ltd, Akishima, Tokyo, JAPAN) at 80 kV. Images were taken with an Orius camera (Gatan, Pleasanton, CA, USA). Electron micrographs were analyzed using the open-source ImageJ software.

**Behavioral analyses**. Adult mice of atg5f/f and thy1 cre;atg5f/f genotype were used for this experiment (11 animals per group). Tamoxifen (75 mg/kg) was administered at P45 both to control (atg5f/f) and to thy1-cre;atg5f/f animals by i.p. injections for 5 consecutive days. Behavioral testing was examined 15 days after the end of tamoxifen injections. Mice were introduced in the PhenoTypers at 10am for habituation (one animal per PhenoTyper). Animals were deprived of food since then, and water was provided ab libitum throughout the experiment. After 5 h and before the beginning of the experiment, a wall with three holes was introduced (Cognition wall). Pellets are given as a reward to the animals every five entries (non-consecutive) from the correct entrance. The whole protocol lasting 4 days, consisted of two discrete phases. The first two days there was the initial discrimination learning (IDL) and the two last the reversal learning (RL). During IDL the animals learn to gain food from the left hole of the wall, while in the RL the entrance is changed to the right, reversing the already acquired learning[62].

The total number of entries needed to reach a criterion of 80%, was calculated per fraction of mice and was used to define the initial discrimination and the reverse learning efficacy. The number of total errors to reach the 80% criterion, during reversal learning, was also monitored, as well as the total number of entries through the cognition wall during the four days of the experiment. The side bias was calculated by the number of entries of the animal from the left, middle or right

entrance at the first 30 entries after the beginning of the protocol. In this way the preference of a mouse for a specific entrance was excluded.

The aforementioned experimental protocol and analysis were also applied, to validate the cognitive flexibility of atg5f/f and thy1cre;atg5f/f animals in the presence of TAT-GluR23Y (ANASPEC, AS-64429), which inhibits the internalization of GluR2-containing AMPARs. TAT-GluR23A (ANASPEC, AS-64984) was used as a control peptide (13 animals per genotype or treatment). During the reversal-learning phase, specifically the third and fourth day during the experiment, animals were subjected to an intraperitoneal injection of either the TAT-GluR23Y (3 µmol/kg) or TAT-GluR23A (3 µmol/kg), as previously described[64].

Automated screening data after cognition wall experiment were conducted by AHCODA Data analysis. Raw data after the initial analysis were further analyzed in GraphPad prism 8.4.3.

**Stereotaxic injections**. Adult mice (P60) were anesthetized with a mixture of ketamine (75 mg/kg) and dormitor (1 mg/kg) and placed in a stereotaxic frame. A sagittal scalp incision was made to expose the skull and holes were drilled bilaterally above the dorsal hippocampus (anteroposterior = −1.9, mediolateral = ±1.5, ventral = −1.5 mm from bregma) according to mouse atlas (Paxinos and Franklin 2001). Each hemisphere was microinjected with either AAV-Camk2-shscrambled-mCherry or AAV-Camk2-shatg5-GFP virus (titer $1.37 \times 10^{11}$ GC/ml and $3.89 \times 10^{12}$ GC/ml respectively, Vectorbuilder, 1.0 µl/side) through glass micropipette connected via polyethylene tubing to a microsyringe (Hamilton) at a rate of 0.12 µl/min. Following microinjection, a micropipette was left in place for an additional 5 min to ensure diffusion of vector and then slowly retracted. The incision was closed with ethicon coated vicryl, and mice treated with 0.5 ml of saline to prevent dehydration and were awakened by the intramuscular injection of Antisedan (2 mg/kg). During the post-surgery recovery period, paracetamol was added to the drinking water (500 mg/250 ml) for 72 h.

**Analysis of confocal images**. All images of neuronal dendrites in this study were captured as a z-stack as well as an orthogonal projection with the following image acquisition settings: ×63 magnification, 3× zoom, 0.3 µm step size, and high resolution (1024 × 1024 pixels), using an LSM900 or an SP8 laser scanning confocal microscope. Following image acquisition, images were analyzed by ImageJ software and quantified blindly to the result.

For GluA2 surface labeling, quantification of fluorescence was achieved by measuring the raw integrated density (RawIntDensity) per dendritic length (µm) using the raw images and focusing on a specific region of interest (ROI) of the circumference of each dendrite.

For all other measurements of the structures /puncta per dendritic length (µm), a region of interest (ROI) was formed around the circumference of each dendrite. The number of positive structures was measured by the ComDet plugin version 0.5.0 of Image J with an approximate particle size of 4.0 pixels and intensity threshold (in SD): 5.0 pixels.

**dSTORM experiments and analysis**

*Rat hippocampal neuron culture and transfection*. Sprague-Dawley pregnant rats (Janvier Labs, Saint-Berthevin, France) were sacrificed according to the European Directive rules (2010/63/EU). Primary hippocampal cultures were prepared from E18 rat embryos according to the Banker protocol (Kaech and Banker, 2006). Hippocampi were dissected in Petri dishes filled with HBSS and HEPES, and dissociated by trypsin treatment (0.05%; Gibco) at 37 °C. 4 poly-L-lysine pre-coated 1.5H 18 mm coverslips were introduced in 60-mm dishes, which were pre-plated with 250,000 cells. After 2 h, coverslips were transferred to dishes containing an astrocyte feeder layer. Neuron cultures were maintained in Neurobasal medium supplemented with 2mM L-glutamine and 1X NeuroCult SM1 Neuronal supplement (STEMCELL technologies) at 37 °C and 5% $CO_2$, for 7-8 days.

*Sample preparation and immuno-labeling*. Primary neuronal cultures were treated with 50 µM NMDA (Tocris) for 5 min (Lee et al., 1998) or with 50 µM DHPG (Sigma–Aldrich) for 5 min. After 15 or 60 min, neurons were fixed with PFA (4%). Then, cells were washed three times for 5 min in 1x PBS. PFA was quenched with $NH_4Cl$ 50 mM for 10 min. Cells were then permeabilized with 0.2% Triton X-100 for 5 min and incubated in the presence of BSA (1%) for 30 min. Cells were incubated by anti-LC3 antibody (Sigma) for 60 min at 37 °C. Unspecific staining was blocked by incubating coverslips in 1% BSA for 1 h at room temperature. Primary antibodies were revealed with Alexa 647 coupled secondary antibodies.

*direct STochastic Optical Reconstruction Microscopy (dSTORM)*. dSTORM experiments were done on fixed immunolabeled neurons. dSTORM imaging was performed on a LEICA DMi8 mounted on an anti-vibrational table (TMC, USA), using a Leica HCX PL APO 160×1.43 NA oil immersion TIRF objective and fibber-coupled laser launch (405 nm, 488 nm, 532 nm, 561 nm, and 642 nm) (Roper Scientific, Evry, France) as detailed in Haas et al. 2018.

**Statistical analysis**. For the statistical analyses of the experiments included in this manuscript N > 3 independent experiments were conducted. The graph bars represent mean values ± SEM. GraphPad was used for statistical analysis and graph

design (GraphPad Software 8.4.3). Details on statistical tests used and statistical significance are included in the relative figure legend. For comparison between samples with normal distribution, a two-tailed unpaired Student's *t*-test was used, when comparing two groups. For comparison of more than two groups one-way or two-way ANOVA was used, followed by Tukey's or Bonferroni's test, for the post-hoc analysis. For not normally distributed samples, Mann–Whitney U test was used in order to compare two groups. Survival plots were analyzed using the nonparametric log-rank (Mantel-Cox) test. *P*-values <0.05 were considered statistically significant.

**Reporting summary**. Further information on research design is available in the Nature Research Reporting Summary linked to this article.

## Data availability

All data generated or analysed during this study are included in this published article, its supplementary files and accompanying source data files. The raw data of the proteomic experiments have been deposited in the PRIDE repository (Project accession: PXD030079). Source data are provided with this paper.

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

## Acknowledgements

We thank Leonardo Restivo (Neurobau behavioral facility of the University of Lausanne) for support with behavioral experiments, and Irina Kolotueva (electron microscopy facility of the University of Lausanne) for support in electron microscopy. We also thank Prof. Manuel Mameli for critical reading of the manuscript and Dr. Nikolaos Lempesis for support with bioinformatics analyses. We also thank SciDraw for the schematic in Fig. 7d (doi.org/10.5281/zenodo.3925903, author Luigi Pertucco). This work was supported by an ERC starting grant (NEUROPHAGY) to V.N. and an ERC advanced grant to D.C. (DynSynMem (787340)). The work in C.B. lab was supported by the SNSF NCCR Synapsy 51NF40-158776 and SNSF 310030-182651 grants.

## Author contributions

V.N., E.K., and A.D.D. conceived and designed the study. A.K. performed all electrophysiology experiments. K.S. provided intial support for electrophysiology experiments. Y.D. performed electron microscopy experiments. M.M.S., P.H., and F.S. performed the proteomic analyses and F.S. performed the biostatistical analyses of the proteomics. C.C., E.H., and D.C. performed and analyzed the super-resolution experiments. V.M. performed the stereotaxic injections and C.B. provided input on the manuscript. E.K., E.I., and A.D.D. carried out all other experiments and analyzed data. V.N., E.K., and A.D.D. wrote the manuscript.

## Competing interests

The authors declare no competing interests.
