## [Peer Review File · Nature Communications]

Reviewers' Comments:

Reviewer #1:

Remarks to the Author:

Daskalaki, et al. investigates the role of macroautophagy in long-term depression like mechanism, a developmental process necessary for developmental pruning of dendritic spines. Autophagy on pre-synaptic side has been previously studied, however its role in post synaptic side processes are less known. This paper aims to address if autophagy is required for degradation of post synaptic components, thereby attempting to characterise long term depression (LTD) mechanism in dendrites of cultured hippocampal neurons, hippocampal slices and conditional knockout mice. The study mainly focuses on NMDAR and mGluR-LTD. Induction of LTD was done by treating cultured neurons with NMDA or DHPG or low frequency stimulation (LFS) of Shaffer collateral fibers in the hippocampus. The authors report that induction of LTD increases LC3 positive autophagic vesicles (AV) which colocalises with other autophagy markers such as WIPI2, ULK1, Atg101, Atg13 and FIP200, indicating AVs are locally formed in post synaptic dendrites following LTD. NMDA, DHPG or LFS induced LTD in hippocampal slices showed similar results as observed in cultured neurons. Induction of autophagy was not possible when cells were treated with inhibitors for NMDAR and mGluR, signifying the importance of LTD induction on AV formation. Similarly, cells treated with inhibitors for ULK1 inhibited GluA2 internalisation and both forms of LTD, thereby suggesting a role of autophagy in LTD. Proteomic profiling of purified hippocampal AVs upon LTD induction showed an enriched population of post synaptic proteins including the AMPA receptor complex and scaffold proteins important for receptor internalisation. Conditionally Atg5-ablated mice further showed decreased LTD implicating that autophagy is indeed an essential mechanism for regulating LTD. Mice having a reduced Atg5 levels showed deficit in flexible choice behaviour which is a LTD mediated behaviour. Overall, the work of Daskalaki and colleagues provide evidence that LTD requires post-synaptic autophagy of surface receptors in dendrites. However, several concern remain.

Major points

- 1) Figure 1c: The authors should examine whether the observed increase in LC3-positive structures are dependent on ATG7 or ATG5 and hVPS34.
- 2) Figure 3: The authors should provide evidence that the ULK1 complex components actually colocalize with WIPI2 or LC3B.
- 3) Are the identified post-synaptic autophagy cargo candidates GluA2 and PSD95 actually delivered to lysosomes? The authors should address this by probing for the colocalization of both proteins with LAMP1/2.
- 4) The authors should show the inhibitory effect of SBI-0206965 by monitoring the abundance or subcellular distribution of autophagy cargo such as LC3 or p62.
- 5) Does PSD95 localize to LC3 conjugated to single or double endomembrane?
- 6) The authors should monitor the abundance of GluA2 and PSD95 upon ULK1 inhibition. Also, the authors should check for the effect of hVps34 blockage in this setting.
- 7) The flow of the manuscript connotes that the AV preparation is on post-synaptic autophagic vesicles. The authors should clearly state that this is actually not the case. Moreover, it is not clear how the authors control for co-purifying proteins in this this preparation? Further, the authors should highlight known autophagy cargo proteins (e.g. p62) found in their preparation. It is also worth comparing their candidates to other autophagosomal repertoires recorded using similar approaches (e.g. Mancias et al. Nature 2014).
- 8) Figure 5e/f is missing a loading control to which the levels of cargo candidates are normalized to.
- 9) The authors should probe whether the protease protection of their candidates (or at least some

of them) depends on ULK1 and hVps34 inhibition.

10) Since a previous study this year showed autophagy inhibition is necessary for LTD induction (Hongmei Shen, et al, Nat comm, June 2020), the authors should discuss this discrepancy with their own findings in more detail.

Minor points

11) Introduction – 3rd paragraph first line can be reframed for better understanding.

12) Fig 1d is missing the untreated control for comparison.

13) Fig 2a: The figure labelling control or NMDA/DHPG treatment missing. In Fig 2e, the scale bar scale is missing.

14) Fig 3f: The labels are not clear as to which one is 15 min or 1hr after induction. The colour legend of each asterisks should be mentioned in figure legend. Fig legend 3g should be reframed.

15) Fig 4a: The staining Ab (GluA2) should be mentioned on the image. Fig 4f is missing a control panel for comparison. Fig 4g: The y-axis label should indicate it is colocalization %. Fig legend 4i and 4j indicated (f) but it should be (h)

16) Page 7, paragraph 2, last line it should be Fig 3g and not 2g.

17) The authors should include the data that they refer to as “data not shown” in the supplements.

Reviewer #2:

Remarks to the Author:

Overall assessment

The authors present evidence to suggest a novel role for the autophagy protein degradative pathway in the induction of synaptic depression in the mouse hippocampus. Using neuronal cultures and hippocampus slices, the authors demonstrate that markers of autophagy are increased by chemical treatments and electrical stimulation patterns that induce LTD through both NMDARs and mGluRs. LTD induction increased the amount of autophagosomal structures in neuronal dendrites as well as the amount of synaptic proteins within isolated autophagosomes. Further, pharmacological and genetic inhibition of autophagy was found to prevent LTD induction in acute hippocampal slices. Finally, partially inhibiting autophagy throughout the brain was found to impair cognitive flexibility.

The overall findings of the paper are of potential interest to a wide readership, as it is relevant for studies of synaptic function, synaptic plasticity, cognition and the autophagy field as well. However, there are a number of concerns the authors must address, in particular the quality of the electrophysiology, in order to strengthen the findings and conclusions of the study.

General concerns

- The overall low quality of the electrophysiology throughout the manuscript makes it difficult to draw conclusions from these experiments for the following reasons:
 - o Many of the sample traces displayed indicate poor slice health, as the fiber volley in some traces (e.g. Figure 4C) is larger than the EPSP response
 - o The sample traces also all look quite distinct from each other, indicating highly variable electrode positioning and slice quality. In addition to large differences in fEPSP size between experiments, the waveform varies between experiments (some traces the FV is distinct, others it is not discernable) and in other cases the waveform changes from baseline to post-LTD induction (e.g. Fig. S4C)
 - o The baseline in many experiments is highly unstable and too short to accurately determine if the recordings are stable before LTD induction. A 20-30 minute baseline with minimal drift and variability is needed

- o The shape of the LTD induction curves are not consistent with the literature for many experiments. Following LFS, there should be a greater degree of depression which decays to a stable LTD over the first ~30 minutes post-LFS, which was not observed for most LFS experiments. Further, NMDA application, especially when given for 10 minutes (which is longer than the accepted standard in the literature of 3-5 minutes) should induce a robust depression of the fEPSP to nearly 0, as opposed to the slow decline observed by the authors. A similar issue occurs for DHPG-LTD as well, which should have a larger acute phase of LTD during application.
- o The fEPSP peak amplitude is quantified not the fEPSP slope. As changes in the peak amplitude can reflect alterations to not only synaptic AMPARs, but also altered inhibition and/or voltage-gated ion conductances, the fEPSP slope needs to be quantified instead which more accurately reflects only AMPAR currents
- o The LFS protocol used by the authors to induce LTD for electrophysiology experiments was 1200 pulses at 1.4 Hz. Why was this protocol used instead of the standard 900 pulses at 1 Hz? Further, given this different protocol the authors should determine whether this protocol induces an NMDAR-dependent only LTD or if mGluRs are activated as well
 - A recent publication by Shen et al., (2020, Nature Communications) also examined the role of autophagy in hippocampal long-term depression and cognitive behaviours. Given the overlap in focus and opposing findings, the authors need to cite and discuss this paper in their manuscript
 - For many of the example western blot images shown, the loading controls were not displayed, which is critical for assessing the band density for the protein of interest. Further, to enhance clarity, the order of the treatment conditions should be the same in the example images as in the graph
 - In the ICC and IHC experiments examining the increase in autophagy markers following NMDA or DHPG treatment, is there a negative control region, such as the soma where LTD is not expected to occur to determine if autophagic markers are generally increased throughout the cell or only in synaptic regions where LTD is occurring?

Specific concerns:

Figure 1

- Why did the authors choose to analyze only the 15-minute time point? It is important to understand whether autophagy is only activated transiently during the induction phase of LTD or continuously upregulated throughout the maintenance phase. For two of the key proteins examined, LC3 and ULK1, the authors should perform a time-course experiment to analyze changes at 5, 15, 30, 45 and 60 minutes following LTD induction.
- Ifenprodil and MTEP/JNJ were found to inhibit the increase of LC3-II following NMDA and DHPG application, respectively. However, the authors should confirm that these treatments do indeed block the decrease in surface GluA2
- In the Methods section the number of days in vitro of the neurons in culture used for these and any other experiments using cultured neurons needs to be indicated

Figure 2

- In the Methods it is stated that rat hippocampal neurons seeded on astrocytes were used for the dSTORM imaging. Why were these neurons/conditions used only for these experiments?
- For the dSTORM example images, it would help if the authors clarified on the example images themselves i) what condition was being shown (i.e. baseline, or post-stimulation) and what the U-shape structure was that was being quantified
- Given the 4 proteins examined independently in 4E are hypothesized to be part of the same structure, the authors should examine whether the puncta for these proteins co-localize following NMDA/DHPG stimulation which would provide support for this notion
- Using ICC, Atg101 was found to be drastically increased following DHPG treatment, while in the Western blot data there is no change. Why is there such a discrepancy between assays? Further, the representative images of the blots for Atg101 seem very faint. Were the antibody/conditions used to detect Atg101 in Western blot sufficiently sensitive?
- The band for LC3-II is much closer to 17 kDa in 2F, while elsewhere in the manuscript the LC3-II band is closer to 10 kDa, which is consistent with the literature. Is the band being quantified in this figure indeed the right size for LC3-II?
- As LC3-II is a marker of intermediate structures in autophagy, it has been found that increased LC3-II can indicate either increased or decreased autophagic flux (Mizushima and Yoshimori, 2007). Therefore, the authors should use known activators and inhibitors of autophagy in their

experimental conditions to determine if increase LC3-II is indeed an indicator of enhanced autophagy

Figure 3

- In acute brain slices, the cut surface of the slices (up to ~50 microns) is damaged. When the authors performed immunohistochemistry imaging from slices (3A-E), did they ensure the imaged were coming from greater than 50 microns below the surface of the slice?
- In the example images of 3A-E, it can be seen that some of the autophagy markers are increased in regions outside of MAP2+ dendrites (e.g. Atg13). Were the quantified puncta in the Figure only those that were co-localized with MAP2 to ensure the signal measured was from principal neurons?
- For the experiments in 3F-I, it is noted in the text that the control condition is that neurobiotin labelling was not performed, and so AVs in dendrites cannot be quantified as there are no labelled dendrites to being with. However, there is no control condition in which dendrites are labelled but LTD is not induced. As such, the authors should include a new condition in which LFS is delivered in the presence of AP5 to block LTD induction but control for any effects of the increased stimulation/neuronal activity due to the LFS itself.
- In addition, for the experiments in 3F-I, how did the authors determine which spines were stimulated and which were not? Given the neurobiotin was applied generally to the area of recording, the spines that would be labelled will be a mixed population of both stimulated and non-stimulated spines (the latter being the predominant type).
- Finally, for the quantification in 3G-I, the N value is the number of dendrites imaged. How many slices from how many animals did these dendrites come from?

Figure 4

- In the sample traces from the SBI-treated group in 4C, the fiber volley is substantially increased from baseline to post-LFS. The authors should quantify the fiber volley amplitude for this experiment in order to determine to what degree it increases after LFS and if this occurs only in the SBI-treated group.
- Why was SBI-0206965 used to inhibit autophagy in the electrophysiology experiments while bafilomycin was used for the same purpose in biochemistry experiments? Preferably, given the off-target effects of each drug, both inhibitors would be used for both electrophysiology and biochemistry experiments.
- The authors should test whether pharmacological activation of autophagy is sufficient to induce LTD to strengthen their conclusions of the causal role of autophagy in LTD
- In 4I and J, the authors use a one-way ANOVA. Given there are 2 independent variables (NMDA/DHPG and +/- baf.) a 2-way ANOVA is the appropriate statistical analysis
- Generally, in the literature, chemical LTD, especially in the early phases, is associated with changes to GluA phosphorylation and reduced surface expression, but not changes in total protein of GluA or PSD-95. Why is such a large decrease in total PSD-95 and GluA2 protein levels observed rapidly following LTD in the author's conditions?
- Is the quantification of GluA2 and PSD-95 in 4I and J normalized to beta-actin loading control?

Figure 5

- In 5A, why is the amount of LC3-II roughly equivalent in the nuclei fraction as the AV enriched fraction? Would it not be expected that the AV enriched fraction should have substantially more LC3-II?
- Does the method for isolating AVs used in this Figure enrich for those in pyramidal cells specifically, as autophagy occurs in all cell types?
- For 5B, in addition to the close-up images of putative AVs, a lower zoom version of the image should be shown so that the number of these structures relative to everything else contained in the AV fraction can be assessed
- In 5E, how is it known that the number of AVs loaded in each lane is equivalent? The authors should include a loading control (e.g. LC3-II) and normalize the protein values in each lane to LC3-II to ensure that the increase in synaptic proteins is not due to an increase in the number of isolated AVs
- The authors mention multiple times in the text that under basal conditions, autophagy is restricted in neurons to the axon tip. If this is the case, why are post-synaptic proteins observed in AVs under control conditions, in some cases to the same level as in the PSD fraction (e.g. ITPKA,

SAP97)?

Figure 6

- For the Atg5 conditional KO mice, were the control mice (CreER negative) given tamoxifen as well to control for any effects of tamoxifen administration?
- LFS in Thy1-cKO mice induced a small potentiation effect. It would be interesting to know if LTP is also enhanced in these mice or if autophagy only plays a role in LTD
- Basal synaptic properties (I/O and PPR curve) need to be assessed in the cKO and shRNA mice
- LFS-LTD is well known to be much more robustly induced in younger mice (>P28) and for many labs not possible to induce in adult mice. Why was the amount of LFS-LTD actually higher in the control adult mice in the shRNA experiments (5G) compared to young control mice in 5A?
- The authors state that the justification for performing the shRNA knockdown experiments is to examine the role of autophagy in postsynaptic CA1 neurons only. Given there is a commercially available mouse line (CAMKII-Cre, T29-1) which has CA1-restricted Cre expression that onsets at ~P17, why was the shRNA method chosen instead?
- What is the knockdown efficiency of the shRNA on Atg5 levels themselves?

Figure 7

- While the justification of using Atg5^{flx}/WT mice to only partially inhibit autophagy to avoid any effects of full knockout is understandable, why a different Cre-driver line was used compared to Figure 6 is unclear. The same experiments could have been performed using the Thy1-CreER mouse that had been already shown to have an LTD deficit
- Electrophysiology experiments were not performed in the nestin-cHET mice and thus it is not known if there even is an LTD deficit in these mice
- Further, as nestin-Cre mice express Cre in all neurons, there is no way to link changes to LTD in CA1 post-synaptic pyramidal neurons to the observed behavioural phenotype
- The authors should perform the following experiments to help clarify these issues;
 - o At a minimum use the Thy1-CreER mice instead of Nestin-Cre, but ideally using the CAMKII-Cre mice crossed with Atg5-floxed mice
 - o In both control (Cre negative) and cKO mice, administer the TAT-GluA23Y peptide, which has been shown to block AMPAR endocytosis induced by LTD (Amadian et al., 2004; Brebner et al., 2005; Fox et al., 2007; Dong et al., 2013)
 - o The expected result is that TAT-GluA23Y in control mice will induce a cognitive flexibility deficit similar to the vehicle-treated Atg5 cKO, and that administration of TAT-GluA23Y in these cKO mice will not cause a further deficit
 - o These experiments will better delineate i) the role of autophagy specifically in CA1 neurons in cognitive flexibility and ii) if any changes observed are due to blockade of LTD (as indicated by an occlusion of the effect of a peptide already known to inhibit LTD)

Reviewer #3:

Remarks to the Author:

Kallergi et al in this Ms deal with the possible role of autophagy in the turnover of postsynaptic proteins during LTD. They show that induction of chemical LTD (cLTD) by either NMDAR or mGluR stimulation elicits an elevation of autophagic proteins and vacuoles in dendrites and at postsynaptic elements in cultured neurons and in acute hippocampal slices line with previous work on the induction of autophagy by cLTD (Shehata et al J Neurosci 2012). Application of SBI-0206956, an inhibitor of the autophagic ULK1 kinase, is further shown to block cLTD and GluA2 endocytosis and leads to increased localization of PSD95 to LC3 positive puncta in neurons. These data correlate with the presence of postsynaptic proteins in AV-enriched fractions from brain. Finally, low-frequency stimulation-induced LTD is shown to be occluded upon knockdown of the autophagy protein Atg5, while heterozygous loss of Atg5 in neurons in mice leads to impaired odor reversal learning. From these data the authors propose a model according to which autophagic turnover of postsynaptic receptors and associated PSD proteins is required for LTD.

Overall, though I find the study interesting in principle, at this stage I remain unconvinced that the observed loss of LTD in slices from Atg5-cKOs indeed relates to defective endocytic internalization and degradation of AMPARs and associated PSD proteins.

Specific comments:

1. A major conceptual problem is the interplay between autophagy and endocytosis. In Fig 4a,b it is shown that inhibition of ULK1 by SBI-0206956 not only causes a shift from LTD to LTP but also abrogates GluA2 endocytosis, in fact more potently than peptide-mediated dynamin inhibition. This is puzzling as dozens of studies on autophagy have failed to detect a "retrograde" action of failed autophagy on endocytosis, while conversely, some studies suggest that endocytosis or at least endocytic proteins can mediate early steps of autophagy. These findings in my view cast doubt on the specificity of the drug and/ or the role of ULK1 as a regulator of autophagy as opposed to other possible functions. In my view these experiments need to be repeated under conditions of specific genetic ablation of autophagy. The cKO-Atg5 data shown in Fig6 suggest that at least the physiological effects may be related, however the underlying mechanism remains enigmatic in my view.

2. ULK1 inhibition via SBI-0206956 greatly reduces fEPSPs in the traces shown in Fig4c, an effect the authors surprisingly seem to have ignored. How is this depression of basal transmission explained that also clashes with recent observations in Atg5 cKO slices, in which increased excitatory transmission was observed (Kuijpers et al Neuron 2020) and with the authors' own data shown in Fig 6? I again wonder about the specificity of the SBI-0206956 effect and phenotype.

3. In Fig5 the authors present proteomic data to identify the autophagic cargos targeted during LTD. In my view these data are too preliminary to allow any firm conclusions. I am not impressed by what the authors refer to as "enrichment of LC3". In fact, in the WB shown in panel a there is no enrichment of LC3 in the alleged AVs. I suspect that the isolated material, in addition to AVs shown in the EM images in panel b, comprises a number of contaminants that may arise from the biochemical fractionation procedure. The isolated AVs, hence, need to be compared to material isolated from Atg5 KO animals and only then, it can be concluded that the identified proteins likely represent autophagic cargo. Alternatively, correlative super-resolution LM/ EM analysis could be used to show that the AVs seen in the EM indeed contain the alleged cargo proteins.

4. A prediction from the hypothesis that autophagy and subsequent lysosomal turnover play a major role in LTD is that autophagic cargos accumulate in the conditional absence of Atg5 upon LTD induction in slices or neurons from genetically altered mice. For example, one would expect GluA1 > GluA2 as well as CaMKII and PICK1 to display prolonged half-lives under such conditions. Can this be observed? In fact, the title of the paper claims to show that dendritic autophagy degrades postsynaptic proteins, while this bold claim in my view is not or only insufficiently supported by the data.

5. I wonder whether U-shaped phagophores can be reliably distinguished from autophagosomes by STORM microscopy. Under-sampling of single molecules make the reconstruction of entire organelles and their shapes difficult. Moreover, it is unclear to me how the amount of LC3 not associated with phagophores shown in Fig S2a was quantified. These data also seem to clash with the EM morphometry shown in Fig. 3f+h, in which an increase in complete double membrane autophagic vesicles (i.e. non-phagophores!) is observed. From the STORM data one would expect an accumulation of incomplete U-shaped early structures, which, however does not seem to be observed at the EM level.

6. I was surprised to see that for the in vitro experiments neurons isolated from E15 mice were cultured for only 6-7d, a time frame during which hardly any synapses form. Is this a typo? Typically, hippocampal neurons are cultured for at least 12-15d to display functional synapses and synaptic responses. Also, it is unclear from the results, methods, and legends whether hippocampal or cortical neurons were used for the respective experiment.

Minor points:

7. A standard test used to assess the role of LTD at the behavioral level is spatial reversal learning, e.g. using the Morris water maze test. Do Atg5 Nestin-cHet display defects in this paradigm?

8. Do Atg5 nestin-cHet mice show defects in LTD similar to thy1-cKO?

9. Fig 1: The authors state that LC3 density decreases with increasing distance from the soma. Where were the ROIs taken in Fig 1C? I guess the total dendrite is measured -right? Are all dendrites from one single neuron taken or just the apical dendrites? This should be specified in the text.

10. Fig S1A/B: Why does BafA in control conditions not result in increased LC3-II? From the WB it looks as if in neurons subjected to LTD, LC3-II is already increased in the absence of BafA (as should be the case according to Fig 1C).

11. Figure 2:

2A: It is not clear to me what we actually see here. What do the different panels and colors represent? Furthermore, alleged autophagosomes and phagophores should be marked (but see my comment #5 above). The permeabilization method is not specified in the methods section.

2E: Images do not seem to be representative for the quantification (I see many more puncta in the images compared to the quantification).

2F: LC3 quantifications are missing.

12. Fig 3F: Please mark the vesicle-like structures inside spines and specify what the red arrow and yellow asterisk refer to in the figure legend.

13. Fig 3H: The authors use ciliobrevin to inhibit transport of somatic autophagosomes into dendrites and thereby provide evidence that autophagosomes are produced locally. However, as also indicated by the authors, ciliobrevin would likely inhibit trafficking of autophagosomes to the soma and possibly autophagosome biogenesis. It is therefore difficult to interpret these changes in autophagosomes number, especially as the control condition (+ CBD) is missing. Additional experiments to assess LC3-mRFP puncta dynamics/appearance upon LTD induction could provide additional support for the claim that autophagosomes are produced locally in dendrites.

14. Figs. 4C, S4: What do the red and black traces represent (\pm LFS)? Please specify in the legend.

Reviewer #1 (Remarks to the Author):

Daskalaki, et al. investigates the role of macroautophagy in long-term depression like mechanism, a developmental process necessary for developmental pruning of dendritic spines. Autophagy on pre-synaptic side has been previously studied, however its role in post synaptic side processes are less known. This paper aims to address if autophagy is required for degradation of post synaptic components, thereby attempting to characterise long term depression (LTD) mechanism in dendrites of cultured hippocampal neurons, hippocampal slices and conditional knockout mice. The study mainly focuses on NMDAR and mGluR-LTD. Induction of LTD was done by treating cultured neurons with NMDA or DHPG or low frequency stimulation (LFS) of Shaffer collateral fibers in the hippocampus. The authors report that induction of LTD increases LC3B positive autophagic vesicles (AV) which colocalises with other autophagy markers such as WIPI2, ULK1, Atg101, Atg13 and FIP200, indicating AVs are locally formed in post synaptic dendrites following LTD. NMDA, DHPG or LFS induced LTD in hippocampal slices showed similar results as observed in cultured neurons. Induction of autophagy was not possible when cells were treated with inhibitors for NMDAR and mGluR, signifying the importance of LTD induction on AV formation. Similarly, cells treated with inhibitors for ULK1 inhibited GluA2 internalisation and both forms of LTD, thereby suggesting a role of autophagy in LTD. Proteomic profiling of purified hippocampal AVs upon LTD induction showed an enriched population of post synaptic proteins including the AMPA receptor complex and scaffold proteins important for receptor internalisation. Conditionally Atg5-ablated mice further showed decreased LTD implicating that autophagy is indeed an essential mechanism for regulating LTD. Mice having a reduced Atg5 levels showed deficit in flexible choice behaviour which is a LTD mediated behaviour. Overall, the work of Daskalaki and colleagues provide evidence that LTD requires post-synaptic autophagy of surface receptors in dendrites. However, several concern remain.

We thank this Reviewer for his/her constructive comments. We have performed most of the suggested experiments and the new results have further strengthened our findings. Below is a point-by-point response.

Major points

1) Figure 1c: The authors should examine whether the observed increase in LC3B-positive structures are dependent on ATG7 or ATG5 and hVPS34.

These are two important points that we have experimentally addressed, as detailed below.

A) **Atg5-dependence:** In order to ensure that the LC3B-positive structures are Atg5-dependent, we knocked down endogenous *atg5* in cultured neurons. This was achieved by infecting neuronal cultures at div6 with an AAV virus expressing 4 shRNA sequences against *atg5* (*sh-atg5*), under the CamK2a promoter. As a control, neuronal cultures were infected with the same virus expressing 4 scrambled shRNA sequences (*sh-scramble*). Analyses were performed 10 days post-infection. As shown in **Figure S1d**, *sh-atg5* was effective in reducing the protein levels of Atg5 and of LC3B-II (**Figure S1e**). As shown in **Figure 1d**, chemical NMDAR- and mGluR-LTD triggered a significant

increase in dendritic LC3B-positive structures in *sh-scramble* expressing neurons, but this was largely abrogated in *sh-atg5* expressing neurons.

These findings thus indicate that the rapid appearance of LC3B-positive structures in dendrites following LTD are Atg5-dependent.

B) Vps34-dependence: To test whether the LC3B structures are Vps34-dependent, we pharmacologically blocked Vps34, which is a catalytic subunit of the PI3K complex, by Wortmannin. Wortmannin has been widely used in the field to block autophagy initiation. **Figure S1c** shows its effectiveness in our neuronal cultures, as treatment of neurons with 500nM Wortmannin for one hour significantly decreased the levels of LC3B-II. As shown in **Figure 1c**, Wortmannin treatment completely abolished the appearance of dendritic LC3B-positive structures after chemical LTD. Similar results were obtained after treatment with SBI-0206965, a selective inhibitor of ULK1 kinase (discussed later, in point 4). Taken together, these results indicate that dendritic LC3B-positive structures require Vps34 and the autophagy initiation machinery.

2) Figure 3: The authors should provide evidence that the ULK1 complex components actually colocalize with WIPI2 or LC3B.

We tried to address this point, however, the antibodies that work well in our hands for the ULK1-complex components and for LC3B are all raised in rabbit and therefore they cannot be combined. Moreover, recent work suggests that LC3B may not interact with the ULK1 complex, at least in non-neuronal cells¹. In co-localization experiments with WIPI2, only a small fraction of approximately 30% of each ULK1-complex component co-localized with WIPI2. As the early steps of initiation are highly dynamic and very rapid, we cannot easily interpret these data.

3) Are the identified post-synaptic autophagy cargo candidates GluA2 and PSD95 actually delivered to lysosomes? The authors should address this by probing for the colocalization of both proteins with LAMP1/2.

This is also an important point. We have not used LAMP1/2 because in our experience, and in line with published findings^{2,3}, in neurons these markers are not reliable for labeling degradative compartments. Instead, to address this point, we have used BafilomycinA1, a drug that inhibits the V1-ATPase proton pump and prevents the acidification of lysosomes. As shown in **Figure 4d**, the decrease of PSD-95 and of GluA2 levels after NMDAR- and mGluR-LTD is completely prevented in the presence of BafilomycinA1, indicating that their degradation happens at the lysosome. To ensure that their lysosomal degradation requires macroautophagy, we tested their levels in control conditions and after chemical LTD in neurons treated with SBI-0206965 to block the initiation of AV biogenesis. As shown in **Figure 4e**, this treatment also prevented the decrease in PSD-95 and GluA2 levels after both NMDAR- and mGluR-LTD. Finally yet importantly, we also examined their levels after knock-down of *atg5*. As shown in **Figure 4f**, both PSD-95 and GluA2 levels failed to decrease after LTD in *sh-atg5* expressing neurons, as compared to *sh-scramble* expressing controls. Therefore, these results demonstrate that the reduction of PSD-95 and GluA2 levels after NMDAR- and mGluR-LTD requires the autophagy-lysosome system.

4) The authors should show the inhibitory effect of SBI-0206965 by monitoring the abundance or subcellular distribution of autophagy cargo such as LC3B or p62.

We have now shown that SBI-0206965 treatment significantly reduces the levels of LC3B-II in cultured neurons. These findings are in **Figure S1c**. Similarly, dendritic LC3B-positive structures failed to increase after NMDAR- or mGluR-LTD in neurons treated with SBI-0206965. These results are now in **Figure 1c**. Please see the corresponding panels in response to point 1B.

5) Does PSD95 localize to LC3B conjugated to single or double endomembrane?

The recent findings that LC3B can associate with single endomembranes (in non-neuronal cells) have clearly indicated that this process is totally independent of the AV biogenesis machinery⁴⁻⁷. Therefore, the fact that dendritic LC3B-positive structures can not be induced after chemical LTD in neurons treated with Wortmannin or with SBI-0206569 (which block two different steps of the initiation machinery), strongly suggests that these structures are not related to endomembranes. Moreover, we performed super-resolution experiments after labeling neurons with LC3B and the early endosome marker EEA1⁸. Labeling was performed 15 minutes after an NMDA or DHPG pulse. As shown in the **Extra data_R1_point 5** panel, we found no co-localization of LC3B-positive U-shaped dendritic structures with EEA1, neither in the NMDA nor in the DHPG condition.

6) The authors should monitor the abundance of GluA2 and PSD95 upon ULK1 inhibition. Also, the authors should check for the effect of hVps34 blockage in this setting.

Please see our response to point 3, which also includes the answer to this point, namely the effect of ULK1 inhibition on GluA2 and PSD-95 levels after LTD. We have also examined the effects of Wortmannin on the internalization of GluA2 and found that similar to SBI-0206965 and to *sh-atg5*, it also prevents its removal from the membrane. Representative images are shown here (**Extra data_R1_6**). We have not included these findings in the manuscript, due to space limitation.

7) The flow of the manuscript connotes that the AV preparation is on post-synaptic autophagic vesicles. The authors should clearly state that this is actually not the case. Moreover, it is not clear how the authors control for co-purifying proteins in this this preparation? Further, the authors should highlight known autophagy cargo proteins (e.g. p62) found in their preparation. It is also worth comparing their candidates to other autophagosomal repertoires recorded using similar approaches (e.g. Mancias et al. Nature 2014).

We have paid attention to state more clearly that AVs are purified from hippocampal slices, under control conditions or after NMDAR-LTD.

We have addressed the purity of our AV preparation in the following ways:

A) We tried to purify AVs from nestin-cKO hippocampal slices, following the exact same procedure, but no measurable material was obtained. This suggests that the preparation mainly consists of AVs and contamination with other structures is minimal.

B) We used magnetic beads to further immuno-purify autophagic vesicles from the AV preparation with an antibody against LC3B. An IgG was used as a control. As shown in **Figure S5e**, the majority of the AV preparation (input, lane 1) was immunoprecipitated with the LC3B Ab (lane 2), as can be seen from the top panel of the total protein stain. By contrast, almost no material was precipitated by the control IgG (lane 3). This was also reflected by the amount of proteins that was left in the unbound (flow-through) material. In the case of the LC3B-IP, after centrifugation, the pelleted unbound material very small (arrowhead in Eppendorf picture insert for lane 4) and contained a very small fraction of the input (lane 4, protein stain). We could still detect a very small amount of LC3B in this unbound material (LC3B blot, lane 4), suggesting that the IP was highly but not 100% effective in precipitating AVs. In the case of the IgG-IP, after centrifugation, the unbound material formed a big pellet (arrowhead in Eppendorf-tube picture insert for lane 5) and its protein and LC3B content were similar to the input (lane 5). Importantly, the levels of PSD-95 were reduced in the unbound of the LC3B-IP (lane 4) compared to the unbound of IgG-IP (lane 5), indicating that this synaptic protein is not a contaminant but is precipitated in the LC3B-positive fraction. These findings are consistent with our previous work^{9,10}.

We want to point out that p62 is shown in **Figure 5g** and has also been used to normalize the levels of other cargo proteins.

As suggested by the Reviewer, we have compared our hippocampal-slice AV cargo to that of Mancias et al., 2014¹¹, where autophagic cargo was proteomically profiled in cultured cell lines. As shown in the Venn diagram in **Figure S5c**, we found that 69% of the hippocampal slice AV cargo is common with the Mancias et al., cargo, while the remaining 31% was specific to the hippocampal slice. This comparison is also summarized in the new **Table S1**. We also performed cellular component enrichment analysis for the common and the unique cargo. The common cargo was mainly composed of proteins involved in mitochondrial respiration, ER to Golgi transport and translation machinery, all processes that are housekeeping and common among different cell types. By contrast, the unique cargo contains synaptic proteins, which are involved in both pre- and post-synaptic functions and excitability.

8) Figure 5e/f is missing a loading control to which the levels of cargo candidates are normalized to. The blot has been normalized to total protein levels. We have clearly indicated this in the figure legend now.

9) The authors should probe whether the protease protection of their candidates (or at least some of them) depends on ULK1 and hVps34 inhibition.

It is not clear to us what the reviewer means by this comment. The protection assays of the synaptic cargo proteins were performed in AVs purified from hippocampal slices (control and LTD conditions). Technically, it's not easy to add treatments to this workflow, as each purification requires pulling together the hippocampal slices of several animals. Therefore, doing these treatments in triplicates would require a large number of mice. Moreover, as ULK1 and Vps34 inhibition would interfere with AV biogenesis, we would expect a reduced yield of AVs for analysis.

10) Since a previous study this year showed autophagy inhibition is necessary for LTD induction (Hongmei Shen, et al, Nat comm, June 2020), the authors should discuss this discrepancy with their own findings in more detail.

As the Reviewer indicates, our results contradict the findings of a recent study¹², showing that autophagy inhibition is both necessary and sufficient for LTD induction. We believe that the differences largely originate from using different tools and different developmental windows for autophagy ablation.

A) Shen et al., use the T29-1 mice, *Atg5F/F;CA1-Cre+*, in order to ablate *atg5* specifically in the CA1 area. There are, in our view, two problems with this approach:

- As described in the original publication that characterized these Cre deleters (also known as T29-1)¹³, Cre expression starts already in the third post-natal week, namely at the onset of developmental spine pruning, that occurs between P20 and P30 in the hippocampus. Previous work has clearly demonstrated that autophagy is required for developmental spine pruning¹⁴, and that ablation of *atg5* under the *CamK2a* promoter results in a significant increase in the number of spines because of failed pruning. Therefore, pruning is very likely impaired in this mouse, which can surely interfere with the responses to LFS stimulation. We consider this a limitation in the Shen et al., paper. Our approach circumvents this problem by knocking down *atg5* in the adult CA1 area, long after the end of pruning.
- Shen et al., have tried to induce LTD in adult hippocampal slices (from control and *Atg5F/F;CA1-Cre+* mice), using a protocol of 900 pulses at 1Hz. While this protocol works in young animals, it is well known in the field that it cannot maintain LTD in adult slices. As shown in Figure 10a of their paper, the response of their control slices returns to baseline within 20 minutes of stimulation (producing a short depression but not an LTD). Therefore, it is difficult to compare genotypes when the control does not respond with a sustained LTD. In our study we use a protocol (3 x 1200 pulses at 2Hz, with an inter-stimulus interval of 10 minutes)¹⁵ that works well in inducing a sustained LTD in the adult.

B) Shen and colleagues use rapamycin and trehalose as autophagy inducers. Rapamycin inhibits the TOR pathway and by doing so it affects protein synthesis, not only autophagy. Trehalose is a very pleiotropic drug and its effects as an autophagy inducer are strongly debated^{16,17}.

However, it is also possible that the discrepancy actually reveals important differences in the interplay between autophagy and synaptic plasticity during different developmental windows, as Shen and colleagues performed most experiments at an earlier time-point, between P16-19. We have discussed these differences in our revised manuscript.

Minor points

11) Introduction – 3rd paragraph first line can be reframed for better understanding.

Thank you, we have rephrased it.

12) Fig 1d is missing the untreated control for comparison.

The comparison here is between LTD conditions, before and after blocking the relevant glutamate receptors.

13) Fig 2a: The figure labelling control or NMDA/DHPG treatment missing. In Fig 2e, the scale bar scale is missing.

Thank you, we have added the labeling and the missing scale bar.

14) Fig 3f: The labels are not clear as to which one is 15 min or 1hr after induction. The colour legend of each asterisks should be mentioned in figure legend. Fig legend 3g should be reframed.

We have now clearly marked the 15 and 1hour time-point on the figure. We have also mentioned the color of asterisks and what they show in the figure legend.

15) Fig 4a: The staining Ab (GluA2) should be mentioned on the image. Fig 4f is missing a control panel for comparison.

Fig 4g: The y-axis label should indicate it is colocalization %. Fig legend 4i and 4j indicated (f) but it should be (h)

4a: We have included the GluA2 surface labeling.

4g: We have changed the y-axis labeling.

4f: As LC3B-positive structures are largely absent from dendrites under control conditions, we have examined the co-localization of PSD-95/LC3B only under LTD conditions.

16) Page 7, paragraph 2, last line it should be Fig 3g and not 2g.

We have proof read the revised manuscript and hope that there are no more typographical mistakes.

17) The authors should include the data that they refer to as “data not shown” in the supplements.

We have now included the data in Figure S2c.

Reviewer #2 (Remarks to the Author):

Overall assessment

The authors present evidence to suggest a novel role for the autophagy protein degradative pathway in the induction of synaptic depression in the mouse hippocampus. Using neuronal cultures and hippocampus slices, the authors demonstrate that markers of autophagy are increased by chemical treatments and electrical stimulation patterns that induce LTD through both NMDARs and mGluRs. LTD induction increased the amount of autophagosomal structures in neuronal dendrites as well as the amount of synaptic proteins within isolated autophagosomes. Further, pharmacological and genetic inhibition of

autophagy was found to prevent LTD induction in acute hippocampal slices. Finally, partially inhibiting autophagy throughout the brain was found to impair cognitive flexibility.

The overall findings of the paper are of potential interest to a wide readership, as it is relevant for studies of synaptic function, synaptic plasticity, cognition and the autophagy field as well. However, there are a number of concerns the authors must address, in particular the quality of the electrophysiology, in order to strengthen the findings and conclusions of the study.

We thank this Reviewer for finding our results interesting and relevant to many fields across neuroscience. We have taken seriously the comments on the quality of the electrophysiology. We acknowledge that some data that may have appeared inadequate had been generated in a previous lab, using an old system. We have now repeated **all** electrophysiology experiments that were in doubt (in terms of quality) using our new experimental set-up. We are happy to report, that our conclusions still hold true and are further reinforced with the newly added datasets. A point-by-point response is provided below.

General concerns

- The overall low quality of the electrophysiology throughout the manuscript makes it difficult to draw conclusions from these experiments for the following reasons:

- o Many of the sample traces displayed indicate poor slice health, as the fiber volley in some traces (e.g. Figure 4C) is larger than the EPSP response.

- o The sample traces also all look quite distinct from each other, indicating highly variable electrode positioning and slice quality. In addition to large differences in fEPSP size between experiments, the waveform varies between experiments (some traces the FV is distinct, others it is not discernable) and in other cases the waveform changes from baseline to post-LTD induction (e.g. Fig. S4C)
- o The baseline in many experiments is highly unstable and too short to accurately determine if the recordings are stable before LTD induction. A 20-30 minute baseline with minimal drift and variability is needed.

- o The shape of the LTD induction curves are not consistent with the literature for many experiments. Following LFS, there should be a greater degree of depression which decays to a stable LTD over the first ~30 minutes post-LFS, which was not observed for most LFS experiments. Further, NMDA application, especially when given for 10 minutes (which is longer than the accepted standard in the literature of 3-5 minutes) should induce a robust depression of the fEPSP to nearly 0, as opposed to the

slow decline observed by the authors. A similar issue occurs for DHPG-LTD as well, which should have a larger acute phase of LTD during application.

These points refer to Figure 6a-c, Figure 4c and Figure S4 of the original manuscript. As mentioned above, we have repeated these experiments. We invite the Reviewer to see our revised figures. We have a stable baseline of 20min before the LTD-inducing protocol, as suggested by the Reviewer. Also, in our new experiments the waveform and shape of our curves are consistent with what is described in the literature. These experiments are now presented in **Figure 6** and **Figure 7** of the revised manuscript.

o **The fEPSP peak amplitude is quantified not the fEPSP slope.** As changes in the peak amplitude can reflect alterations to not only synaptic AMPARs, but also altered inhibition and/or voltage-gated ion conductances, the fEPSP slope needs to be quantified instead which more accurately reflects only AMPAR currents.

As suggested by the Reviewer, all field recordings are now quantified and presented as fEPSP slopes.

o The LFS protocol used by the authors to induce LTD for electrophysiology experiments was 1200 pulses at 1.4 Hz. Why was this protocol used instead of the standard 900 pulses at 1 Hz? Further, given this different protocol the authors should determine whether this protocol induces an NMDAR-dependent only LTD or if mGluRs are activated as well.

It is well-accepted in the field that NMDAR-dependent LTD can be induced in hippocampal slices with several protocols that are considered “standard” in the field and they use 900 or 1200 pulses and frequencies in the range of 1-3Hz^{15,18,19}. As shown in **extra data_R2_1**, the LTD triggered by our protocol can be completely blocked in the presence of AP5, indicating that it is NMDAR-dependent.

• A recent publication by Shen et al., (2020, Nature Communications) also examined the role of autophagy in hippocampal long-term depression and cognitive behaviours. Given the overlap in focus and opposing findings, the authors need to cite and discuss this paper in their manuscript.

In our revised manuscript we have cited the publication from the lab of Prof. Zheng Li and have discussed several points that could explain the conflicting results. Please also see our extended response to Reviewer #1, point 10.

• For many of the example western blot images shown, the loading controls were not displayed, which is critical for assessing the band density for the protein of interest. Further, to enhance clarity, the order of the treatment conditions should be the same in the example images as in the graph.

This point does not concern **many** blots, as the reviewer stated, but only one, which was Fig.4h in the original and Figure 4d in the revised manuscript. The loading control is not displayed because we have quantified to total protein levels. This is now clearly stated in the figure legend. This approach is preferred when applying treatments, such as BafilomycinA1, which may affect many proteins. A similar quantification approach has been employed in Figure 4e-f of the revised manuscript.

• In the ICC and IHC experiments examining the increase in autophagy markers following NMDA or DHPG treatment, is there a negative control region, such as the soma where LTD is not expected to occur to determine if autophagic markers are generally increased throughout the cell or only in synaptic regions where LTD is occurring?

As suggested by the reviewer, we quantified the LC3B intensity in the somatic region of cultured neurons, under control conditions or 15 minutes after NMDAR- or mGluR-LTD. As shown in **Figure S1b**, the LTD-triggering stimuli failed to increase LC3B intensity in the soma. In the case of mGluR-LTD there was a small but significant decrease in the soma.

Specific concerns:

Figure 1

• Why did the authors choose to analyze only the 15-minute time point? It is important to understand whether autophagy is only activated transiently during the induction phase of LTD or

continuously upregulated throughout the maintenance phase. For two of the key proteins examined, LC3B and ULK1, the authors should perform a time-course experiment to analyze changes at 5, 15, 30, 45 and 60 minutes following LTD induction.

As suggested by the reviewer, we also analyzed the number of LC3B-positive structures in dendrites at two additional time-points after chemical NMDAR- and mGluR-LTD, namely at 30 and 60 minutes. As shown in **Figure S1a**, dendritic LC3B-positive structures remain elevated at both time-points, suggesting that their induction is not only rapid but also sustained.

This was also confirmed by super-resolution experiments after LC3B labeling, 60 minutes after NMDAR- and mGluR-LTD. As shown in **Figure 2b**, the density of dendritic U-shaped LC3B-positive structures remains elevated in the LTD conditions, compared to control. These findings are in line with a paper from the lab of Prof. Huan Ma, showing that autophagy is transcriptionally induced after LTD and is required for LTD maintenance (*in press* in Cell reports).

- Ifenprodil and MTEP/JNJ were found to inhibit the increase of LC3B-II following NMDA and DHPG application, respectively. However, the authors should confirm that these treatments do indeed block the decrease in surface GluA2.

These drugs have been used extensively in the field of synaptic plasticity, both in vitro and ex-vivo and their effects on GluA2 surface levels are well described. For a reference please see^{20,21}.

- In the Methods section the number of days in vitro of the neurons in culture used for these and any other experiments using cultured neurons needs to be indicated.

We have now clearly indicated in the materials and methods that we use div16-17 neurons

Figure 2

- In the Methods it is stated that rat hippocampal neurons seeded on astrocytes were used for the dSTORM imaging. Why were these neurons/conditions used only for these experiments?

These experiments were performed in the lab of Dr. Eric Hossy and Prof. Daniel Choquet. It is indeed reassuring that the increase in endogenous LC3B-positive structures in dendrites after LTD is not restricted to mouse neurons but can also be recapitulated in another lab, using rat neurons cultured in a different way.

- For the dSTORM example images, it would help if the authors clarified on the example images themselves i) what condition was being shown (i.e. baseline, or post-stimulation) and what the U-shape structure was that was being quantified.

We have now modified the figure and clearly indicated the conditions. We have also circled the U-shaped structures for more clarity.

- Given the 4 proteins examined independently in 4E are hypothesized to be part of the same structure, the authors should examine whether the puncta for these proteins co-localize following NMDA/DHPG stimulation which would provide support for this notion.

This would be indeed great to show, however, the only antibody that works for each component for immunocyto/histo-chemistry is for all cases raised in rabbit. Therefore, these antibodies cannot be combined.

- Using ICC, Atg101 was found to be drastically increased following DHPG treatment, while in the Western blot data there is no change. Why is there such a discrepancy between assays? Further, the representative images of the blots for Atg101 seem very faint. Were the antibody/conditions used to detect Atg101 in Western blot sufficiently sensitive?

The antibody works well for Western blot, as it identifies a band of the expected molecular weight. The fact that it works better for immunocytochemistry, does not mean that the Western blot results are not valid, especially given the differences between control and LTD conditions.

- The band for LC3B-II is much closer to 17 kDa in 2F, while elsewhere in the manuscript the LC3B-II band is closer to 10 kDa, which is consistent with the literature. Is the band being quantified in this figure indeed the right size for LC3B-II? The LC3B-II band runs at 14kD and how close it appears to the LC3B-I band depends on how long the gel was run. Longer running separates the bands more. However, as LC3B-II is small, one can lose it with a longer run (as it can easily run out of the gel). My lab has extensive experience in quantifying endogenous LC3B-II in neurons and in the brain and we are indeed sure that we are quantifying the correct band.

- As LC3B-II is a marker of intermediate structures in autophagy, it has been found that increased LC3B-II can indicate either increased or decreased autophagic flux (Mizushima and Yoshimori, 2007). Therefore, the authors should use known activators and inhibitors of autophagy in their experimental conditions to determine if increase LC3B-II is indeed an indicator of enhanced autophagy

We agree with this comment, which is why we have used BafilomycinA1 to block lysosomal acidification and the flux of the LC3B-II contained in the inner AV membrane. As shown in **Figure S1f**, treatment of neurons with 50nM BafilomycinA1 during and one hour after the NMDA and DHPG pulses results in a significant accumulation of endogenous LC3B-II in the LTD conditions, as compared to

control. Of note, this concentration/duration of BafilomycinA1 treatment is considered low in the field and as seen here it is not sufficient to induce a strong accumulation of LC3B-II in the control condition. These findings, taken together with the induction of ULK1-complex components in dendrites after LTD, clearly indicate that the increased dendritic LC3B-positive structures and increased LC3B-II levels after LTD are explained by increased and not decreased autophagy.

Figure 3

- In acute brain slices, the cut surface of the slices (up to ~50 microns) is damaged. When the authors performed immunohistochemistry imaging from slices (3A-E), did they ensure the imaged were coming from greater than 50 microns below the surface of the slice?

This is an important detail. Indeed, in all conditions, free-floating hippocampal sections of 100µm were prepared with a vibratome, using high vibration frequency and low speed to minimize surface damage. Imaging was always performed in the mid-depth of the sections.

- In the example images of 3A-E, it can be seen that some of the autophagy markers are increased in regions outside of MAP2+ dendrites (e.g. Atg13). Were the quantified puncta in the Figure only those that were co-localized with MAP2 to ensure the signal measured was from principal neurons?

It could be sometimes observed that ULK1-complex components also increased in what appeared as glial cells in the *stratum radiatum*, near dendrites. However, we have not investigated the effect of LTD treatments on glial autophagy. We have only quantified the signal that co-localized with MAP2.

- For the experiments in 3F-I, it is noted in the text that the control condition is that neurobiotin labelling was not performed, and so AVs in dendrites cannot be quantified as there are no labelled dendrites to being with. However, there is no control condition in which dendrites are labelled but LTD is not induced. As such, the authors should include a new condition in which LFS is delivered in the presence of AP5 to block LTD induction but control for any effects of the increased stimulation/neuronal activity due to the LFS itself.

The control condition in Figure 3f is neurobiotin delivered in the absence of LFS. However, in Figure S3a, the control condition is no neurobiotin, in order to ensure that the addition of DAB does not by itself give a non-specific signal. On page 6 there was a sentence which was misleading, as it refers to Fig.S3a and not to Fig.3f. In the revised manuscript, we have clarified these two different controls, which serve different purposes. We have no reason to believe that the effects on post-synaptic AVs is not specific, as the emergence of dendritic LC3B-positive puncta in Figure 1e and of Ulk1-complex components (Figure S2b), after NMDAR- and mGluR-LTD are prevented by ifenprodil and MTEP/JNJ treatments, respectively. Moreover, as shown in **extra data_R2_1**, we have quantified the density of AVs in buttons forming synapses onto neurobiotin-labeled dendrites and found that it was not different between control and LFS conditions.

- In addition, for the experiments in 3F-I, how did the authors determine which spines were stimulated and which were not? Given the neurobiotin was applied generally to the area of recording, the spines that would be labelled will be a mixed population of both stimulated and non-stimulated spines (the latter being the predominant type).

In this study we did not determine which spines are stimulated and which not. In fact, autophagic structures are observed mainly in the dendritic shaft and not in spines. Therefore, the comparison is between control and LTD dendritic shafts of the recording area.

- Finally, for the quantification in 3G-I, the N value is the number of dendrites imaged. How many slices from how many animals did these dendrites come from?

We analyzed 3 animals/condition and 3 slices per animal. We have now included this information in the figure legend.

Figure 4

- In the sample traces from the SBI-treated group in 4C, the fiber volley is substantially increased from baseline to post-LFS. The authors should quantify the fiber volley amplitude for this experiment in order to determine to what degree it increases after LFS and if this occurs only in the SBI-treated group.

This is not the case and must be a misunderstanding based on a single trace that was not well-selected. Please see the representative traces in the revised Figure 6.

Moreover, as shown in **extra data_R2_2**, the fiber volley amplitude at different intensities of stimulation is not significantly different between control and SBI-0206965 conditions (N=8 animals per condition, two way ANOVA, $F(1,70)=0.9272$, $P=0.3389$).

Finally, SBI-0206965 does not have any effect on basal neurotransmission. In the revised manuscript, we present the normalized fEPSP baseline before and after SBI treatment (**Figure S6b**), the paired pulse ratio plotted for different interval times (**Figure S6c**) as well as the fEPSP slope-fiber volley amplitude curve (**Figure S6d**). In all

cases, SBI-0206965 didn't show any significant differences compared to control (aCSF). These findings are in line with our recent publication (**Compans et al., 2021, Nature Communications**), where we show that SBI-0206965 alone does not impact on mEPSC amplitude, while it fully blocks NMDAR-dependent LTD²⁰.

- Why was SBI-0206965 used to inhibit autophagy in the electrophysiology experiments while bafilomycin was used for the same purpose in biochemistry experiments? Preferably, given the off-target effects of each drug, both inhibitors would be used for both electrophysiology and biochemistry experiments.

We thank the Reviewer for giving us the opportunity, with this comment, to explain the important differences of the drugs used in autophagy research. It is important to realize that SBI-0206065 and bafilomycinA1 act differently. SBI-0206065 targets the kinase activity of ULK1, therefore preventing the most upstream events of autophagic vesicle biogenesis. It was used in electrophysiology and in surface labeling experiments in order to determine the requirement of autophagic vesicle biogenesis for AMPAR internalization and LTD induction. Bafilomycin, on the other hand, acts on lysosomes and prevents their acidification. It has no direct effect on autophagic vesicle biogenesis or on cargo selection. It was used in the biochemical experiments to measure the autophagic flux after LTD and to demonstrate that GluA2 and PSD95 are degraded by the lysosome, as blockade of lysosomal acidification by BafA1 caused their accumulation. In fact, we do not think that BafilomycinA1 treatment in electrophysiology would (or should) block LTD, as it does not interfere with the removal of synaptic proteins from the post-synaptic density by the autophagic machinery.

• The authors should test whether pharmacological activation of autophagy is sufficient to induce LTD to strengthen their conclusions of the causal role of autophagy in LTD.

This comment also connotes to the pharmacological tools (rapamycin) used in the Shen et al paper. However, rapamycin is NOT a specific inducer of autophagy, as by blocking the mTOR pathway it also suppresses protein translation. Moreover, in the Shen et al., paper *Atg5^{F/F};CA1-Cre⁺* slices are not compared in the absence and presence of rapamycin, but only compared to control slices (in either condition), as shown in Figure 2c-d of their paper. Therefore, even in these experiments the autophagy-dependent effects of rapamycin remain unclear. In brief, we would be happy to test if autophagy activation in postsynaptic neurons is sufficient for LTD induction, but we are not aware of a specific autophagy inducer.

• In 4I and J, the authors use a one-way ANOVA. Given there are 2 independent variables (NMDA/DHPG and +/- baf.) a 2-way ANOVA is the appropriate statistical analysis.

We have changed the statistical analysis using two-way ANOVA.

• Generally, in the literature, chemical LTD, especially in the early phases, is associated with changes to GluA phosphorylation and reduced surface expression, but not changes in total protein of GluA or PSD-95. Why is such a large decrease in total PSD-95 and GluA2 protein levels observed rapidly following LTD in the author's conditions?

We do not agree with this comment. Previous work has clearly shown that PSD95 is degraded rapidly after LTD and that a reduction in its levels in whole cell lysates can be observed as early as 15 minutes after LTD²².

• Is the quantification of GluA2 and PSD-95 in 4I and J normalized to beta-actin loading control?

They are normalized to total protein levels. We have clarified this now in the Figure legend.

Figure 5

• In 5A, why is the amount of LC3B-II roughly equivalent in the nuclei fraction as the AV enriched fraction? Would it not be expected that the AV enriched fraction should have substantially more LC3B-II?

We apologize for the confusion we may have caused, as the blot shown was qualitative.

We have replaced it with a new blot (Figure 5a), where equal protein amounts were loaded for each fraction. As one can see, LC3B-II is enriched in the final AV preparation.

• Does the method for isolating AVs used in this Figure enrich for those in pyramidal cells specifically, as autophagy occurs in all cell types?

We apologize for not having described this better. We purify AVs from all cells of the hippocampal slice. This also explain the fact that the vast majority of the cargo is unchanged between control and LTD conditions.

- For 5B, in addition to the close-up images of putative AVs, a lower zoom version of the image should be shown so that the number of these structures relative to everything else contained in the AV fraction can be assessed.

In the revised manuscript (**Figure S5a, top panel**) we have now included a lower magnification of the electron micrographs.

- In 5E, how is it known that the number of AVs loaded in each lane is equivalent? The authors should include a loading control (e.g. LC3B-II) and normalize the protein values in each lane to LC3B-II to ensure that the increase in synaptic proteins is not due to an increase in the number of isolated AVs.

The same μg of protein have been loaded in control and LTD lanes and they are normalized to p62, a cargo protein that does not change in the time-point where the AVs were purified. The fact that p62 is stable between control and LTD conditions is also suggested by the quantitative mass-spec analysis.

- The authors mention multiple times in the text that under basal conditions, autophagy is restricted in neurons to the axon tip. If this is the case, why are post-synaptic proteins observed in AVs under control conditions, in some cases to the same level as in the PSD fraction (e.g. ITPKA, SAP97)?

We don't know whether these postsynaptic proteins are recycled after reaching the post-synapse or potentially during their synthesis in the ER. This question is beyond the scope of this manuscript. However, we agree with the Reviewer that describing the autophagic cargo in the brain is a fundamental question that needs to be addressed in the future.

Figure 6

- For the Atg5 conditional KO mice, were the control mice (CreER negative) given tamoxifen as well to control for any effects of tamoxifen administration?

This is an important point and we have clarified it in the text. Yes, both *atg5^{ff}* (control) and *thy1-cre^{ERT2};atg5^{ff}* (thy1-cKO) animals were administered tamoxifen.

- LFS in Thy1-cKO mice induced a small potentiation effect. It would be interesting to know if LTP is also enhanced in these mice or if autophagy only plays a role in LTD.

We agree with the Reviewer that this is a very interesting question and we are in fact addressing it in a separate project. This is beyond the scope of this manuscript.

- Basal synaptic properties (I/O and PPR curve) need to be assessed in the cKO and shRNA mice.

As suggested by the Reviewer, we have assessed the basal synaptic properties both in *thy1-cKO* animals (compared to control) and in *sh-atg5* injected animals (compared to *sh-scramble*). These new experiments are included in Figure S7 of the revised manuscript. We did not observe any differences on basal synaptic properties at P22 (same age as LTD recordings were performed) between *thy1-cKO* and control mice (Figure S7d-e). Similarly, no differences were observed at P120 (same age as LTD recordings) between *sh-atg5* and *sh-scramble* sides of the hippocampus (Figure 7f-g).

- LFS-LTD is well known to be much more robustly induced in younger mice (>P28) and for many labs not possible to induce in adult mice. Why was the amount of LFS-LTD actually higher in the control adult mice in the shRNA experiments (5-6G) compared to young control mice in 5-6A?

We remind the Reviewer that LTD was induced in the young and old animals using different LFS protocols, both in terms of frequency and duration. It is, therefore, not possible to compare and contrast the magnitude of the response between the two experiments. Using a different protocol for adult animals is of paramount importance to induce sustained LTD. In fact, in the Shen et al., paper LTD was not maintained by their LFS protocol (900 pulses at 1Hz) in control adult slices (Figure 10a of their paper), as fEPSPs returned to baseline within 20 minutes after the end of stimulation, making it a non-credible control against which the *Atg5^{F/F};CA1-Cre⁺* slices were compared (see also our response to next point).

- The authors state that the justification for performing the shRNA knockdown experiments is to examine the role of autophagy in postsynaptic CA1 neurons only. Given there is a commercially available mouse line (CAMKII-Cre, T29-1) which has CA1-restricted Cre expression that onsets at ~P17, why was the shRNA method chosen instead?

This comment also connotes strongly with the Shen et al., paper. The reviewer urges us to use the *Atg5^{F/F};CA1-Cre⁺* mice that were used in the Shen et al., paper, instead of our CA1-specific knockdown of *atg5* in adult mice. We strongly disagree with this suggestion, because in these animals ablation of *atg5* starts in the third postnatal week¹³, namely before the period of developmental spine pruning that occurs between P20 and P30 in the hippocampus. Previous work has clearly demonstrated that autophagy is required for developmental pruning¹⁴. Therefore, pruning is likely impaired in the *Atg5^{F/F};CA1-Cre⁺* mice, leading to an increased number of rigid spines in the adult, which can surely interfere with the responses to LFS stimulation. Therefore, this deleter is inadequate for studying the direct effects of autophagy deficiency on LTD. We also believe that this is a limitation of the Shen et al., paper, as they completely ignore the developmental spine

pruning deficits that these mice are likely to carry and their indirect effects on LTD. Our approach circumvents this problem by knocking down *atg5* in the adult CA1, long after the end of pruning.

- What is the knockdown efficiency of the shRNA on Atg5 levels themselves?

As suggested by the Reviewer we have now included this information. It is well known in the field that the available Atg5 antibodies only recognize the Atg5/Atg12 complex and not the monomeric Atg5. We observe a significant decrease of about 80% in the intensity of this band (now included in Figure 7f).

Figure 7

- While the justification of using Atg5flx/WT mice to only partially inhibit autophagy to avoid any effects of full knockout is understandable, why a different Cre-driver line was used compared to Figure 6 is unclear. The same experiments could have been performed using the Thy1-CreER mouse that had been already shown to have an LTD deficit. We fully agree with this point and have now behaviorally assessed the thy1-cKO animals, following the experiments recommended by this Reviewer. Please see below for details.

- Electrophysiology experiments were not performed in the nestin-cHET mice and thus it is not known if there even is an LTD deficit in these mice. Further, as nestin-Cre mice express Cre in all neurons, there is no way to link changes to LTD in CA1 post-synaptic pyramidal neurons to the observed behavioural phenotype.

We have in fact performed electrophysiology recordings in the nestin-HET mice, but they show a different response compared to the thy1-cKOs. The main difference is that while in both animals LTD fails to be induced, in nestin-HET there is no potentiation, whereas the thy1-cKOs exhibit potentiation. These differences may arise indeed from ablation of autophagy in all cells and from early in development under the nestin promoter. In our revised manuscript we have removed the nestin-HET data and replaced them with behavioral tests with the thy1-cKO animals.

- The authors should perform the following experiments to help clarify these issues;

- o At a minimum use the Thy1-CreER mice instead of Nestin-Cre, but ideally using the CAMKII-Cre mice crossed with Atg5-floxed mice.

As mentioned, we have replaced the Nestin-cKO mice with the thy1-cKO mice (see also below). However, we do not agree with the CamK2a-cKO suggestion for reasons we explained earlier (please see response to Reviewer #1, point 10). Briefly, these behavioral tests are established in adult mice and the CamK2a promoter starts being expressed before the developmental spine pruning period. Therefore, we would be analyzing mice that have increased number of spines, due to the described role of autophagy in this process.

- o In both control (Cre negative) and cKO mice, administer the TAT-GluA23Y peptide, which has been shown to block AMPAR endocytosis induced by LTD (Amadian et al., 2004; Brebner et al., 2005; Fox et al., 2007; Dong et al., 2013)

- o The expected result is that TAT-GluA23Y in control mice will induce a cognitive flexibility deficit similar to the vehicle-treated Atg5 cKO, and that administration of TAT-GluA23Y in these cKO mice will not cause a further deficit.
- o These experiments will better delineate i) the role of autophagy specifically in CA1 neurons in cognitive flexibility and ii) if any changes observed are due to blockade of LTD (as indicated by an occlusion of the effect of a peptide already known to inhibit LTD).

We thank the Reviewer for this excellent suggestion. We have performed these experiments and the results are now presented in **Figure 8** and **Figure S8** of the revised manuscript. First, we compared adult *atg5^{fl/fl}* (control) and *thy1-*

cre^{ERT2};atg5^{fl/fl} (*thy1-cKO*) in the CognitionWall test of behavioral flexibility²³. Both genotypes were administered with tamoxifen for 5 consecutive days starting at P45 (after the end of developmental spine pruning) and experiments were performed 15 days after the end of tamoxifen injections.

None of genotypes exhibited a side bias (left-middle-right holes) (**Figure S8a**) and all had similar responses in the initial discrimination learning (**Figure 8a**). However, unexpectedly, in the reversal learning, the *thy1-cKO* animals exhibited a slight but significantly faster response in reaching the 80% criterion, compared to the control animals (**Figure 8b**). This could not be explained by differences in mobility, as both genotypes had similar responses in the total entries during reversal learning (**Figure 8c**) and in the total distance moved (**Figure 8d**).

Given these findings, we wondered whether the better performance of the *thy1-cKO* animals is LTD-dependent, which would be incompatible with our electrophysiology findings. To this end, we used the TAT-GluR23Y peptide which blocks LTD by preventing AMPAR internalization, but has no effect on LTP. As a control we used TAT-GluR23A, which was previously shown not to have any effect on LTD. The peptides were administered by intraperitoneal injection to the control and *Thy1-cKO* animals only on the reversal days of the experiment (day 3 and 4), one hour before the beginning of the experiment (see revised materials for details). Before peptide administration, the four groups had similar responses in discrimination learning (**Figure 8e**, and in line with the finding in Figure 8a) and did not exhibit any side bias (**Figure S8b**).

thy1-cKO animals administered with the control peptide (TAT-GluR23A) exhibited again a slight but significant faster response in reversal learning, compared to control animals that were administered the control peptide (TAT-GluR23A) (**Figure 8f**, and in line with Figure 8b). As expected, TAT-GluR23Y significantly worsened the performance of control animals in reversal learning (**Figure 8f**), with some animals never reaching the 80% criterion. However, TAT-GluR23Y had **no effect** on the performance of *thy1-cKO* animals in reversal learning (**Figure 8f**). Of note, animals in all conditions exhibited similar mobility, as shown by the number of entries

(**Figure 8f**).

during the reversal phase (**Figure 8g**) and the total distance travelled (**Figure 8h**). These findings clearly indicate that the better performance of the *thy1*-cKOs in the reversal learning is not LTD-dependent. We speculate that it may be mediated by the potentiation responses that are induced in these animals following LTD-triggering stimuli.

Reviewer #3 (Remarks to the Author):

Kallergi et al in this Ms deal with the possible role of autophagy in the turnover of postsynaptic proteins during LTD. They show that induction of chemical LTD (cLTD) by either NMDAR or mGluR stimulation elicits an elevation of autophagic proteins and vacuoles in dendrites and at postsynaptic elements in cultured neurons and in acute hippocampal slices line with previous work on the induction of autophagy by cLTD (Shehata et al J Neurosci 2012). Application of SBI-0206956, an inhibitor of the autophagic ULK1 kinase, is further shown to block cLTD and GluA2 endocytosis and leads to increased localization of PSD95 to LC3B positive puncta in neurons. Thee data correlate with the presence of postsynaptic proteins in AV-enriched fractions from brain. Finally, low-frequency stimulation-induced LTD is shown to be occluded upon knockdown of the autophagy protein Atg5, while heterozygous loss of Atg5 in neurons in mice leads to impaired odor reversal learning. From these data the authors propose a model according to which autophagic turnover of postsynaptic receptors and associated PSD proteins is required for LTD.

Overall, though I find the study interesting in principle, at this stage I remain unconvinced that the observed loss of LTD in slices from *Atg5*-cKOs indeed relates to defective endocytic internalization and degradation of AMPARs and associated PSD proteins.

We thank the Reviewer for finding the study interesting and for his/her constructive suggestions. We have performed many experiments during the revisions and have addressed the major points that were raised, as detailed below.

Specific comments:

1. A major conceptual problem is the interplay between autophagy and endocytosis. In Fig 4a,b it is shown that inhibition of ULK1 by SBI-0206956 not only causes a shift from LTD to LTP but also abrogates GluA2 endocytosis, in fact more potently

than peptide-mediated dynamin inhibition. This is puzzling as dozens of studies on autophagy have failed to detect a "retrograde" action of failed autophagy on endocytosis, while conversely, some studies suggest that endocytosis or at least endocytic proteins can mediate early steps of autophagy. These findings in my view cast doubt on the specificity of the drug and/ or the role of ULK1 as a regulator of autophagy as opposed to other possible functions. In my view these experiments need to be repeated under conditions of specific genetic ablation of autophagy. The cKO-Atg5 data shown in Fig6 suggest that at least the physiological effects may be related, however the underlying mechanism remains enigmatic in my view.

We thank the Reviewer for this comment, which allows us to elaborate on this important point of the autophagy-endocytosis interplay. First of all, to eliminate the doubt on the specificity of the pharmacological inhibition (SBI-0206965), or the role of ULK1 in general, we have added an experiment where GluA2 surface levels are examined under baseline conditions and after LTD in neurons with a knock-down of *atg5* (infected with a virus expressing 4 *sh-atg5* sequences under the Camk2a promoter). As a control, neurons were infected with a virus expressing *sh-scramble* sequences under the Camk2a promoter. These results are shown in **Figure 4b** of the revised manuscript and complement the findings with SBI-0206965, shown in **Figure 4a**. We found that while chemical NMDAR- and mGluR-LTD strongly reduced the surface levels of GluA2 in *sh-scramble* neurons, they failed to do so in *sh-atg5* neurons. These results are also reflected in the protein levels of PSD95

and GluA2 before and after LTD in neurons. The levels of both GluA2 and PSD95 significantly decrease in control neurons after LTD, but not in neurons treated with SBI-0206965 (**Figure 4e**) or expressing *sh-atg5* (**Figure 4f**).

These findings are also consistent with the results in our recent publication, showing the co-localization of LC3B and PSD95 rapidly after NMDAR-LTD, by super-resolution microscopy⁹.

However, under baseline conditions *sh-scramble* and *sh-atg5* neurons had similar levels of surface GluA2 (**Figure 4b**) and this is also the case between untreated and SBI-treated neurons (**Figure 4a**). Therefore, these findings suggest that a distinction should be made between constitutive and

induced, or regulated, endocytosis.

To our knowledge, all previous studies, such as the one from the group of Prof. Volker Haucke, have examined this relationship under baseline conditions²⁴.

The interdependence between autophagy and endocytosis is complex and may depend on the cellular context. In non-neuronal cells, Ravikumar and colleagues²⁵ demonstrated that in HeLa cells clathrin heavy-chain interacts with Atg16L1 and is involved in the formation of Atg16L1-positive early autophagosome precursors. Inhibition of clathrin-mediated internalization reduced the formation of both Atg16L1-positive precursors and mature autophagosomes (monitored by LC3B puncta). In neurons, we tested whether blocking clathrin-mediated endocytosis by a dynamin inhibitor affects the emergence of dendritic LC3B puncta under control conditions and upon chemical NMDAR- or mGluR-LTD. We found that dynamin inhibition did not have an effect in any of the conditions tested (**Figure S4a**). Therefore, we believe that clathrin-mediated endocytosis is not required for autophagic vesicle formation in dendrites. At the same time, we don't claim that defective autophagy impairs the endocytic machinery. Instead, our findings suggest that when autophagy is impaired, AMPAR subunits fail to internalize not because the endocytic machinery is impaired but because PSD95, a key protein that anchors them to the postsynaptic density, fails to be degraded by autophagy. This interpretation is also consistent with our recent publication²⁰.

2. ULK1 inhibition via SBI-0206956 greatly reduces fEPSPs in the traces shown in Fig4c, an effect the authors surprisingly seem to have ignored. How is this depression of basal transmission explained that also clashes with recent observations in Atg5 cKO slices, in which increased excitatory transmission was observed (Kuijpers et al Neuron 2020) and with the authors' own data shown in Fig 6? I again wonder about the specificity of the SBI-0206956 effect and phenotype.

This is not the case, please see also our response to Reviewer #2 - response to Figure 4 comment. We have included new experiments that can be found in Figure 6 and Figure 7 of the revised manuscript. SBI does not decrease the baseline, and

perhaps this is a misinterpretation of a trace that is not well chosen on our side and may have caused confusion. We now clearly show in **Figure 6** and **Figure S6**, that SBI-0206965 treatment does not affect the fEPSP baseline and basal synaptic properties. This is also in line with our recent paper, showing that SBI alone does not impact on mEPSC amplitude, while it fully blocks NMDAR-dependent LTD⁹.

Importantly, in the Kuijpers et al paper *atg5* was ablated with an *emx1-cre*, leading to recombination of *atg5* embryonically in pallial progenitors and their progeny, namely forebrain excitatory neurons and astrocytes, and analysis is performed at 2-month old animals²⁶. Therefore, there is a prolonged ablation and accumulation of deficits. Here, application of SBI results in acute autophagy blockade and therefore, the results cannot be directly contrasted and compared to those in Kuijpers et al.

In the case of our *thy1-cKO* slices, *atg5* is ablated at P15 and recordings performed at P22. Again, the duration and developmental stages involved are different. As shown in **Figure S7d-e** (see panels below), this brief ablation of *atg5* does not impact basal synaptic properties.

We discuss these differences in our revised manuscript. We believe that our conditions, which result in brief autophagy deficiency, are ideal for revealing the post-synaptic roles of autophagy, because pre-synaptic deficits are not manifested under these conditions. By contrast, the model chosen in Kujipers et al., namely prolonged autophagy ablation, reveals very interesting pre-synaptic deficits. Therefore, we consider the two studies to be complimentary and not contradictory.

3. In Fig5 the authors present proteomic data to identify the autophagic cargos targeted during LTD. In my view these data are too preliminary to allow any firm conclusions. I am not impressed by what the authors refer to as "enrichment of LC3B". In fact, in the WB shown in panel a there is no enrichment of LC3B in the alleged AVs. I suspect that the isolated material, in addition to AVs shown in the EM images in panel b, comprises a number of contaminants that may arise from the

biochemical fractionation procedure. **The isolated AVs, hence, need to be compared to material isolated from Atg5 KO animals and only then, it can be concluded that the identified proteins likely represent autophagic cargo.** Alternatively, correlative super-resolution LM/ EM analysis could be used to show that the AVs seen in the EM indeed contain the alleged cargo proteins.

We apologize for the confusion we may have caused, as the blot shown in the original manuscript was qualitative. We have replaced it with a new blot (**Figure 5a**), where equal protein amounts were loaded for each fraction. As one can see, LC3B-II is enriched in the final AV preparation.

The concern of the Reviewer that our preparation may contain contaminants is well taken, especially as homogenization of the brain may result in broken synapses that

close to form synaptosome-like structures. We have performed several experiments to ensure that this is not the case. Please see also our response to Reviewer 1 (point 7).

First, as suggested by the Reviewer, we attempted to purify AVs from P25 hippocampal slices prepared from *nestin-cre;atg5^{fl/fl}* mice, starting from the same number of animals and following exactly the same protocol. However, the amount of material obtained in the final AV preparation was negligible and could not be pelleted after centrifugation. This suggests that the vast majority of the AV preparation is comprised of autophagy-related material.

Second, we performed immuno-purification of the intact AVs obtained in the AV preparation with magnetic beads and an antibody against LC3B. As a control, an IgG antibody was used. These results are presented in Figure S5 of the revised manuscript. As shown in **Figure S5e**, the majority of the AV preparation (input, lane 1) was immune-precipitated with the LC3B Ab (lane 2), as can be seen from the top panel of the total protein stain. By contrast, almost no material was precipitated by the control IgG (lane 3). This was also reflected by the amount of proteins that was left in the unbound (flow-through) material. In the case of the LC3B-IP, after centrifugation, the pelleted unbound material was very small (arrowhead in Eppendorf picture insert for lane 4) and contained a very small fraction of the input (lane 4, protein stain). We could still detect a very small amount of LC3B in this unbound material (LC3B blot, lane 4), suggesting that the IP was highly but not 100% effective in precipitating AVs. In the case of the IgG-IP, after centrifugation, the unbound material formed a big pellet (arrowhead in Eppendorf-

tube picture insert for lane 5) and its protein and LC3B content were similar to the input (lane 5). Importantly, the levels of PSD-95 were reduced in the unbound of the LC3B-IP (lane 4) compared to the unbound of IgG-IP (lane 5), indicating that this synaptic protein is not a contaminant but is precipitated in the LC3B-positive fraction.

4. A prediction from the hypothesis that autophagy and subsequent lysosomal turnover play a major role in LTD is that autophagic **cargos accumulate in the conditional absence of Atg5 upon LTD induction in slices or neurons from genetically altered mice**. For example, one would expect **GluA1> GluA2 as well as CaMKII and PICK1 to display**

prolonged half-lives under such conditions. Can this be observed? In fact, the title of the paper claims to show that dendritic autophagy degrades postsynaptic proteins, while this bold claim in my view is not or only insufficiently supported by the data.

As explained in response to point 1, this can be observed

and we have included these experiments in the revised Figure 4. Briefly, the levels of both GluA2 and PSD95 significantly decrease in control neurons after LTD, but not in neurons treated with SBI-0206965 (Figure 4e) or expressing *sh-atg5* (Figure 4f).

5. I wonder whether U-shaped phagophores can be reliably distinguished from autophagosomes by STORM microscopy. Under-sampling of single molecules make the reconstruction of entire organelles and their shapes difficult. Moreover, it is unclear to me how the amount of LC3B not associated with phagophores shown in Fig S2a was quantified. These data also seem to clash with the EM morphometry shown in Fig. 3f+h, in which an increase in complete double membrane autophagic vesicles (i.e. non-phagophores!) is observed. From the STORM data one would expect an accumulation of incomplete U-shaped early structures, which, however does not seem to be observed at the EM level.

STORM can easily distinguish between cytosolic LC3B, which appears as scattered single dots, and LC3B associated with curved membrane structures. The latter appears as a high density of puncta that coalesce onto a U-shaped structure, as shown in Figure 2a (circled structures). However, we do not think that STORM microscopy can distinguish between phagophores and autophagic vesicles, and this is exactly the reason we also performed electron microscopy experiments in hippocampal slices.

6. I was surprised to see that for the in vitro experiments neurons isolated from E15 mice were cultured for only 6-7d, a time frame during which hardly any synapses form. Is this a typo? Typically, hippocampal neurons are cultured for at least 12-15d to display functional synapses and synaptic responses. Also, it is unclear from the results, methods, and legends whether hippocampal or cortical neurons were used for the respective experiment.

We have used cortical neurons at div16-17. There was a typo, which we have corrected.

Minor points:

7. A standard test used to assess the role of LTD at the behavioral level is spatial reversal learning, e.g. using the Morris water maze test. Do Atg5 Nestin-cHet display defects in this paradigm?

Please see our response to Reviewer #2, on page 17-18 (response to Figure 7 point)

8. Do Atg5 nestin-cHet mice show defects in LTD similar to thy1-cKO?

Please see our response to Reviewer #2, on page 17-18 (response to Figure 7 point)

9. Fig 1: The authors state that LC3B density decreases with increasing distance from the soma. Where were the ROIs taken in Fig 1C? I guess the total dendrite is measured -right? Are all dendrites from one single neuron taken or just the apical dendrites? This should be specified in the text.

We have only analyzed secondary dendrites. This is now specified in the manuscript.

10. Fig S1A/B: Why does BafA in control conditions not result in increased LC3B-II? From the WB it looks as if in neurons subjected to LTD, LC3B-II is already increased in the absence of BafA (as should be the case according to Fig 1C). BafilomycinA1 was applied for only one hour at a very low concentration (50nM), as we wanted to assess the immediate

effects one hour after LTD. This short treatment is not sufficient to see an accumulation under baseline conditions. In our experience it takes at least three hours to observe LC3B-II accumulation under baseline conditions with this concentration of BafilomycinA1.

11. Figure 2:

2A: It is not clear to me what we actually see here. What do the different panels and colors represent? Furthermore, alleged autophagosomes and phagophores should be marked (but see my comment #5 above). The permeabilization method is not specified in the methods section.

Please see our response to comment #5. We have added the permeabilization method in the methods section.

2E: Images do not seem to be representative for the quantification (I see many more puncta in the images compared to the quantification).

We have now specified that we only quantify the signal on secondary dendrites.

2F: LC3B quantifications are missing.

Thank you, it has been included.

12. Fig 3F: Please mark the vesicle-like structures inside spines and specify what the red arrow and yellow asterisk refer to in the figure legend.

Thank you, we have done so.

13. Fig 3H: The authors use ciliobrevin to inhibit transport of somatic autophagosomes into dendrites and thereby provide evidence that autophagosomes are produced locally. However, as also indicated by the authors, ciliobrevin would likely inhibit trafficking of autophagosomes to the soma and possibly autophagosome biogenesis. It is therefore difficult to interpret these changes in autophagosomes number, especially as the control condition (+ CBD) is missing. Additional experiments to assess LC3B-mRFP puncta dynamics/appearance upon LTD induction could provide additional support for the claim that autophagosomes are produced locally in dendrites.

We have chosen to perform all experiments using endogenous readouts. Although over-expression of fluorescently labelled LC3B has been used extensively in the field, however many control experiments that prove that these species behave like the endogenous protein is still missing. We strongly believe that over-expressing LC3B in neurons is likely to alter its localization and its response to signaling cascades that are activated after synaptic activity. This may be closely related to the roles of cytosolic LC3B in microtubule-associated processes.

14. Figs. 4C, S4: What do the red and black traces represent (\pm LFS)? Please specify in the legend.

Thank you, this has been clarified in the figure legend.

- 1 Grunwald, D. S., Otto, N. M., Park, J. M., Song, D. & Kim, D. H. GABARAPs and LC3s have opposite roles in regulating
ULK1 for autophagy induction. *Autophagy* **16**, 600-614, doi:10.1080/15548627.2019.1632620 (2020).
- 2 Cheng, X. T. *et al.* Characterization of LAMP1-labeled nondegradative lysosomal and endocytic compartments in
neurons. *The Journal of cell biology* **217**, 3127-3139, doi:10.1083/jcb.201711083 (2018).
- 3 Cheng, X. T. *et al.* Revisiting LAMP1 as a marker for degradative autophagy-lysosomal organelles in the nervous
system. *Autophagy* **14**, 1472-1474, doi:10.1080/15548627.2018.1482147 (2018).
- 4 Cadwell, K. & Debnath, J. Beyond self-eating: The control of nonautophagic functions and signaling pathways by
autophagy-related proteins. *The Journal of cell biology* **217**, 813-822, doi:10.1083/jcb.201706157 (2018).
- 5 Durgan, J. *et al.* Non-canonical autophagy drives alternative ATG8 conjugation to phosphatidylserine. *Molecular
cell* **81**, 2031-2040 e2038, doi:10.1016/j.molcel.2021.03.020 (2021).
- 6 Jacquin, E. *et al.* Pharmacological modulators of autophagy activate a parallel noncanonical pathway driving
unconventional LC3 lipidation. *Autophagy* **13**, 854-867, doi:10.1080/15548627.2017.1287653 (2017).
- 7 Munz, C. Non-canonical functions of autophagy proteins in immunity and infection. *Molecular aspects of medicine*,
100987, doi:10.1016/j.mam.2021.100987 (2021).
- 8 Christoforidis, S., McBride, H. M., Burgoyne, R. D. & Zerial, M. The Rab5 effector EEA1 is a core component of
endosome docking. *Nature* **397**, 621-625, doi:10.1038/17618 (1999).
- 9 Compans, B. *et al.* NMDAR-dependent long-term depression is associated with increased short term plasticity
through autophagy mediated loss of PSD-95. *Nature communications* **12**, 2849, doi:10.1038/s41467-021-23133-9
(2021).
- 10 Nikolettou, V., Sidiropoulou, K., Kallergi, E., Dalezios, Y. & Tavernarakis, N. Modulation of Autophagy by BDNF
Underlies Synaptic Plasticity. *Cell metabolism* **26**, 230-242 e235, doi:10.1016/j.cmet.2017.06.005 (2017).
- 11 Mancias, J. D., Wang, X., Gygi, S. P., Harper, J. W. & Kimmelman, A. C. Quantitative proteomics identifies NCOA4
as the cargo receptor mediating ferritinophagy. *Nature* **509**, 105-109, doi:10.1038/nature13148 (2014).
- 12 Shen, H., Zhu, H., Panja, D., Gu, Q. & Li, Z. Autophagy controls the induction and developmental decline of NMDAR-
LTD through endocytic recycling. *Nature communications* **11**, 2979, doi:10.1038/s41467-020-16794-5 (2020).
- 13 Tsien, J. Z. *et al.* Subregion- and cell type-restricted gene knockout in mouse brain. *Cell* **87**, 1317-1326,
doi:10.1016/s0092-8674(00)81826-7 (1996).
- 14 Tang, G. *et al.* Loss of mTOR-dependent macroautophagy causes autistic-like synaptic pruning deficits. *Neuron* **83**,
1131-1143, doi:10.1016/j.neuron.2014.07.040 (2014).
- 15 Ferreira, A. C. *et al.* Lipocalin-2 regulates adult neurogenesis and contextual discriminative behaviours. *Molecular
psychiatry* **23**, 1031-1039, doi:10.1038/mp.2017.95 (2018).
- 16 Lee, H. J., Yoon, Y. S. & Lee, S. J. Mechanism of neuroprotection by trehalose: controversy surrounding autophagy
induction. *Cell death & disease* **9**, 712, doi:10.1038/s41419-018-0749-9 (2018).
- 17 Yoon, Y. S. *et al.* Is trehalose an autophagic inducer? Unraveling the roles of non-reducing disaccharides on
autophagic flux and alpha-synuclein aggregation. *Cell death & disease* **8**, e3091, doi:10.1038/cddis.2017.501
(2017).
- 18 Debanne, D., Gahwiler, B. H. & Thompson, S. M. Asynchronous pre- and postsynaptic activity induces associative
long-term depression in area CA1 of the rat hippocampus in vitro. *Proceedings of the National Academy of Sciences
of the United States of America* **91**, 1148-1152, doi:10.1073/pnas.91.3.1148 (1994).
- 19 Mellentin, C., Moller, M. & Jahnsen, H. Properties of long-term synaptic plasticity and metaplasticity in organotypic
slice cultures of rat hippocampus. *Experimental brain research* **170**, 522-531, doi:10.1007/s00221-005-0236-2
(2006).
- 20 Izumi, Y., Auberson, Y. P. & Zorumski, C. F. Zinc modulates bidirectional hippocampal plasticity by effects on NMDA
receptors. *The Journal of neuroscience : the official journal of the Society for Neuroscience* **26**, 7181-7188,
doi:10.1523/JNEUROSCI.1258-06.2006 (2006).
- 21 Liu, L. *et al.* Role of NMDA receptor subtypes in governing the direction of hippocampal synaptic plasticity. *Science*
304, 1021-1024, doi:10.1126/science.1096615 (2004).
- 22 Colledge, M. *et al.* Ubiquitination regulates PSD-95 degradation and AMPA receptor surface expression. *Neuron*
40, 595-607, doi:10.1016/s0896-6273(03)00687-1 (2003).

- 23 Rimmelink, E. *et al.* Cognitive flexibility deficits in a mouse model for the absence of full-length dystrophin. *Genes Brain Behav* **15**, 558-567, doi:10.1111/gbb.12301 (2016).
- 24 Kononenko, N. L. *et al.* Retrograde transport of TrkB-containing autophagosomes via the adaptor AP-2 mediates neuronal complexity and prevents neurodegeneration. *Nature communications* **8**, 14819, doi:10.1038/ncomms14819 (2017).
- 25 Ravikumar, B., Moreau, K., Jahreiss, L., Puri, C. & Rubinsztein, D. C. Plasma membrane contributes to the formation of pre-autophagosomal structures. *Nature cell biology* **12**, 747-757, doi:10.1038/ncb2078 (2010).
- 26 Kuijpers, M. *et al.* Neuronal Autophagy Regulates Presynaptic Neurotransmission by Controlling the Axonal Endoplasmic Reticulum. *Neuron* **109**, 299-313 e299, doi:10.1016/j.neuron.2020.10.005 (2021).

Reviewers' Comments:

Reviewer #1:

Remarks to the Author:

The authors did a great job addressing all my comments in a adequate manner by performing numerous new experiments, re-analyzing their data and re-phrasing their manuscript. I do not have any further concerns and I am happy to recommend to accept this manuscript for publication. Well done!

Reviewer #2:

Remarks to the Author:

The manuscript by Kallergi, Daskalaki et al. examines the function of the autophagic pathway in long-term depression in the hippocampus. Pharmacological and genetic manipulations combined with imaging, biochemistry and electrophysiology in cultured neurons and brain slices, as well as behavioural measurements, provide convergent evidence for the necessity of autophagy for LTD. The authors have made a substantial and commendable effort to address the initial comments I had for the manuscript, and the multitude of new experiments that have been presented in the rebuttal have greatly strengthened the manuscripts' conclusions. As such, I recommend that the revised manuscript be accepted pending minor changes to the text as follows:

i) I appreciate the authors incorporating discussion of a previous paper with contradictory results (Shen et al., 2020, Nature Communications) as requested in the previous round of comments, however I do still think that a more nuanced discussion is required given the directly opposing results. The authors note that the deletion of Atg5 from pyramidal neurons in Shen et al. coincides with a developmental pruning window demonstrated to be autophagy-dependent by Tang et al., 2014, Neuron (ref. 20 in manuscript), and as such the results of Shen et al. could be confounded by a pruning defect. While this is possible, there is no direct evidence as Tang et al. examined layer 5 pyramidal neurons in the cortex, not CA1 hippocampal neurons. As the dynamics of dendritic spine turnover is different between the cortex and hippocampus (e.g. Attardo et al., 2015, Nature), it cannot be said for certain that there must be a pruning deficit at the time of measurement in Shen et al. without direct experimental evidence. Further, when mentioning the pitfalls of pharmacological approaches, only issues with the two inducers of autophagy were mentioned, which does not explain how 3 separate inhibitors of autophagy were sufficient to induce LTD. Finally, neither of the above explanations provided by the authors account for the fact that while their manuscript reports an increase in autophagy-related proteins and structures following LTD, Shen et al. report a decrease in the same proteins in the same time window. As such, I think the discussion as written is somewhat misleading to the reader as it implies the contradictory findings of Shen et al. were the result of poor choice of genetic model and pharmacological tools, which I do not think fully encapsulates the possible reasons for the differences in results. While this is not to say that the authors' results are incorrect or incomplete, there may be different conditions (be they experimental or true biological) that can lead to these disparate outcomes that the authors should discuss in more detail.

ii) In their rebuttal, the authors disputed a comment in my initial review, that it is not common in the literature to see total protein level of AMPARs decrease rapidly following LTD. I appreciate the cited literature to the contrary, as this highlights that there are likely conditions that do or do not favour AMPAR protein loss, given there have been multiple papers which do not find AMPAR protein loss following LTD, including the original paper describing the chemical NMDA-LTD approach (Lee et al., 1998, Neuron). As such, I ask that the authors make note of these two possible outcomes following LTD as it will give the reader better context as well as potentially being a crucial factor in determining whether LTD does or does not involve autophagic protein degradation.

Reviewer #3:

Remarks to the Author:

Kallergi et al in their revised Ms have added a substantial amount of new data that have helped to clarify the points raised in my review and to place the study into the context of recent works. I am thus happy to endorse publication of this thorough and important study in Nat Commun. I have only two minor points that can be addressed by textual revision of the final Ms.

1. Wortmannin is a broad spectrum PIK inhibitor that targets essentially all PI3Ks. While there are several specific Vps34 inhibitors available by now (e.g. SAR405, VPS34IN1, Cpd19) that could have been used for these experiments, I suggest to at least mention the caveats associated with Wortmannin in the paper.

2. A short discussion regarding possible explanations for the observed facilitation of atg5-cKO or SBI-treated synapses in response to LFS, NMDA or DHPG and its potential effects on reversal learning would be helpful.

POINT-BY-POINT RESPONSE TO REVIEWERS' COMMENTS

Reviewer #1:

The authors did a great job addressing all my comments in a adequate manner by performing numerous new experiments, re-analyzing their data and re-phrasing their manuscript. I do not have any further concerns and I am happy to recommend to accept this manuscript for publication. Well done!

We thank the reviewer for his/her positive comments and for recommending our manuscript for publication. His/her comments were very constructive and crucial for improving our work.

Reviewer #2:

The manuscript by Kallergi, Daskalaki et al. examines the function of the autophagic pathway in long-term depression in the hippocampus. Pharmacological and genetic manipulations combined with imaging, biochemistry and electrophysiology in cultured neurons and brain slices, as well as behavioural measurements, provide convergent evidence for the necessity of autophagy for LTD. The authors have made a substantial and commendable effort to address the initial comments I had for the manuscript, and the multitude of new experiments that have been presented in the rebuttal have greatly strengthened the manuscripts' conclusions. As such, I recommend that the revised manuscript be accepted pending minor changes to the text as follows:

We thank the reviewer for appreciating our effort to address his/her comments and for recommending our revised manuscript to be accepted for publication, pending minor changes. We are also thankful as these comments helped us to improve our work and place it well in the context of recent studies in the field.

i) I appreciate the authors incorporating discussion of a previous paper with contradictory results (Shen et al., 2020, Nature Communications) as requested in the previous round of comments, however I do still think that a more nuanced discussion is required given the directly opposing results. The authors note that the deletion of Atg5 from pyramidal neurons in Shen et al. coincides with a developmental pruning window demonstrated to be autophagy-dependent by Tang et al., 2014, Neuron (ref. 20 in manuscript), and as such the results of Shen et al. could be confounded by a pruning defect. While this is possible, there is no direct evidence as Tang et al. examined layer 5 pyramidal neurons in the cortex, not CA1 hippocampal neurons. As the dynamics of dendritic spine turnover is different between the cortex and hippocampus (e.g. Attardo et al., 2015, Nature), it cannot be said for certain that there must be a pruning deficit at the time of measurement in Shen et al. without direct experimental evidence. Further, when mentioning the pitfalls of pharmacological approaches, only issues with the two inducers of autophagy were mentioned, which does not explain how 3 separate inhibitors of autophagy were sufficient to induce LTD. Finally, neither of the above

explanations provided by the authors account for the fact that while their manuscript reports an increase in autophagy-related proteins and structures following LTD, Shen et al. report a decrease in the same proteins in the same time window. As such, I think the discussion as written is somewhat misleading to the reader as it implies the contradictory findings of Shen et al. were the result of poor choice of genetic model and pharmacological tools, which I do not think fully encapsulates the possible reasons for the differences in results. While this is not to say that the authors' results are incorrect or incomplete, there may be different conditions (be they experimental or true biological) that can lead to these disparate outcomes that the authors should discuss in more detail.

We thank the reviewer for raising these points, which help us to further refine our discussion of the contradictory findings. We have modified the discussion extensively (please see the highlighted text on page 19-20 of the revised manuscript) to address the points raised:

a) When discussing the possibility that the results of Shen *et al.*,¹ may be confounded by pruning deficits, it was not our intention to claim that this is for certain. This is evidenced by the use of speculative expressions shown below in bold. Moreover, the sentence has been modified (highlighted) to gain specificity, now reading “Given the well-described requirement of autophagy in this process **in the cortex**², these animals **are likely to carry developmental deficits in spine pruning that can alter their LTD response and may explain the different results. However, as pruning periods may vary across different brain areas, this possibility would need to be experimentally examined**”.

b) The reviewer raises a very important point by referring to the 3 different inhibitors of autophagy which were shown to induce LTD in the Shen *et al.*, study. These are Chloroquine (elevates the pH of acidic compartments), Leupeptin (protease inhibitor) and BafilomycinA1 (a vacuolar-type H⁺-translocating ATPase [V-ATPase] inhibitor). It's worth clarifying that all these compounds act on lysosomes and not on the biogenesis of autophagic vesicles. It must be emphasized that in addition to the macroautophagic input, other processes (e.g., endosomal, phagosomal, chaperone-mediated) also carry cargo to the lysosomes and are impacted by these treatments. (Please also see the Guidelines for the use and interpretation of assays for monitoring autophagy (4th edition), 2021, Autophagy). In our study we have not at all considered or investigated the role of lysosomes in LTD and instead chose to inhibit the early steps of autophagic vesicle biogenesis. Therefore, the different results may stem from targeting different steps of the autophagic pathway, in our case the autophagic cargo recognition machinery, while in the Shen *et al* study the autophagic cargo degradation machinery. This is an interesting point which will require further investigation in the future and which we have now included in our discussion.

c) With regards to the opposite results when examining the levels of LC3-II and of LC3-positive puncta in dendrites shortly after LTD, the experimental paradigms used in the two studies are very different. **(1) Imaging experiments:** The Shen *et al.*, study uses cultured hippocampal slices (7 days in vitro) which are prepared from P6-8 animals and which are biolistically transfected to over-express mRFP-tagged LC3 (mRFP-LC3), mRFP-LC3G120A (lipidation-defective LC3 mutant that cannot be incorporated into autophagosomes), or GFP tagged p62. At 3–5 days after transfection autophagy is monitored after inducing LTD (LFS; 1 Hz, 900 pulses). Instead, in our work, we monitor endogenous LC3-puncta in cultured neurons after chemical LTD (NMDA or DHPG pulse). We also monitor autophagic vesicles by electron microscopy in neurobiotin-labeled CA1 *s. radiatum* dendrites in acute hippocampal slices after LTD. For the latter, it's also worth noting that the age of the animals used is different (P16-19 for acute or P6-8 for cultured hippocampal slices in the Shen *et al.*, study, versus P22-P28 in our study). **(2) Biochemical experiments:** We use cultured neurons after cLTD, whereas the Shen *et al* study removed the CA1 area near the recording electrode.

In our view, it's very likely that these different parameters collectively account for the different observations. We have included a sentence in the discussion mentioning the different experimental paradigms, tools and age of mice between the two studies, without going into the details of all experimental differences (due to space limitations).

ii) In their rebuttal, the authors disputed a comment in my initial review, that it is not common in the literature to see total protein level of AMPARs decrease rapidly following LTD. I appreciate the cited literature to the contrary, as this highlights that there are likely conditions that do or do not favour AMPAR protein loss, given there have been multiple papers which do not find AMPAR protein loss following LTD, including the original paper describing the chemical NMDA-LTD approach (Lee *et al.*, 1998, Neuron). As such, I ask that the authors make note of these two possible outcomes following LTD as it will give the reader better context as well as potentially being a crucial factor in determining whether LTD does or does not involve autophagic protein degradation.

It may well be that autophagy is implicated specifically under conditions where LTD is associated with a decrease in the levels of AMPARs, while this may not be the case if LTD is not associated with decreasing AMPAR levels. In our experimental protocols we observe a rapid decrease in the levels of AMPARs and of PSD95 (as also recently reported in Compans *et al.*, 2021)³. It would be interesting to investigate this hypothesis in the future. Therefore, this point is well taken and we have now included it in the discussion as a possible explanation for the different findings between our study and that of the Zheng Li lab.

Reviewer #3:

Kallergi et al in their revised Ms have added a substantial amount of new data that have helped to clarify the points raised in my review and to place the study into the context of recent works. I am thus happy to endorse publication of this thorough and important study in Nat Commun. I have only two minor points that can be addressed by textual revision of the final Ms.

We thank the reviewer for acknowledging the newly added data as substantial, for his/her positive comments on our study and for endorsing our work for publication. His/her comments were very important for improving our work and we are thankful for that.

1. Wortmannin is a broad spectrum PIK inhibitor that targets essentially all PI3Ks. While there are several specific Vps34 inhibitors available by now (e.g. SAR405, VPS34IN1, Cpd19) that could have been used for these experiments, I suggest to at least mention the caveats associated with Wortmannin in the paper.

We have now clarified in the text (page 4) that Wortmannin is a broad PI3K inhibitor (not specific for Vps34).

2. A short discussion regarding possible explanations for the observed facilitation of atg5-cKO or SBI-treated synapses in response to LFS, NMDA or DHPG and its potential effects on reversal learning would be helpful.

We agree with the reviewer that it's important to speculate on the mechanisms that may be involved in the facilitation induced by LTD stimuli in neurons with autophagy impairment. Our foremost hypothesis is that autophagy deficiency may alter the levels of key molecules involved in synaptic plasticity. We were in fact intrigued that the proteomic analysis indicates that CamKII proteins, as well as calmodulin and several phosphatases are contained in the purified AV fraction (please see table 1). Both pharmacological and genetic experiments have shown that activation of CamKII is necessary for sustained potentiation and LTP induction⁴⁻⁶. In fact, active CaMKII can mimic LTP⁷⁻¹¹. Taken together, these studies suggest that CamKII can activate the biochemistry that underlies sustained potentiation. Therefore, one possibility, is that autophagy ablation leads to changes in the levels or activation of CamKII proteins that may change the response of postsynaptic neurons to calcium levels during plasticity. We have now included a paragraph in the discussion on this (please see the highlighted text on page 22 of the revised manuscript).

- 1 Shen, H., Zhu, H., Panja, D., Gu, Q. & Li, Z. Autophagy controls the induction and developmental decline of NMDAR-LTD through endocytic recycling. *Nature communications* **11**, 2979, doi:10.1038/s41467-020-16794-5 (2020).
- 2 Tang, G. *et al.* Loss of mTOR-dependent macroautophagy causes autistic-like synaptic pruning deficits. *Neuron* **83**, 1131-1143, doi:10.1016/j.neuron.2014.07.040 (2014).
- 3 Compans, B. *et al.* NMDAR-dependent long-term depression is associated with increased short term plasticity through autophagy mediated loss of PSD-95. *Nature communications* **12**, 2849, doi:10.1038/s41467-021-23133-9 (2021).
- 4 Giese, K. P., Fedorov, N. B., Filipkowski, R. K. & Silva, A. J. Autophosphorylation at Thr286 of the alpha calcium-calmodulin kinase II in LTP and learning. *Science* **279**, 870-873, doi:10.1126/science.279.5352.870 (1998).
- 5 Malinow, R., Schulman, H. & Tsien, R. W. Inhibition of postsynaptic PKC or CaMKII blocks induction but not expression of LTP. *Science* **245**, 862-866, doi:10.1126/science.2549638 (1989).
- 6 Otmakhov, N., Griffith, L. C. & Lisman, J. E. Postsynaptic inhibitors of calcium/calmodulin-dependent protein kinase type II block induction but not maintenance of pairing-induced long-term potentiation. *The Journal of neuroscience : the official journal of the Society for Neuroscience* **17**, 5357-5365 (1997).
- 7 Hayashi, Y. *et al.* Driving AMPA receptors into synapses by LTP and CaMKII: requirement for GluR1 and PDZ domain interaction. *Science* **287**, 2262-2267, doi:10.1126/science.287.5461.2262 (2000).
- 8 Jourdain, P., Fukunaga, K. & Muller, D. Calcium/calmodulin-dependent protein kinase II contributes to activity-dependent filopodia growth and spine formation. *The Journal of neuroscience : the official journal of the Society for Neuroscience* **23**, 10645-10649 (2003).
- 9 Lledo, P. M. *et al.* Calcium/calmodulin-dependent kinase II and long-term potentiation enhance synaptic transmission by the same mechanism. *Proceedings of the National Academy of Sciences of the United States of America* **92**, 11175-11179, doi:10.1073/pnas.92.24.11175 (1995).
- 10 Pettit, D. L., Perlman, S. & Malinow, R. Potentiated transmission and prevention of further LTP by increased CaMKII activity in postsynaptic hippocampal slice neurons. *Science* **266**, 1881-1885, doi:10.1126/science.7997883 (1994).
- 11 Poncer, J. C., Esteban, J. A. & Malinow, R. Multiple mechanisms for the potentiation of AMPA receptor-mediated transmission by alpha-Ca²⁺/calmodulin-dependent protein kinase II. *The Journal of neuroscience : the official journal of the Society for Neuroscience* **22**, 4406-4411, doi:20026449 (2002).

Reviewers' Comments:

Reviewer #2:

Remarks to the Author:

The authors have done a great job addressing my discussion concerns, and as such I recommend accepting the manuscript for publication with no further changes.

Reviewer #3:

None